**Surface Energy Balance Sensitivity to Meteorological Variability on Haig Glacier, Canadian Rocky Mountains**

S. Ebrahimi and S. J. Marshall

Department of Geography, University of Calgary, 2500 University Drive, NW, Calgary, Alberta T2N 1N4, Canada

*Correspondence to:* S. Ebrahimi (samaneh.ebrahimi@ucalgary.ca)

**Abstract**

Energy exchanges between the atmosphere and the glacier surface control the net energy available for snow and ice melt. This paper explores the response of a mid-latitude glacier in the Canadian Rocky Mountains to daily and interannual variations in the meteorological parameters that govern the surface energy balance. We use an energy balance model to run sensitivity tests to perturbations in temperature, specific humidity, wind speed, incoming shortwave radiation, glacier surface albedo, and winter snowpack depth. Variables are perturbed (i) in isolation, (ii) including internal feedbacks, and (iii) with co-evolution of meteorological perturbations, derived from the North American regional climate reanalysis (NARR) over the period 1979-2014. Summer melt at this site has the strongest sensitivity to interannual variations in temperature, albedo, and specific humidity, while fluctuations in cloud cover, wind speed, and winter snowpack depth have less influence. Feedbacks to temperature forcing, in particular summer albedo evolution, double the melt sensitivity to a temperature change. When meteorological perturbations co-vary through the NARR forcing, summer temperature anomalies remain important in driving interannual summer energy balance and melt variability, but they are reduced in importance relative to an isolated temperature forcing. Covariation of other variables (e.g., clear skies, giving reduced incoming longwave radiation) may be partially compensating for the increase in temperature. The methods introduced in this paper provide a framework that can be extended to compare the sensitivity of glaciers in different climate regimes, e.g., polar, maritime, or tropical environments, and to assess the importance of different meteorological parameters in different regions.

## 1. Introduction

Glaciers and icefields are thinning and retreating in all of the world's mountain regions in response to global climate change (e.g., Marzeion et al., 2014). This is reshaping alpine environments, affecting regional water resources, and contributing to global sea level rise (e.g., Radić and Hock, 2011). A glacier's climate sensitivity can be expressed in terms of the energy or mass balance response to a change in meteorological conditions (Oerlemans and Fortuin, 1992; Oerlemans et al., 1998). For instance, Oerlemans et al. (1998) defined the static glacier sensitivity to temperature, $S_T$, i as:

$$S_T = \frac{\partial B_m}{\partial T} \approx \frac{B_m\ (+1K) - B_m\ (-1K)}{2} \qquad (1)$$

where $B_m(\delta T)$ denotes the mean specific mass balance corresponding to the temperature perturbation $\delta T$. Mass balance sensitivity to precipitation perturbations, $S_P = \partial B_m / \partial P$, can be calculated in the same way.

Braithwaite and Raper (2002) extended the static sensitivity approach to regional scales, with the idea that glaciers within a given climate regime should have similar mass balance sensitivities to variations in temperature and precipitation. This framework has been used in numerous studies to describe glacier sensitivity to climate change (e.g., Dyurgerov 2001; Klok and Oerlemans, 2004; Arendt et al., 2009; Anderson et al., 2010; Engelhardt et al., 2015).

Most studies to date have concentrated on glacier mass balance response to changes in temperature and precipitation. This is sensible, as these are generally the most important meteorological variables affecting glacier mass balance. These two fields are also commonly measured, with long-term records available in many regions. Temperature and precipitation have also received the most attention because regional- to global-scale models of glacier mass balance commonly employ temperature-index methods to parameterize glacier melt (e.g., Marzeion et al., 2014; Clarke et al., 2015), with only these variables as inputs.

While temperature index models have demonstrated reasonable skill in estimating seasonal melt (Ohmura, 2001; Hock, 2005), they are nonetheless missing much of the physics that govern melt. Also, they may be overly sensitive to changes in temperature, without effectively capturing the impact of shifts in other variables such as wind, humidity, or cloud cover. Internal processes and feedbacks, such as surface albedo evolution, may also be absent, since degree-day melt factors are usually taken to be static. Such feedbacks are critical to glacier melt (e.g., Brock et al., 2000; Klok and Oerlemans, 2004; Cuffey and Paterson, 2010).

It is uncertain whether variability in glaciometeorological variables other than temperature and precipitation is important to glacier energy and mass balance. While most large-scale glacier change projections are rooted in temperature sensitivity (as built into temperature-index models), it is generally recognized that the complete surface energy balance is important to glacier melt. For instance, net radiation has been identified as the main source of melt energy for continental glaciers, accounting for ~70-80% of the total melt energy (e.g., Greuell and Smeets, 2001; Oerlemans and Klok, 2002; Klok et al., 2005; Giesen et al., 2008), with shortwave radiation providing the principal energy source. Incoming shortwave radiation is not directly dependent on

temperature. As another example, latent heat fluxes are a significant source of energy in maritime
and tropical environments (Wagnon et al., 1999, 2003; Favier et al., 2004; Anderson et al., 2010),
and their strength is a function of humidity and wind conditions, which are not strongly correlated
with temperature fluctuations. This calls for a broader exploration of glacier sensitivity to climate
variability and change, beyond just the influence of temperature.
Several studies that estimate glacier sensitivity to temperature change use complete models of
energy balance (e.g., Klok and Oerlemans, 2004; Klok et al., 2005; Anslow et al., 2008; Anderson
et al., 2010). The influence of other meteorological variables has been explored in a few studies.
Gerbaux et al. (2005) examine the role of different variables (e.g., temperature, moisture, wind) in
energy balance processes and climate sensitivity in the French Alps. Giesen et al. (2008) note the
importance of cloud cover in modulating interannual variability in summer melt on Midtdalsbreen,
Norway. Sicart et al. (2008) examine three glaciers in different latitudes/climate regimes.
Variations in net shortwave radiation, sensible heat flux, and temperature each contribute to
differences in glacier sensitivity to climate variability between these locations.
We build on these studies through a systematic examination of glacier energy balance and melt
sensitivity. We report the mean melt season conditions on Haig glacier in the Canadian Rocky
Mountains for the period 2002-2012. These reference data are used as a baseline for theoretical
and numerically modelled sensitivity. The same perturbation approach is then used to reconstruct
variations in surface energy balance and melt for the period 1979-2014, based on North American
regional climate reanalyses (NARR) (Mesinger et al., 2006). Our main question is whether
variables other than temperature and precipitation need to be considered to provide a realistic
estimate of glacier sensitivity to climate change for mid-latitude mountain glaciers. Our analysis
in this study is limited to just one site, with a focus on the summer melt season (vs. annual mass
balance). We examine the summer energy balance and evaluate the impact of different variables
in isolation and with more realistic covariance of meteorological conditions.

## 2. Surface Energy Balance and Melt Model

The energy budget at the glacier surface is defined by the fluxes of energy between the atmosphere,
the snow/ice surface, and the underlying snow or ice. The surface energy balance can be written

$$Q_N = Q_S^\downarrow(1 - \alpha) + Q_L^\downarrow - Q_L^\uparrow + Q_H + Q_E + Q_C, \qquad (2)$$

where $Q_N$ is the net energy flux at the surface and $Q_S^\downarrow, Q_L^\downarrow, Q_L^\uparrow, Q_H, Q_E,$ and $Q_C$ represent incoming
shortwave radiation, incoming and outgoing longwave radiation, sensible and latent heat flux, and
subsurface conductive energy flux, respectively. The energy fluxes have units of W m$^{-2}$. The
surface albedo is denoted $\alpha$ and fluxes are defined to be positive when they are sources of energy
to the glacier surface. We neglect the penetration of shortwave radiation and advection of energy
by precipitation and meltwater fluxes.

The net energy $Q_N$ can be positive or negative. When it is negative, as it is for much of the winter
and during the night, the snow or ice will cool or liquid water will refreeze. Positive net energy
will drive surface warming, or on a melting glacier surface with $Q_N > 0$, the net energy flux is
dedicated to generating surface melt. For melt rate $\dot{m}$, this follows

$$\dot{m} = \frac{Q_N}{\rho_w L_f}, \tag{3}$$

where $\rho_w$ is the density of water and $L_f$ is the latent heat of fusion. Melt rates in Eq. (3) have units
of metres water equivalent per second (m w.e. $s^{-1}$).
Numerous studies have shown that incoming shortwave radiation is the dominant term in the
energy balance during the melt season in most glacial environments. Incoming shortwave radiation
(insolation) at the surface has three components: direct and diffuse solar radiation, along with
direct solar radiation that is reflected from the surrounding terrain. Direct solar radiation is the
radiative flux from the direct solar beam, which comes in at a zenith angle $Z$. It is a function of
latitude, time of year, and time of day (e.g., Oke, 1987). Potential direct (clear-sky) incoming solar
radiation on a horizontal surface can be estimated from

$$Q_\phi^\downarrow = Q_0 \cos(Z)\, \varphi_0^{P/P_0 \cos(Z)}, \tag{4}$$

for top-of-atmosphere insolation $Q_0$, clear-sky atmospheric transmissivity $\varphi_0$, air pressure $P$, and
sea-level air pressure $P_0$ (Oke, 1987). Eq. (4) allows potential direct shortwave radiation to be
calculated as a function of the day, year, latitude and elevation.
Longwave radiation can be estimated from the Stefan-Boltzmann equation,

$$Q_L = \varepsilon \sigma T^4, \tag{5}$$

where $\varepsilon$ is the thermal emissivity, $\sigma$ is the Stefan--Boltzmann constant, and $T$ is the absolute
temperature of the emitting surface. Snow and ice emit as near-perfect blackbodies at infrared
wavelengths, with surface emissivity $\varepsilon_s = 0.98\text{-}1.0$. The longwave fluxes are then

$$Q_L^\uparrow = \varepsilon_s \sigma T_s^4, \tag{6}$$

and

$$Q_L^\downarrow = \varepsilon_a \sigma T_a^4, \tag{7}$$

for surface temperature $T_s$, near-surface air temperature $T_a$, and atmospheric emissivity $\varepsilon_a$. Terrain
emissions (i.e. from the surrounding topography) can also contribute to the incoming longwave
radiation, particularly at sites that are adjacent to valley walls.
A spectrally- and vertically-integrated radiative transfer calculation is needed to predict the
incoming longwave radiation from the atmosphere, as this depends on lower-troposphere water
vapour, cloud, and temperature profiles. Because the requisite atmospheric data are rarely available
in glacial environments, $Q_L^\downarrow$ is commonly parameterized at a site as a function of local (2-m)
temperature and humidity. Where available, cloud cover or a proxy for cloud conditions, such as
the atmospheric clearness index, are often used to strengthen this parameterization. Hock (2005)
and Lhomme et al. (2007) provide reviews of some of the parameterizations of atmospheric
emissivity that have been employed in glaciology. We found good results for regression-based
parameterization at two study sites in the Canadian Rocky Mountains (Ebrahimi and Marshall,
159 2015),

$$Q_L^{\downarrow} = (a + be_v + ch)\, \sigma T_a^4 \qquad (8)$$

and

$$Q_L^{\downarrow} = (a + be_v + c\tau)\, \sigma T_a^4, \qquad (9)$$

Here $a$, $b$, and $c$ are regression parameters (different in Eqs. (8) and (9)), $e_v$ is vapour pressure, $h$
is relative humidity, and $\tau$ is the clearness index, calculated from the ratio of measured to potential
direct incoming shortwave radiation.
Solar radiation and cloud data are less commonly available than relative humidity, so Eq. (8) is a
slightly less accurate but more portable version of this parameterization (Ebrahimi and Marshall,
2015). Multiple regressions of $\varepsilon_a$ containing both relative humidity and clearness index were
rejected, as these are highly (negatively) correlated. All-sky longwave parameterizations using
either of these variables are reasonable, with root-mean square errors in mean daily incoming
longwave radiation of about 10 W/m$^2$.
Relative humidity can also be used as a proxy for clearness index if shortwave radiation data are
not available. Summer (JJA) observations at Haig Glacier follow the relation:

$$\tau = 1.3 - 0.01h\,, \qquad (10)$$

for mean daily values of $\tau$ and $h$ ($R^2 = 0.5$). We draw on this below when we need to estimate
perturbations in sky clearness index that are consistent with changes in atmospheric humidity. In
accord with the observational basis of Eq. (10), the clearness index is constrained to be within 0.3
and 1 ($h \in [30, 100\%]$); if daily mean humidity drops below this, we set $\tau = 1$.
Turbulent fluxes of sensible and latent energy in the glacier boundary layer are parameterized from
a bulk aerodynamic method (e.g., Andreas, 2002):

$$Q_H = \rho_a c_p k^2 v \left[ \frac{T_a(z) - T_s}{\ln(^z/_{z_0}) \ln(^z/_{z_{0H}})} \right], \qquad (11)$$

and

$$Q_E = \rho_a L_v k^2 v \left[ \frac{q_a(z) - q_s}{\ln(^z/_{z_0}) \ln(^z/_{z_{0E}})} \right]. \qquad (12)$$

Here $\rho_a$ is the air density, $c_p$ is the specific heat capacity of air, $L_v$ is the latent heat of evaporation,
$k = 0.4$ is von Karman's constant, $v$ is wind speed, and $q$ refers to the specific humidity.
Measurements of temperature and humidity are assumed to be at two levels, height $z$ (e.g., 2 m)
and at the surface-air interface, $s$. For a melting glacier surface, $T_s = 0°C$, and $q_s$ can be taken from
the saturation specific humidity over ice at temperature $T_s$. We estimate $T_s$ from an inversion of
Eq. (6), using measurements of outgoing longwave radiation. In sensitivity tests, where we depart
from the observational constraints, $T_s$ is internally modelled within a subsurface snow model (see
below), taken from the temperature of the upper snow layer.

Parameters $z_0$, $z_{0H}$, and $z_{0E}$ refer to the roughness length scales for turbulent exchange of
momentum, heat, and moisture. We adopt fixed values for each, equivalent for both snow and ice
($z_0 = 3$ mm; $z_{0H} = z_{0E} = z_0/100$), based on closure of the surface energy balance with reference to
observed melt (Marshall, 2014). Atmospheric stability adjustments can be introduced in Eqs. (11)
and (12) to modify the turbulent flux parameterizations for the stable glacier boundary layer (e.g.,
Hock and Holmgren, 2005; Giesen et al., 2008). We do not apply stability corrections, as we are
able to attain closure in modelled and measured summer melt at this site without this. Others have
argued that stability corrections may lead to an underestimation of the turbulent fluxes on mountain
glaciers (e.g. Hock and Holmgren, 2005). This may be related to the low-level wind speed
maximum that is typical of the glacier boundary layer, which introduces strong turbulence and is
not consistent with the logarithmic profile of wind speed that is implicit in Eqs. (11) and (12). It
may also be that the effects of atmospheric stability are absorbed in the roughness values –
roughness values that are adopted to attain closure in the surface energy balance and melt
calculations may be too low, implicitly accounting for the stable boundary layer.

Subsurface temperatures are modelled through a multi-layer, one-dimensional model of heat
conduction and meltwater percolation and refreezing in the upper 10 m of the glacier, the
approximate depth of penetration of the annual temperature wave (Cuffey and Paterson, 2010).
This depth includes the time-varying seasonal snow layer and the underlying firn or ice. The
temperature solution follows

$$\rho_s c_s \frac{\partial T}{\partial t} = \frac{\partial}{\partial z}\left(-k_t \frac{\partial T}{\partial z}\right) + \varphi_t, \tag{13}$$



where $\rho_s$, $c_s$, and $k_t$ are the density, heat capacity, and thermal conductivity of the subsurface snow,
firn, or ice and $\varphi_t(z)$ is a local source term that accounts for latent heat of refreezing,

$$\varphi_t = \rho_w L_f \dot{r}/\Delta z. \tag{14}$$



The refreezing rate $\dot{r}$ has units m s$^{-1}$, $\varphi_t$ has units W m$^{-3}$, and $\Delta z$ is the thickness of the layer in
which the meltwater refreezes.

Refreezing is calculated from a hydrological model that is coupled with the subsurface thermal
model. We track the volumetric liquid water fraction, $\theta_w$, in the snow/firn pore space, and if
conductive energy loss occurs in a subsurface layer where liquid water is present, this energy is
diverted to latent enthalpy of freezing, rather than cooling the snow. Temperatures cannot drop
below 0°C until $\theta_w = 0$. Liquid water is converted to ice in the subsurface layer.

We model meltwater drainage by assuming that water percolates uniformly, with hydraulic
conductivity $k_h$ and neglecting horizontal transport (i.e. assuming only gravity-driven vertical
drainage). Local water layer thickness can be expressed $h_w = \theta_w \Delta z$. The local water balance is then

$$\frac{\partial h_w}{\partial t} = -k_h \frac{\partial h_w}{\partial z} - \dot{r}, \tag{15}$$



where the final term accounts for water that is removed through internal refreezing. In principle,
this is a source/sink term that could also include internal melting (e.g., from shortwave radiation
penetration or percolation of warm rainwater), but we do not consider these processes. We assume
an irreducible water content of 3% for the melting snowpack (Colbeck, 1974), and the maximum
volumetric water content is equal to the porosity, $\theta$, although drainage in the seasonal snowpack
is efficient and $\theta_w$ is always much less than $\theta$.

*Numerical Energy Balance and Subsurface Temperature Model*

For the energy balance sensitivity experiments in this study, we use a combination of directly
observed and modelled glaciometeorological variables. Where we report the directly observed
surface energy balance, for the 2002-2012 reference state, we drive the energy balance model with
observed 30-minute data, including measured albedo and longwave radiation fluxes. Turbulent
heat fluxes and subsurface heat conduction are modelled from Equations (11-15).
Where we do sensitivity tests or run the model with other meteorological input, such as from
climate models, we need to allow for internal feedbacks such as freely-determined albedo
evolution and changes in incoming radiation that will attend changes in atmospheric conditions
(e.g., cloud cover, humidity). The energy balance and melt model that we employ is based on daily
mean meteorological inputs, in order to make our approach compatible with output from climate
models or reanalyses, as well as parameterizations that operate on a daily timescale (Eqs. 8-10). A
parameterized diurnal cycle is introduced for temperature and shortwave radiation (see below), in
order to capture the effects of overnight refreezing and the fraction of the day that experiences melt
(when $Q_N$ and $T_s > 0$). The model uses a variable time step from 10 minutes to 1 hour to allow for
stability of the subsurface temperature prognosis.

The subsurface temperature model has 33 layers, with 10-cm layers until 0.6-m depth, 20-cm
layers from 0.6-2 m, and 40-cm layers from 2-10 m. The upper boundary forcing comes from the
conductive heat flux at the snow/ice-air interface, $Q_C = -k_t \partial T/\partial z$, modelled from a three-point
forward finite-difference approximation of $\partial T/\partial z$. We use a two-step solution, for the temperature
(Eq. 13), then the meltwater drainage (Eq. 15). The temperature solution is implicit for the
temperature diffusion, with latent heat release from refreezing (the source term in Eq. 13)
calculated from the previous time step within the hydrological model. Hydraulic conductivity in
Eq. (15) is assigned the value $k_h = 10^{-4}$ m s$^{-1}$, near the low end of estimates reported by Campbell
et al. (2006). Meltwater is assumed to drain instantaneously when it reaches the snow-ice interface.

The 10-m subsurface model consists of the seasonal snowpack of thickness $d_s(t)$, overlying either
firn or ice. The grid is fixed with respect to the surface, and each layer is assigned a density, thermal
conductivity, and heat capacity according to the medium (snow, firn, or ice). Snow and firn density
are modelled as a function of depth and the liquid water and ice content,

$$\rho_s = \rho_i(1 - \theta) + \rho_w\theta_w + \rho_i\theta_i \,, \tag{16}$$

for porosity $\theta$, liquid water fraction $\theta_w$, and ice fraction $\theta_i$. Densities $\rho_s$, $\rho_i$, and $\rho_w$ refer to snow,
ice and water, respectively. We prescribe a decrease in porosity with depth following $\theta(z) = 0.6 -$
0.05z, parameterized to represent the measured summer snow densities at the site ($\rho_s$ = 350-550
kg m$^{-3}$) and give reasonable estimates of firn density, up to $\rho_s$ = 820 kg m$^{-3}$ at 10-m depth.
Snow accumulates, melts, or undergoes densification on a daily time step, with snow thickness d
varying continuously (vs. discretely) within the fixed-grid framework. At depth $d$ below the
surface, the grid cell has a weighted combination of thermal properties and densities to reflect the
mixture of snow and either firn or ice in that layer. We do not have a model for snow
accumulation through the winter months. We treat this simply, and linearly accumulate snow
from the start of winter until the start of the following melt season, with the accumulation rate set
to give a match to the observed May snowpack thickness for each year. These data are available
through annual winter mass balance surveys on the glacier, including a snowpit that provides
depth and density measurements at the AWS site.
The steps in the energy balance and melt model are as follows:
1. Daily mean values are input for temperature, incoming shortwave and longwave radiation, air
pressure, specific humidity, and wind speed, as well as minimum and maximum temperature.
2. A diurnal temperature cycle is parameterized as a cosine wave with a lag $\tau_t$ = 4 hours to give
the maximum temperature at 16:00, as per local observations, with an amplitude $A_t = (T_{max} - T_{min})/2$
(Figure 1a). For time $t$ (hour of the day) and period $P_t$ = 24 hours,

$$T(t) = -A_t \cos\left[\frac{2\pi(t-\tau_t)}{P_t}\right]. \tag{17}$$

3. A diurnal cycle for incoming shortwave radiation is parameterized as a half-cosine wave with
a period $P_{sw}(d) = 2h_s(d)$, where $d$ is the day of year and $h_s$ is the number of hours of sunlight on
day $d$ (Figure 1b). Defining lag $\tau_{sw}$ and amplitude $A_{sw}$,

$$Q_s^\downarrow(t) = \max\left\{-A_{sw} \cos\left[\frac{2\pi(t-\tau_{sw})}{P_{sw}}\right], 0\right\}. \tag{18}$$

Sunlight hours are calculated as a function of latitude, $\theta$, and day of year, based on the equation
for the sunset hour $h_{ss}$ (e.g., Liou, 2002):

$$\cos(h_{ss}) = -\tan(\delta)\tan(\theta), \tag{19}$$

where $\delta$ is the solar declination angle (solar latitude as a function of day of year). Sunlight hours
$h_s = 2h_{ss}$. The lag also varies with the day of year, and is calculated by setting peak shortwave
radiation to occur at noon: $2\pi(12 - \tau_{sw})/P_{sw} = \pi$. This gives $\tau_{sw} = 12 - h_s$ hours. Amplitude $A_{sw}$ is
calculated by integrating the area under the cosine curve and equating this to the average daily
incoming shortwave radiation, $Q_{Sd}^\downarrow$. This gives $A_{sw} = 12\pi Q_{Sd}^\downarrow/h_s$ W m$^{-2}$. This treatment implicitly
includes cloud effects that reduce incoming shortwave radiation on a given day (via $Q_{Sd}^\downarrow$), but
distributed evenly through the day. This neglects any systematic tendency for afternoon vs
morning clouds. For simplicity, we also neglect the effect of zenith angle on atmospheric
transmittance (i.e., lower transmittance for larger atmospheric path lengths in the morning and late
afternoon), although this could be built into a more refined model.
4. We assume that wind, incoming longwave radiation, air pressure, and specific humidity are
constant through the day, held to the mean daily value. For sensitivity tests, $Q_L^{\downarrow}$ is calculated
following Eq. (8) and the daily mean value of $Q_S^{\downarrow}$ is perturbed from Eq. (10) and $dQ_S^{\downarrow} = d\tau$.
5. Relative humidity has a diurnal cycle following temperature, assuming constant daily humidity
but adjusting $h$ for consistency with the effect of temperature on saturation vapour pressure.
6. Albedo is also modelled on a daily basis for the sensitivity studies. When the seasonal snowpack
is melted away, albedo is set to the observed bare-ice value at the site, $\alpha_i = 0.25$. For fresh or dry
snow, a fixed value $\alpha_0 = 0.86$ is used. The snowpack thickness is initialized on May 1 of each year,
set to the observed value measured during the annual winter mass balance survey. During the melt
season, which is assumed to start after this date, seasonal snow albedo decreases as a function of
cumulative positive degree days ($\sum PDD$) following Hirose and Marshall (2013),

$$\alpha_s(d) = \alpha_0 - k_\alpha \sum PDD\,(d). \tag{20}$$

A minimum value of 0.4 is set for old snow. We parameterize the effects of summer snow fall on
albedo and mass balance through a stochastic model of summer precipitation events (Marshall,
2014). Precipitation events are set to occur randomly, with 25 events occurring from May through
September as the default setting. Precipitation totals vary randomly, between 1 and 10 mm w.e.,
with snow at temperatures below 0°C, rainfall above 2°C, and rain/snow partitioning increasing
linearly over the range 0-2°C.  Following a summer snow event, surface albedo is reset to $\alpha_0$, and
its albedo begins to decay following Eq. (20). This treatment allows a natural transition to end-of-
summer conditions, when fresh snowfall in September or October does not melt away.
7. Subsurface temperatures and the conductive heat flux, $Q_C$, are modelled with 10-minute to one-
hour time steps (chosen for stability of the temperature solution). The updated surface temperature
$T_s$ is used for the calculation of outgoing longwave radiation (Eq. 6), sensible heat flux (Eq. 11),
and latent heat flux (via $q_s$ in Eq. 12) for the next time step.
8. The hydrology model calculates meltwater drainage and refreezing. Annual meltwater runoff is
then the sum of all meltwater that drains, while summer mass balance is equal to the meltwater
runoff minus the total summer snowfall, nominally for the period May 1 to September 30 at this
site. This allows for some meltwater retention as either liquid water or refrozen ice within the snow
or firn. We neglect water storage in the englacial and subglacial hydrology systems.

## 3.  Field Site and Observational Data

Reference meteorological conditions, surface energy balance fluxes, and snow conditions are
based on *in situ* measurements at Haig Glacier in the Canadian Rocky Mountains for the period
2002-2012 (Marshall, 2014). Winter mass balance measurements are carried out each May. These
observations provide an 11-year record of observed snow depth and summer melt from an
automatic weather station (AWS) located near the median elevation of the glacier, 2660 m (Figure
2). This is the upper ablation area of the glacier, which generally undergoes a transition from
seasonal snow to exposed glacier ice in August.

Table 1 summarizes the mean observed meteorological and conditions at Haig Glacier over the 11-year reference period. Data coverage is incomplete, particularly in the winter months, as we transitioned to summer only measurements (May-Sept) after 2009. For the 11 years, data coverage is as follows for most sensors (e.g., temperature, shortwave radiation): JJA - 90% (909 of 1012 days); MJJAS – 86% (1441 of 1683 days); annual – 63% (2519 of 4018 days). There are more missing longwave radiation data, as the sensor was not installed until July 2003. The corresponding numbers are: JJA – 76%; MJJAS – 70%; annual – 46%.

Missing data are gap-filled from a weather station that has operated continuously in the glacier forefield since 2001, at an elevation of 2325 m. The forefield AWS has more complete data coverage than the glacier AWS, above 90% for all variables. Observational data are used to adjust for the altitudinal and environmental differences between the sites, through either a monthly offset (e.g., $T_G = T_{FF} − \Delta T$), or a scaling factor β (e.g., $v_G = \beta_v v_{FF}$). Here, subscripts $G$ and $FF$ refer to the glacier and forefield AWS sites. The monthly factors are calculated from the set of all available overlapping data for the two stations. The temperature offset approach is equivalent to a lapse rate, or can be expressed that way for distributed modelling over the glacier. In this study we consider only the point energy balance at the glacier AWS site. If both stations are missing data, gap-filling is done through assignment of mean daily observational data.

To give a sense of the complete data record, Figure 3 shows examples of the full record, for air temperature, modelled surface temperature, and the energy fluxes. Average June to August (JJA) air and surface temperature are 5.2°C and −0.6°C, respectively, and 98% of JJA days reach surface temperatures of 0°C (melting conditions) in the 11-year record. The surface energy fluxes in Fig. 3b illustrate the dominance of net radiation in governing net energy at this site (Table 2).

Mean daily values for the 11-year record are plotted in Figure 4. As is typical for mid-latitude glaciers, net radiation is the main energy flux that drives glacier melt at this site (Fig. 4c). Net radiation is negative in the winter, when shortwave inputs are low, albedo is high, and longwave cooling gives a radiation deficit. Net radiation is positive in the summer and increases through the melt season. This is driven by increases in net shortwave radiation as snow albedo declines at the site and then melts away to expose the underlying glacier ice (Fig. 4a). Measurements at the AWS site indicate a seasonal snow albedo decrease from about 0.8 to about 0.4 each summer, which may be due to a combination of increased snow water content, grain metamorphosis in the temperate snowpack, and increasing concentration of impurities through the melt season (e.g., Cuffey and Paterson, 2010).

Median daily melt rates for the period 2002-2012 are plotted in Fig. 4d, along with the interquartile range. On average, 65% of the annual glacier melt occurs in the months of July and August. Net energy peaks in August, when the low-albedo glacier ice is exposed. Sensible heat flux peaks in July, and is the other main source of energy contributing to glacier melt. On average for JJA, net radiation and sensible heat flux constitute 70% and 30% of the net energy, respectively. Latent heat flux represents a small sink of energy, and conductive heat flux is a minor source of energy.

The energy balance and snowpack models have been developed and tested elsewhere (Marshall, 2014; Ebrahimi and Marshall, 2015), so we do not present the model validation in detail here.

Comparisons are favorable between AWS observations (*e.g.*, in situ albedo, SR50-inferred melt),
the model driven with 30-minute AWS data, and the 'daily' version of the model used here, which
includes parameterizations of albedo, incoming longwave radiation, and the diurnal temperature
and shortwave radiation cycles (Section 2). The simplified daily model loses some reality, but its
overall performance is excellent.
As an example, glacier AWS data from summer 2015 is used as an independent test of the model,
with its default parameterizations. Observed melt at the AWS site was $3.1 \pm 0.1$ m w.e. in summer
2015, while the melt model forced by 30-minute AWS data gives 3.04 m w.e. and the
parameterized, daily version of the model gives 2.98 m w.e. Taking the 30-minute AWS-driven
results as the reference, the RMS error in the daily melt predictions for the parameterized model
is 3% (0.7 mm w.e., relative to a daily mean value of 22.7 mm w.e.). Departures from the
observations are primarily associated with the albedo, which is over-estimated in summer 2015.
Overall the parameterized daily model has good skill and is an appropriate tool for the sensitivity
analyses presented here.

## 4. Theoretical Sensitivity of the Surface Energy Balance

Surface energy balance processes and summer melt rates depend on various meteorological
influences (Eqs. 4-11). Warm summers generally cause high melt rates and promote negative mass
balance, but the energy balance is sensitive to other weather conditions as well. To examine these
sensitivities, meteorological variables in Tables 1 and 2 can be perturbed one at a time or in
combination to examine the impact on summer melt at the Haig Glacier AWS site. Perturbations
are introduced with respect to the mean JJA meteorological conditions from 2002-2012.
Theoretical sensitivities are calculated in this section by differentiating the net energy balance with
respect to each meteorological variable. This is akin to generating a Jacobian matrix for $Q_N$, based
on partial derivatives of the dependent variables in the surface energy balance. One cannot gauge
the most important meteorological influence on surface energy and mass balance from the
sensitivities to a unit change in each variable. For instance, a change in specific humidity of 1 g
kg$^{-1}$ equals 3.3 standard deviations, with respect to the interannual (JJA) variability (Table 1). In
contrast, summer temperature has a standard deviation of 0.8°C, so a 1°C temperature change is a
smaller perturbation. To allow a direct comparison of the theoretical sensitivities and to give a
simple representation of their natural, interannual variability, we perturb each variable by one
standard deviation, based on the values reported in Tables 1 and 2.
We consider the core summer months, JJA, to calculate the theoretical sensitivity because the
glacier surface is at melting point for most of this time (Fig. 3a), which is a necessary condition to
relate net energy to melt. More than 80% of the annual melt also occurs in this season (Table 2
and Fig. 4d), so meteorological forcing over this period has the highest impact on glacier melt.
*Sensitivity to Temperature*
Air temperature appears directly in the expressions for $Q_L^{\downarrow}$ and $Q_H$. Temperature change may also
influence the surface energy balance through influences on other variables, such as atmospheric
moisture ($Q_E$). For a melting glacier surface, where surface and subsurface temperatures are at
0°C, air temperature changes do not directly influence $Q_L^\uparrow$ or $Q_C$. To estimate the magnitude of
temperature sensitivity, we differentiate each energy balance flux with respect to temperature.
For incoming longwave radiation, Eq. (7), the resulting temperature sensitivity is:

$$\frac{\partial Q_L^\downarrow}{\partial T} = 4\sigma\varepsilon_a T_a^3 + \sigma T_a^4 \frac{\partial \varepsilon_a}{\partial T}. \tag{21}$$

This general form applies to a range of formulations for $\varepsilon_a$, such as those of Brutsaert (1975),
Lhomme et al. (2007), or Sedlar and Hock (2009). Adopting the parameterization in Eq. (8), which
performs well at Haig Glacier,

$$\frac{\partial Q_L^\downarrow}{\partial T} = 4\sigma\varepsilon_a T_a^3 + \sigma T_a^4 \left( b\frac{\partial e_v}{\partial T} + c\frac{\partial h}{\partial T} \right). \tag{22}$$

The last two terms reflect potential feedbacks of temperature change on humidity. While we are
only considering perturbations to temperature in this section, vapour pressure and relative humidity
cannot both remain constant under a temperature change. We first assume that relative humidity $h$
remains constant, under which conditions we assume that cloud cover and sky clearness will be
unchanged. For constant $h$, $e_v$ scales with temperature following the Clausius-Clapeyron relation
for saturation vapour pressure,

$$\frac{\partial e_v}{\partial T} = \frac{h}{100}\frac{\partial e_s}{\partial T} = \frac{h}{100}\left( \frac{L_v e_s}{R_v T_a^2} \right) = \frac{L_v e_v}{R_v T_a^2}, \tag{23}$$

where $R_v = 461.5$ J kg$^{-1}$ °C$^{-1}$ is the gas law constant for water vapour.
For the mean JJA meteorological conditions at Haig Glacier, Eqs. (22) and (23) give $\partial Q_L^\downarrow/\partial T =$
4.7 W m$^{-2}$ °C$^{-1}$. Temperature increases affect $Q_L^\downarrow$ through both the direct effect of higher emission
temperatures and the indirect effect of higher atmospheric emissivity, with these two terms in Eq.
(21) contributing 4.0 and 0.7 W m$^{-2}$ °C$^{-1}$, respectively.
The temperature sensitivity of sensible and latent heat fluxes follow

$$\frac{\partial Q_H}{\partial T} = \frac{\rho_a c_p k^2 v}{\ln(^Z/_{z_0})\ln(^Z/_{z_{0H}})}, \tag{24}$$

and

$$\frac{\partial Q_E}{\partial T} = \frac{\rho_a L_p k^2 v}{\ln(^Z/_{z_0})\ln(^Z/_{z_{0E}})}\left( \frac{\partial q_v}{\partial T} \right), \tag{25}$$

where

$$\frac{\partial q_v}{\partial T} \approx \frac{R_d}{P R_v}\left( \frac{\partial e_v}{\partial T} \right), \tag{26}$$

for the dry gas-law constant $R_d = 289$ J kg$^{-1}$ °C$^{-1}$ and air pressure $P$, under the assumption that air
pressure and density are constant for small changes in temperature. Table 3 gives the turbulent

flux sensitivities for mean JJA conditions at Haig Glacier. Perturbations to both $Q_H$ and $Q_E$ are positive with an increase in temperature and the assumption of constant $h$. In combination with the increase in $Q_L^{\downarrow}$, net energy over the summer months is augmented by 12 W m$^{-2}$ for a 1°C increase in temperature. Interannual variations in summer temperature (1σ) equal 0.8°C, giving a net energy perturbation $\delta Q_{N\sigma} = +10$ W m$^{-2}$ (Table 3).

Fluctuations in energy balance can be related to melt rates through their combined influence on $Q_N$, with $\delta \dot{m} = \delta Q_N / \rho_w L_f$. Table 3 summarizes these impacts on summer melt, assuming a JJA melt season (92 days). The 1-σ temperature increase ($\delta Q_{N\sigma} = 10$ W m$^{-2}$) is equivalent to 236 mm of meltwater at the AWS site, a 10% increase over the reference JJA melt, 2320 mm w.e. These are the direct impacts of higher temperatures, not accounting for feedbacks or non-linearity in the seasonal evolution of melt conditions. These calculations assume that melting conditions prevail throughout the summer and all of this energy can be directed to snow/ice melt, which is not strictly true. We include them because estimates of the potential influence on summer melt provide an intuitive way to understand and compare sensitivities. We consider more realistic relations between net energy and melt in the modelled sensitivities of Section 5.

This initial scenario assumes that the warmer atmosphere contains more moisture, which is not necessarily the case. For instance, high summer temperatures in this region are commonly associated with ridging and subsidence, i.e. hot, dry conditions. If we assume that $q_v$ is invariant with temperature (case 2 in Table 3), there is no feedback on the latent heat flux and the increase in net energy is less than with constant $h$: $\delta Q_{N\sigma} = 6.6$ W m$^{-2}$ and $\delta m_\sigma = 157$ mm w.e.

However, there are additional feedbacks associated with relative humidity. If $q_v$ is invariant, relative humidity must change to be consistent with the temperature perturbation. As an example, an increase of 1°C with no change in $q_v$ corresponds to a decrease of 6% in mean summer $h$ at our site, to 61%. This lowers the atmospheric emissivity in Eq. (8), reduces the incoming longwave radiation, and impacts $\partial \varepsilon_a / \partial T$ in Eq. (22). To be internally consistent, reduced humidity anomalies should also be associated with changes in cloud cover. For the 1°C temperature increase, the 6% decrease in relative humidity corresponds to an increase in clearness index of 0.06 (Eq. 10), from 0.63 to 0.69.

The effects of these radiation feedbacks are given in Table 3. Reduced relative humidity decreases $Q_L^{\downarrow}$ and increases $Q_S^{\downarrow}$. The resulting increase in shortwave radiation partially offsets the decline in $Q_L^{\downarrow}$, but there is an overall reduction in net radiation. For our parameterizations of the incoming radiation fluxes as a function of humidity, the effect of drier air on longwave radiation is stronger than the shortwave radiation feedback. This reduces the overall sensitivity to temperature change relative to the first two cases, with $\delta Q_{N\sigma} = 5.3$ W m$^{-2}$ and $\delta m_\sigma = 125$ mm w.e. Note that these temperature scenarios are all idealized, neglecting albedo feedbacks and other indirect effects of a temperature change. These feedbacks are assessed in Section 5.

*Sensitivity to Humidity and Wind*

Similar derivatives and energy balance sensitivities can be derived with respect to the other meteorological variables, to explore the sensitivity of summer melt to different weather conditions. The sensitivity of sensible and latent heat fluxes to wind perturbations follow:


$$\frac{\partial Q_H}{\partial v} = \frac{\rho_a c_p k^2 (T_a - T_s)}{\ln(Z/z_0)\ln(Z/z_{0H})} \; , \tag{27}$$


and

$$\frac{\partial Q_E}{\partial v} = \frac{\rho_a L_p k^2 (q_v - q_s)}{\ln(Z/z_0)\ln(Z/z_{0E})} \; , \tag{28}$$


while the sensitivity to humidity is:

$$\frac{\partial Q_E}{\partial q_v} = \frac{\rho_a L_p k^2 v}{\ln(Z/z_0)\ln(Z/z_{0E})} \; . \tag{29}$$


Incoming longwave radiation is also affected by perturbations in humidity, following:

$$\frac{\partial Q_L^{\downarrow}}{\partial q_v} = \sigma T_a^4 \frac{\partial \varepsilon_a}{\partial q_v} = \sigma T_a^4 \left( b \frac{\partial e_v}{\partial q_v} + c \frac{\partial h}{\partial q_v} \right). \tag{30}$$


Table 3 summaries the theoretical sensitivities for specific humidity and wind perturbations of 1 g
kg$^{-1}$ and 1 m s$^{-1}$, respectively, assuming that temperature is unchanged. For the humidity, we
present two scenarios: the first with perturbations to only the specific and relative humidity, and
the second including the expected effects of an increase in relative humidity on cloud cover.

Changes in humidity directly impact the latent heat flux, and may also influence incoming
longwave radiation and cloud cover (hence, incoming shortwave radiation). We consider the
effects of a humidity perturbation with and without radiative feedbacks in Table 3. For $\delta q_v = 1$ g
kg$^{-1}$ and fixed temperature, mean summer relative humidity increases by 12%, to 79%, and $Q_E$
and $Q_N$ increase by 10.5 W m$^{-2}$. Interannual variations in $q_v$ equal 0.3 g kg$^{-1}$, giving $\delta Q_{N\sigma} = 3.2$ W
m$^{-2}$, corresponding to a 76-mm (3%) increase in summer melt.

Where radiation feedbacks are included, the increases in specific and relative humidity have a
strong influence on the atmospheric emissivity in Eq. (8), giving an increase in $Q_L^{\downarrow}$ of 24 W m$^{-2}$.
This is partially offset by cloud feedbacks associated with the increased humidity. Following Eq.
(10), $\delta h = 12\%$ equates to a decrease in atmospheric transmissivity of 0.11, which strongly
attenuates incoming shortwave radiation. This reduces the net radiation by 19 W m$^{-2}$, but the
radiation feedbacks remain positive. The net impact of a 1-$\sigma$ humidity perturbation $\delta q_v = 0.3$ g kg$^{-1}$
is then 4.7 W m$^{-2}$, corresponding to a 112-mm (5%) increase in summer melt.

Wind perturbations have straightforward linear effects on $Q_H$ and $Q_E$, giving a net sensitivity
$\partial Q_N / \partial v = +7$ W m$^{-2}$ (m s$^{-1}$)$^{-1}$. Sensible heat flux increases and evaporative cooling decreases
slightly. Winds have a low interannual variability at this site, 0.2 m s$^{-1}$, so the associated net energy
anomaly is $\delta Q_{N\sigma} = 2$ W m$^{-2}$, equivalent to 50 mm w.e. in summer melt.

*Sensitivity to the Radiation Fluxes*

Net shortwave radiation is affected by variations in top-of-atmosphere insolation, the clearness
index (i.e. cloud conditions), and surface albedo. Our functional relationship for net shortwave
radiation is $Q_{Snet} = Q_S^{\downarrow}(1-\alpha_S) = Q_{S\phi}\tau(1-\alpha_S)$, for potential direct insolation $Q_{S\phi}$ and clearness index
$\tau$. From Eq. (4), sensitivity to top-of-atmosphere insolation $Q_0$ follows

$$\frac{\partial Q_{Snet}}{\partial Q_0} = \tau\,(1-\alpha_S)\cos(Z)\,\varphi_0^{P/P_0\cos(Z)}\,, \tag{31}$$


An anomaly of 1 W m$^{-2}$ in the top-of-atmosphere insolation, $Q_0$, gives $\delta Q_S^{\downarrow} = 0.6$ W m$^{-2}$, and the
net radiation impact is further reduced to 0.3 W m$^{-2}$ by the surface albedo. The net impact of top-
of-atmosphere solar variability, such as sunspot cycles, is therefore small.

In contrast, incoming radiation fluxes and energy balance are strongly sensitive to atmospheric
transmissivity, which in turn is largely governed by cloud cover. Direct, independent variations in
incoming shortwave and longwave radiation are reported in Table 3 for fluctuations of 10 W m$^{-2}$
and for 1-$\sigma$ variations in each. Sensitivity is moderate, of order 6% of the net energy.

It is more appropriate to consider co-variations of these radiation fluxes that can be expected in
association with changes in cloud cover. We can estimate through the sky clearness index, $\tau$, as
parameterized via Eqs. (9) and (10), which relate the atmospheric emissivity and relative humidity
to clearness index. As an example, reduced cloud cover may be associated with a 1-$\sigma$ increase in
$\tau$ of 0.1, from 0.63 to 0.73. This translates to an increase in net shortwave energy of 16 W m$^{-2}$
(Table 3), but the change in cloud cover also impacts incoming longwave radiation. Clearer skies
in the example of Table 3 give lower $h$, lower $e_v$, and lower $Q_L^{\downarrow}$. Latent heat flux also declines.
The overall result is a reduction in net energy for an increase in $\tau$. A 1-$\sigma$ increase (+0.04) gives a
3% reduction in net energy.

*Sensitivity to Albedo*

The sensitivity to albedo changes is comparatively high. An change in albedo of 0.1 creates an
energy balance perturbation of more than 100 W m$^{-2}$ at local noon in mid-summer. The magnitude
of this effect varies with latitude, time of year, and atmospheric transmissivity. Integrated over the
daily solar path and over the summer, an albedo increase of 0.1 reduces net solar radiation by $-23$
W m$^{-2}$. Measurements at the site indicate an interannual albedo variability of 0.06, equivalent to
14% of the net energy or $\delta m_\sigma = -323$ mm w.e.

*Summary*

Overall, the results indicate a strong sensitivity of the summer energy balance and melt to
temperature and albedo, with weaker influences from cloud conditions, humidity, and wind speed.
These theoretical sensitivities are idealized, however, and neglect many important feedbacks and
glaciometeorological interactions that occur in glacier environments. The next two sections
examine the energy balance sensitivity at Haig Glacier within an energy balance-melt model. This
allows an estimate of feedbacks associated with the evolution of albedo, interannual variability in
weather conditions, and meteorologically-consistent covariance of weather variables.

## 5. Modelled Sensitivity of the Surface Energy Balance

We use a point model of surface energy balance, described in detail in Section 2. For all numerical experiments described below, we use the daily model with parameterizations of the longwave radiation fluxes, atmospheric clearness, diurnal cycles of temperature and shortwave radiation, and surface albedo evolution, following Eqs. (6), (8), (10), (17), (18), and (20). Surface temperature is modelled from the subsurface temperature model. The mean daily forcing for the energy balance and snowpack models is taken from the glacier AWS data, and the model is run year-round for the period 2002-2012. The May 1 snowpack thickness (winter accumulation) is specified for each year based on the measured winter mass balance at the AWS site.

Perturbations to the observed weather are used to repeat the sensitivity analyses of section 4, but with a realistic evolution of each summer melt season rather than the mean summer conditions. Meteorological variables are perturbed as follows: ±2°C for temperature, ±50% for specific humidity and wind, ±0.1 for the sky clearness index (a proxy for cloud cover), and ±0.1 for albedo. Increments are set to give 41 realizations in each case, spanning the range of the perturbation. For example, temperature increments of 0.1°C are applied for the range −2 to 2°C. Each perturbation is prescribed for all days in the original data, and the energy balance program is run for the period 2002-2012. In each experiment, all other meteorological variables are held constant except for those that are direct impacted by a perturbation (e.g., relative humidity changes with temperature).

Table 4 lists the response of mean summer (JJA) net energy, $Q_N$, to the different meteorological perturbations. Changes in the energy fluxes can be examined in response to the perturbations, e.g., $\Delta Q_N$ as a function of temperature anomalies, $\delta T$. We plot these values to give sensitivity curves (e.g., Figures 5 and 6), and the slope of each curve is a measure of the sensitivity, e.g., $dQ_N/dT$. Values in Table 4 are calculated through linear regression. The relationships area generally nonlinear, so we compute the regressions for the region of the sensitivity curve within ±1 standard deviation (±1 σ) of the reference value for each variable. This samples a more linear range and allows a better comparison with the derivatives in Table 3. Standard deviations refer to the interannual variability, as reported in Table 1. Table 4 also lists the change in net energy associated with a 1-σ increase in each variable.

There are multiple scenarios for temperature, shown in the first four cases in Table 4. These cases represent different assumptions about the way in which atmospheric moisture and radiation fluxes respond to a temperature perturbation. The first two cases follow the assumption that relative humidity does not change. Hence, a temperature change $\delta T$ is attended by a change in specific humidity, $\delta q_v$, to maintain constant $h$. This impacts latent heat flux and atmospheric emissivity. Cases 1 and 2 show the net energy sensitivity to this scenario without and with albedo feedbacks. The next two cases include albedo feedbacks, but assume no change in specific humidity, $\delta q_v = 0$; hence relative humidity must respond. Cases 3 and 4 are without and with atmospheric radiation feedbacks to the changed relative humidity.

Summer melt sensitivity for the four different temperature perturbation scenarios is plotted in Figure 5. Case 1 lacks albedo feedbacks and corresponds to a net energy sensitivity of 13 W m$^{-2}$ C$^{-1}$, which is comparable to the theoretical temperature sensitivities in Table 3. This is due to direct temperature/humidity impacts on incoming radiation fluxes, sensible heat flux, and latent heat flux.

Cases 2-4 include albedo feedbacks. This can be considered to be more realistic, and the albedo
feedbacks have a roughly two-fold amplification effect on the temperature perturbation. Under
constant $h$, $dQ_N/dT = 27$ W m$^{-2}$ C$^{-1}$ (cf. Figure 6a), representing a 28% increase in summer melt
for a 1°C warming. This decreases by 6-10 W m$^{-2}$ C$^{-1}$ in cases 3 and 4, where $q_v$ is held constant.
Some of the reduced energy comes from the elimination of latent energy feedbacks. Case 4, with
atmospheric radiation feedbacks, reduces energy further as decreased cloud cover (via higher $\tau$)
reduces incoming longwave radiation more strongly than it increases shortwave fluxes in the
model. Here too, the numerical model gives a similar result to the theoretical prediction.

Figure 6a plots the response of the different surface energy fluxes for the reference model, case 2.
Net shortwave radiation dominates the temperature response, over $Q_H$, $Q_E$, and $Q_L^{\downarrow}$. Figures 6b-
6d provide similar details for perturbations in humidity, wind, clearness index, and albedo (cases
5-9 in Table 4). Sensitivity to humidity changes is relatively strong, through the combined impacts
of latent and longwave fluxes (Fig. 6b). Case 6 is shown in this figure, including feedbacks on the
atmospheric radiation. Incoming longwave radiation is strongly augmented by the increases in
absolute and relative humidity, and accounts for about 70% of the net energy sensitivity to specific
humidity. It is partially offset by cloud feedbacks, however, which reduce incoming shortwave
radiation.

For increases in both temperature and humidity, the mean summer latent heat flux switches sign
from negative (Table 2) to positive; that is, latent heat flux becomes a source rather than sink of
energy under warmer and wetter conditions. In contrast, latent heat flux remains negative, but
small, under increases in wind speed (Figure 6c). Energy balance sensitivity to wind perturbations
is primarily associated with the sensible heat flux.

Net energy perturbations due to albedo and clearness index in Figure 6d are independent of each
other, but are plotted together for convenience. Albedo sensitivity over the range ±0.1 is relatively
high, with a decrease in net energy of 27 W m$^{-2}$ (28%) for an increase in albedo of 0.1. Changes
in sky clearness index (atmospheric transmissivity) have a lower impact, due to the compensating
influences on incoming shortwave and longwave radiation. Reduced cloud cover (higher $\tau$) gives
an overall reduction in net energy at our site, as longwave radiation effects are dominant.

*Sensitivity to Winter Snow Accumulation*

Changes in the winter mass balance also influence the summer melt season. Interannual variability
in the amount of snow is implicit in the simulations, as the spring (May 1) snowpack depth is
initialized with the measured winter mass balance for each year, $b_w$ (Marshall, 2014). However,
these experiments do not control for the influence of snow depth on summer melt extent.

To examine this, we force the energy balance model over a range of winter mass balance
conditions, $b_w \in [0.36, 2.36]$ m w.e. This is ±1 m w.e. relative to the mean observed value at the
AWS site, 1.36 ± 0.27 m w.e. The melt model is run through 11 years of weather, 2002-2012, with
the different values of winter mass balance as an initial condition. Figure 7 plots the average
evolution of seasonal snowpack depth and albedo from May through September for this suite of
experiments. Transitions from seasonal snow to ice span from early July to mid-September.

Albedo spikes in Fig. 7b are due to summer snow events, which become more frequent as temperatures cool in September.

The net energy balance perturbations that accompany these scenarios are shown for two choices of the minimum snow albedo (Fig. 7c). Observations of late-summer snow at the site are in the range 0.3-0.4, the two values presented here. The plot is asymmetric; net energy is more sensitive to reduced winter snow depths, which result in an earlier transition to exposed glacier ice. A 20% ($1\sigma$) reduction in $b_w$ gives a net energy increase of about $4\,\mathrm{W\,m^{-2}}$ (4%), and the sensitivity increases non-linearly with increasingly lower snow depths. The influence from a deep winter snowpack is comparatively muted: $1\text{-}2\,\mathrm{W\,m^{-2}}$ reductions in $Q_N$ for a 20% increase in the winter snow thickness. Perturbations in $Q_N$ asymptote once seasonal snow is deep enough to survive through the summer.

The influence of the winter snowpack at this site is similar in magnitude to the net energy impacts of interannual variations in wind speed, but less important to the summer melt than observed variations in temperature, albedo, or cloud cover. This result is partly due to the relatively low contrast between late-summer snow albedo and bare-ice albedo at this site. If late-summer snow has a higher albedo, a deep winter snowpack is more effective at reducing the net energy and summer melt. The shape of the sensitivity curve would change for locations with higher-albedo snow, and also for sites in the lower ablation zone, where ice is exposed early in the melt season. A heavy winter snowpack would have a comparatively stronger role in this case. The result in Figure 7 is therefore more site-specific than for the other meteorological perturbations.

## 6. NARR-based Surface Energy Balance Reconstructions, 1979-2014

To examine energy balance sensitivity over a longer time period and with joint variation in meteorological variables, we run the energy balance model forced by North American Regional Reanalysis (NARR) atmospheric reconstructions from 1979 to 2014 (Mesinger et al., 2006). This provides a more complete picture of interannual variability, while comparison of NARR predictions with measurements over the period 2002-2012 also allows us to assess the skill with which fluctuations in surface energy balance and summer melt can be captured in an atmospheric model that does not explicitly resolve the alpine and glacier conditions.

We use a perturbation approach as in Section 5, taking NARR daily meteorological fields as anomalies relative to the mean NARR conditions for the period 2002-2012. Anomalies in near-surface temperature, specific humidity, wind speed, pressure, incoming shortwave radiation and incoming longwave radiation are used to drive the model for the 36-year period 1979-2014. Perturbations are introduced as anomalies relative to the mean observed conditions. NARR input fields allow us to introduce multiple perturbations at once, with magnitudes that are physically meaningful and meteorologically-consistent covariance of variables.

NARR has an effective spatial resolution of 32 km, and we extract mean daily data from the grid cell over Haig Glacier. This grid cell has an elevation of 2214 m, about 450 m lower than the AWS site. By using daily weather anomalies, we avoid most biases associated with the different altitude of the NARR grid cell. However, variations in some fields such as specific humidity, pressure, and temperature can be larger at lower elevations and over non-glacierized land surface types. Since we use meteorological fluctuations as perturbations, this is potentially problematic. Inspection of the summer variance in the different meteorological inputs over the reference period 2002-2012

indicates that this does not appear to be an issue. Standard deviations of each variable, calculated
from mean JJA values, are as follows: temperature, 0.8°C; specific humidity, 0.2 g kg$^{-1}$; wind
speed, 0.3 m s$^{-1}$; incoming shortwave radiation, 6 W m$^{-2}$; and incoming longwave radiation, 3 W
m$^{-2}$. Temperature, humidity, and wind values are equivalent to the observed range of variability
from 2002-2012 (Table 1), but the radiation fluxes are less variable. The effects of a lower
elevation in the NARR grid cell appear to be less than those associated with systematic biases in
the reanalysis, e.g., not enough variability in cloud conditions.
The energy balance model requires an estimate of winter snow accumulation. We base this on
cumulative NARR precipitation for the period September to May of each year, normalized to the
observed value of 1.36 m w.e. at the Haig Glacier AWS site. This permits interannual variability
in the winter snowpack thickness to be included in the simulations, by scaling the mean observed
value up or down based on the NARR winter precipitation totals. We use this as an initial condition
for the melt model (i.e., for May 1 snow depth).
We examine the sensitivity of net summer energy balance and melt to interannual variations in
each weather variable in the NARR forcing. Table 5 reports the NARR-based surface energy fluxes
and melt for JJA and MJJAS, averaged over the period 1979-2014. Mean values are all within 2
W m$^{-2}$ of the reference surface energy fluxes (Table 2), derived from the in situ data, but there are
some significant differences in the standard deviation, which is a measure of the interannual
variability. As noted above, incoming shortwave radiation has about half of the variability in the
36-year NARR record as observed in the 11-year measurement period, and variance in incoming
longwave radiation is also less than observed. This implies more uniform summer cloud conditions
in the reanalysis, compared to the observational period.
Average summer albedo is also less variable in the model than the observations, and the mean
value in the NARR-forced model is too low for May through September (0.55 vs. an observed
value of 0.60). Most of this difference is associated with a low value of September albedo in the
model; we are generally underestimating September snow events and predicting too late a
transition from end-of-summer to the winter accumulation season. This transition occurs sometime
in September or October each year in our study period. September is mixed on the glacier, with
fresh snowfall alternating with periods of melting. This raises the average albedo on the glacier,
but our albedo parameterization does not fully capture this.
Figure 8a plots time series of the NARR-forced surface energy balance terms, and Figures 8b-8d
shows the relations between net energy and selected meteorological variables. These provide a
visual indication of the strength of each variable as a predictor of summer melt. Regressions
through these data points give estimates of net energy sensitivity, e.g. $\partial Q_N/\partial T$, as seen in actual
realizations of the summer weather conditions. These gradients can be thought of as the melt
sensitivity to interannual variability or trends in each weather variable.
The resulting sensitivities are given in Table 6, as well as linear correlation coefficients between
$Q_N$ and all glaciometeorological variables that are used in the energy balance model. These
simulations are forced with NARR radiation flux anomalies, so we do not parameterize the
incoming longwave or shortwave radiation in these tests. The clearness index, $\tau$, is not used, but it
can be calculated from the NARR relative humidity estimate, via Eq. (10), or more directly through
the fraction of incoming shortwave radiation relative to the clear-sky potential radiation. We test
both approaches and find similar results. Values for $\partial Q_N/\partial \tau$ reported in Table 6 are averaged from
the two approaches. We also report the direct relation between NARR total cloud cover and net
energy; cloud cover is available in the reanalysis, but we do not have *in situ* data to compare with.
Temperature and albedo have the strongest influences on summer energy balance and melt.
Fluctuations in specific humidity and incoming longwave radiation also correlate strongly with
interannual variability in the summer energy budget. Wind speed, cloud conditions, and incoming
shortwave radiation do not strongly contribute to the year-to-year variations in summer melt over
the NARR period. There is a weak, positive relationship between the clearness index and net
radiation in the NARR-forced results, indicating that increased shortwave radiation associated with
reduced cloud cover has a stronger role than the associated reduction in longwave radiation.
These sensitivities can be compared with those in Section 5 (Table 4), but they differ in that the
NARR forcing has multiple joint perturbations. This is realistic as the meteorological variables co-
vary systematically, but it means that it is not possible to isolate the role of a single variable, such
as temperature. A temperature change impacts several of the energy fluxes, but coincident changes
in, e.g., humidity and radiation fluxes, may reinforce or reduce the temperature impacts. Results
in Table 6 should therefore be interpreted as the 'net' or 'effective' influence of each weather
variable on the summer energy balance, and some of them may have correlations that are more
coincidental than casual. Most results are nonetheless similar in magnitude to the theoretical and
modelling results (Tables 3 and 4), which are based on the *in situ* data. The largest exception is the
relation between clearness index (cloud cover) and net energy, which is opposite in sign.

## 7. Discussion

We have taken three different approaches to estimate summer (JJA) energy balance and melt
sensitivity at Haig Glacier: (i) theoretical, perturbing one variable at a time, (ii) a numerical model,
restricting model experiments to single perturbations but allowing for internal feedbacks to be
modelled, and (iii) through perturbations from a regional climate reanalysis, allowing multiple
variables to change at once. Here we briefly summarize and interpret the integrated results from
these different methods.
*Haig Glacier Energy Balance Sensitivities and Feedbacks*
Interannual variations in temperature and albedo have the strongest influence on summer energy
balance in all three approaches to assessing Haig Glacier melt sensitivity (Figure 9). Fluctuations
in humidity and longwave radiation are also important, while variations in cloud cover ($\tau$), wind
speed, and the winter snowpack thickness are less influential on the summer energy budget and
melt extent at this site.
Temperature changes are generally thought of as the main driver of glacier advance and retreat,
through combined influences on the surface energy budget, snow accumulation, and summer melt
season. Sensitivities to temperature are commonly expressed as the change in summer or net mass
balance per unit warming. Sample mass balance sensitivities reported in the literature are $-0.6$ m
w.e. $°C^{-1}$ on Morteratschgletscher, Switzerland (Klok and Oerlemans, 2004) and Illecillewaet
Glacier, British Columbia (Hirose and Marshall, 2013), $-0.68 \pm 0.05$ m w.e. $°C^{-1}$ for a suite of
glaciers in Switzerland (Huss and Fischer, 2016), and $-0.86$ m w.e. $°C^{-1}$ on South Cascade Glacier,
Washington (Anslow et al., 2008). Values as high as $-2.0$ m w.e. $°C^{-1}$ are reported for Brewster
Glacier, New Zealand (Anderson et al., 2010).

These values are for the annual mass balance, but they are dominated by the summer melt response
to warming. They represent a melt sensitivity of about 30% $°C^{-1}$ for the examples in the Alps and
western North America. When we introduce temperature perturbations in the absence of albedo
feedbacks, we find a relatively muted energy balance response, about 13 % $°C^{-1}$. The increase in
net energy is distributed about equally across the sensible heat flux, incoming longwave radiation,
and latent heat flux, and we have similar results for both the theoretical and numerically-modelled
temperature perturbations. Albedo feedbacks increase the net energy sensitivity to 28 % $°C^{-1}$ or
$-0.66$ m w.e. $°C^{-1}$, in accord with previous studies. The exact number depends on assumptions
about humidity; if specific humidity increases with temperature (e.g., by holding relative humidity
constant), temperature sensitivity is higher.

The albedo feedback results from two main ways that temperature influences the seasonal albedo
evolution. A more intense melt season gives rise to a lower snow albedo and an earlier transition
from seasonal snow cover to glacial ice. We do not explicitly model impurities or snow-albedo
processes (e.g., grain metamorphism, effects of snow-water content on the albedo), but we
parameterize the seasonal albedo evolution as a function of cumulative *PDD* (Eq. 20), which
makes the model directly sensitive to temperature perturbations.

Temperature changes have several additional, indirect impacts, including: (i) a longer melt season,
(ii) a greater fraction of time with surface temperatures at the melting point during the year, i.e.,
with reduced overnight cooling and refreezing, and (iii) an increase in the frequency of summer
rain vs. snow events. Summer snow events have an important impact on surface albedo, with fresh
snow strongly attenuating melt. Each of these processes contributes to the strong impact of
temperature anomalies on glacier melt. Combined with the albedo feedbacks, these processes and
help to explain why glaciers are strongly sensitive to temperature change.

Direct changes to albedo have an influence on summer energy balance and melt extent that is
comparable to the temperature influence, ~17% for a change in albedo equal to the interannual
albedo fluctuations, 0.06. Mean summer albedo differences arise as a feedback to other
meteorological forcings that drive the summer snow melt, but interannual albedo variations also
occur more directly, as a consequence of summer snowfall events, as a function of winter
accumulation totals, or due to impurity loading (e.g., black carbon deposition). The latter has been
observed in association with forest fires in British Columbia. Strong fire seasons occurred twice
during our period of study, in 2003 and 2015, and each left a measurably darker glacier surface.
For instance, the average albedo recorded at the AWS site in August 2003 was 0.13.

We found a relatively weak influence of winter mass balance on the summer melt extent. A low
snowpack depth has a greater impact, through an earlier transition to low-albedo bare ice. A deep
winter snowpack has the opposite influence, supporting a higher average summer albedo, but the
influence is weaker because the AWS site is in the upper ablation area, where the seasonal
snowpack persists until late summer in most years. The effects of greater winter accumulation
plateau once there is enough snow to survive the summer; beyond this point, additional snow has
no effect on the summer albedo or melt extent. Sensitivity to winter mass balance would likely be
stronger at lower altitudes on the glacier, and for the overall glacier mass balance.
Humidity changes can also be considered a feedback to temperature, but this is not certain; specific
humidity varies as a function of local- to synoptic-scale moisture sources and weather patterns,
and these are not necessarily coupled to temperature conditions. For instance, warm conditions at
Haig Glacier often accompany anticyclonic ridging in the summer months, during which time
southerly flows and upper-level subsidence promote dry, clear-sky conditions (low $q_v$ and $h$). At
other times, westerly flows bring warm, moist Pacific air masses and humidity, temperature, and
cloud cover co-vary. Interannual variability in specific humidity has a significant impact on
summer energy and melt extent, an ~8% change for a perturbation of $0.3\,\mathrm{g\,kg^{-1}}$ (1σ). This effects
net energy through impacts on the latent heat flux and incoming longwave radiation. The latter is
partially compensated by accompanying changes in incoming shortwave radiation.
With all three methods, cloud cover shows up as a relatively weak influence on summer net energy
at this site, ~4% for a 1-σ variation in the clearness index (Figure 9). This result is a consequence
of the offsetting effects of cloud cover on the shortwave and longwave fluxes. The sign of the
relationship is also uncertain. In isolation, interannual fluctuations in shortwave and longwave
radiation have a moderate influence on the summer net energy (Figure 9), so these are important;
they are just not simply related to the cloud cover index, τ.
*NARR Results*
NARR results are broadly consistent with the *in situ*-based and theoretical sensitivities, in terms
of the relative importance of different meteorological parameters to interannual variability in
summer energy balance and melt. The influence of interannual temperature fluctuations appear to
be weaker than the other sensitivity experiments would suggest, ~15% °C$^{-1}$. All feedbacks
discussed above are active in the NARR-based simulations. The impacts of temperature variability
on net energy and melt could be partially compensated by other systematic changes in the energy
budget. For instance, warm temperatures are often associated with calm, clear-sky conditions that
reduce the incoming longwave radiation and the turbulent fluxes.
Temperature nonetheless emerges as the most important variable explaining interannual variations
in net energy. Mean summer net energy and temperature are highly correlated ($r = 0.84$). This
reinforces the argument that temperature indices offer a good proxy for net energy and summer
melt extent (e.g., Ohmura, 1987).
There are two other discrepancies in the NARR-forced results. Year-to-year variance in incoming
radiation fluxes is less than observed, pointing to poor representation of interannual cloud
variability in the reanalysis. The variability is still positively correlated with the in situ data (e.g.,
$r = 0.50$ for the correlation between incoming JJA shortwave radiation in NARR and in the data
from 2002-2012). Hence, NARR is picking up some of the observed variability, but it is muted.
The sensitivities to the radiation fluxes may still be representative, as there is still some interannual
variability for which on can assess the relation between $Q_N$ and the radiation fluxes. However, the
poor representation of the radiation fluxes and cloud conditions can be expected to reduce the skill
of NARR-forced mass and energy balance reconstructions; this requires further study.
The other main difference with the NARR forcing is a switch in sign in the sensitivity to changes
in cloud cover, as analyzed through either $\tau$ or the NARR-predicted total cloud cover. Clear-sky
conditions have a positive relation with $Q_N$ in the NARR-driven simulations, signalling that
incoming shortwave radiation fluxes exert more influence than incoming longwave fluxes for net
summer energy. Clear-sky conditions (less cloud cover) give increased shortwave radiation and a
lesser decrease in longwave radiation, resulting in increased net energy. The theoretical and *in situ*
sensitivities predict the opposite result, reduced net energy with clearer skies. The relationship is
relatively weak, so it is possible that there are confounding variables in the NARR simulations
once again, such as temperature effects masking the cloud relationship.
We do not test the ability and skill of NARR-forced energy and mass balance reconstructions here.
This requires further study. In general, the perturbation method eliminates biases in the mean
NARR variables, but a realistic representation of the variability and long-term trends in reanalysis
fields is important to realistic representations of the glacier mas balance record and meltwater
runoff. It would be instructive to analyze the synoptic weather patterns and weather anomalies in
high-melt vs. low-melt summers in the NARR-driven simulations. We recommend an
investigation of specific weather systems and their associated meteorological and energy balance
conditions in followup work.
*Representativeness of the Results*
We have designed the sensitivity approach and the model to be applicable in regional studies, e.g.
in a distributed model of glacier energy balance, forced by climate model reanalyses or projections.
However, we did not expand our scope to other sites within the present study. In principle, the
theoretical sensitivities (i.e. from the same set of equations) could be calculated for different
baseline meteorological conditions, such as maritime or tropical environments. The method, rather
than the specific Haig Glacier results, could be exported to other glacierized environments.
At regional scales, Haig Glacier energy balance sensitivities might be more transferrable, since
similar summer climate conditions prevail across the Canadian Rocky Mountains (Ebrahimi and
Marshall, 2015). Regional, multi-year reconstructions of glacier meltwater runoff might be
feasible through a perturbation approach to summer mass balance, driven by meteorological
anomalies from station data or climate models. This needs to be tested, however, for sensitive
parameterizations such as the albedo model. It is uncertain whether the Haig glacier bare-ice and
old-snow albedo are regionally representative.
Within Haig Glacier itself, our AWS site is in the upper ablation area, near the equilibrium ELA.
Results are specific to the snow and ice albedo, snowpack depth, and meteorological/energy
balance conditions at this location. We have not examined the representativeness of the results to
other parts of the glacier, but summer melt extent and mass balance at the AWS site are strongly
correlated with glacier-wide mass balance. We recommend additional work to calculate an average
set of glacier sensitivities and assess whether the values presented here are representative. We

suspect that sensitivity of net energy to winter snow depth and the strength of albedo feedbacks will vary across the glacier.

*Recommended Model Improvements*

Model improvements are recommended with respect to our treatment of the glacier surface albedo and precipitation modelling. The energy balance, albedo, and melt models perform well in the core summer melt season, June through August, when summer snowfall is infrequent and impacts on the albedo are transient. We systematically underestimate September albedo, however; better treatments of late-summer snow accumulation and the transition to the winter accumulation season are needed.

Our meltwater drainage model is also simplistic. We assume that water drains efficiently from the glacier surface, but in fact water has been observed to pond and refreeze on the surface. Re-melting of this superimposed ice consumes energy and reduces the total summer runoff.

A more realistic treatment of year-round snow accumulation is also needed in order to carry out model-based glacier mass balance reconstructions. We rely on observed winter mass balance for the studies here, but historical reconstructions and future projections require a way to reliably estimate snow accumulation from climate models. NARR precipitation in the Haig Glacier grid cell poorly represents the observed winter accumulation totals.

We have done tests to verify that the daily, parameterized model performs well relative to direct forcing with 30-minute AWS data, but some simplifications embedded in the daily model need to be examined. For instance, we assume constant cloud cover/clearness index over the day; systematic diurnal variations in cloud cover would affect the net radiation in ways that we do not capture. Overnight clouds serve to increase energy flux to the glacier, while daytime clouds reduce the incoming radiation. Effects like these become complicated to model or parameterize, but could bias our sensitivity results to cloud cover.

## 8. Conclusions

Sensitivity studies presented here extend the foundational work of Oerlemans and Fortuin (1992) and others, which has generally been done on glacier mass balance sensitivity to changes in temperature and precipitation. Our study is limited to summer mass balance at one location, but our results offer insight into the influence of different meteorological variables and energy fluxes, their year-to-year variability, and the role of isolated vs. collective forcings, feedbacks, and interactions on summer melt extent.

There is a good correspondence between the theoretical sensitivities and those derived from the numerical energy balance model, when feedbacks are omitted. This supports the potential application of the theoretical sensitivities to explore energy balance sensitivities under different climate regimes. This method can be transferred directly to other sites.

Temperature and albedo variations exert the strongest controls on year-to-year variability in summer melt at our site. While albedo can fluctuate independent of temperature, e.g., through the

influence of the winter snowpack depth or aerosol loading, it is also a powerful feedback mechanism to temperature and melt season evolution. In our model, albedo feedbacks give a two-fold increase in the net energy balance sensitivity to a temperature perturbation, amplifying the summer melt response from 13% $°C^{-1}$ to ~28% $°C^{-1}$. Temperature and albedo fluctuations are also the strongest influences on interannual melt variations in the NARR-forced surface energy balance, but the melt sensitivity to temperature variations is about 15% $°C^{-1}$, weaker than our result from the control experiments. This may be because the co-variation of other variables in the surface energy balance partially offsets the temperature forcing.

Humidity fluctuations are also effective in influencing the net energy, through their impacts on latent heat flux and incoming radiation fluxes. Wind speed, cloud conditions, and the winter snowpack thickness are less important to the summer energy balance and melt extent at our site. The relationship with cloud conditions is statistically weak and we do not have confidence in the sign; we recommend further work to assess the influence of cloud cover on summer net radiation at this site and elsewhere.

Our results suggest that it is may be reasonable to model glacier melt sensitivity at this site to temperature forcing, while ignoring variability and change in other weather conditions such as wind speed and cloud cover. This is the implicit premise in temperature-index melt models, and they can be tuned to work well at our site. We hesitate to recommend this though. Albedo feedbacks are crucial to include in assessments of glacier response to temperature change, and are not physically represented in most temperature-index models. Variations in humidity and their influence on melt are not negligible, and all terms in the surface energy budget contribute to the daily and interannual fluctuations in net energy.

Our modelling approach for surface energy balance is well-suited to a distributed energy balance model, applying the perturbation approach to larger scales (e.g., mountain ranges). Climate models simulate all of the relevant meteorological fields, and both past reanalyses and future projections can be driven using the perturbation approach introduced here. Meteorological sensitivities under different climate regimes (e.g., maritime, polar, or tropical conditions) can also be explored using this framework, to help understand regional differences in glacier sensitivity to climate variability and change.

**Acknowledgements**

This contribution benefitted from detailed reviews and insights of two anonymous reviewers and the Editor. It is much-improved from the reviewers' suggestions. We thank the Natural Sciences and Engineering Research Council (NSERC) of Canada for long-term support of the Haig Glacier study. S. Ebrahimi is financially supported through NSERC and the Alberta Water Research Institute project Predicting Alberta's Water Future. We are indebted to numerous graduate students and research assistants who helped to collect data and maintain instrumentation at Haig Glacier since 2001.

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

**Tables**

Table 1. Mean monthly weather conditions ± one standard deviation at Haig Glacier, Canadian Rocky Mountains, May to September 2002-2012. Data are from automatic weather station measurements at an elevation of 2660 m, in the upper ablation zone of the glacier.

| Month | $T$ (°C) | $h$ (%) | $e_v$ (hPa) | $q_v$ (g/kg) | $P$ (hPa) | $v$ (m/s) |
|---|---|---|---|---|---|---|
| May | −1.4 ± 1.1 | 73 ± 4 | 4.0 ± 0.4 | 3.4 ± 0.4 | 743.0 ± 2.4 | 2.8 ± 0.2 |
| June | 2.6 ± 0.9 | 73 ± 6 | 5.5 ± 0.5 | 4.6 ± 0.4 | 748.1 ± 1.4 | 2.6 ± 0.2 |
| July | 6.9 ± 1.4 | 62 ± 5 | 6.4 ± 0.4 | 5.3 ± 0.3 | 751.2 ± 1.6 | 2.8 ± 0.3 |
| August | 5.9 ± 1.1 | 64 ± 7 | 6.1 ± 0.4 | 5.1 ± 0.4 | 750.8 ± 1.4 | 2.5 ± 0.2 |
| Sept | 2.1 ± 1.8 | 71 ± 10 | 5.0 ± 0.4 | 4.2 ± 0.3 | 748.4 ± 1.8 | 3.0 ± 0.4 |
| JJA | 5.1 ± 0.8 | 67 ± 4 | 5.7 ± 0.4 | 4.8 ± 0.3 | 750.0 ± 1.1 | 2.6 ± 0.2 |
| MJJAS | 3.2 ± 0.7 | 69 ± 4 | 5.3 ± 0.3 | 4.3 ± 0.3 | 748.3 ± 1.4 | 2.7 ± 0.2 |

Table 2. Mean monthly surface energy balance terms ± one standard deviation at Haig Glacier, Canadian Rocky Mountains, May to September 2002-2012. Radiation fluxes and albedo values are from automatic weather station measurements and the turbulent fluxes and subsurface heat conduction are modelled from the AWS data. Fluxes are in W m$^{-2}$ and melt totals are in m w.e.

| Month | $Q_S^{\downarrow}$ | $\alpha_s$ | $Q_L^{\downarrow}$ | $Q_L^{\uparrow}$ | $Q_H$ | $Q_E$ | $Q_C$ | $Q_N$ | $melt$ |
|---|---|---|---|---|---|---|---|---|---|
| May | 249 ± 24 | 0.76 ± 0.04 | 258±12 | 299±4 | 7±4 | −11±3 | 5±2 | 22±12 | 0.20±0.10 |
| June | 237 ± 23 | 0.70 ± 0.05 | 276±14 | 310±2 | 17±4 | −5±4 | 3±1 | 56±21 | 0.45±0.16 |
| July | 240 ± 19 | 0.57 ± 0.06 | 275±8 | 313±1 | 38±9 | 1±5 | 1±1 | 109 ±27 | 0.88±0.21 |
| August | 205 ± 25 | 0.38 ± 0.07 | 273±11 | 312±1 | 32±7 | −1±3 | 2±1 | 123 ±22 | 0.99±0.18 |
| Sept | 140 ± 30 | 0.59 ± 0.09 | 271±13 | 306±3 | 23±12 | −6±3 | 3±2 | 42 ±21 | 0.34±0.16 |
| JJA | 227 ± 14 | 0.55 ± 0.06 | 275±6 | 312±1 | 29±3 | −2±3 | 2±1 | 97 ±19 | 2.32±0.45 |
| MJJAS | 215 ± 17 | 0.60 ± 0.04 | 271±7 | 308±1 | 23±4 | −4±3 | 3±1 | 71 ±15 | 2.86±0.59 |

1243

**Table 3.** Surface energy balance sensitivity to meteorological perturbations over a melting glacier surface, from direct feedbacks only. Calculations are for mean JJA conditions at Haig Glacier. All energy flux perturbations are expressed in $W\,m^{-2}$. $\delta Q_{N\sigma}$ is the net energy perturbation for a 1-$\sigma$ increase in the variable. The melt perturbation, $\delta m_\sigma$, has units of mm w.e., and is calculated assuming that $\delta Q_{N\sigma}$ holds for JJA (92 days).

| Perturbation | $\delta Q_S^\downarrow$ | $\delta\alpha$ | $\delta Q_S^{net}$ | $\delta Q_L^\downarrow$ | $\delta Q_H$ | $\delta Q_E$ | $\delta Q_N$ | $\delta Q_{N\sigma}$ | $\delta m_\sigma$ |
|---|---|---|---|---|---|---|---|---|---|
| $\delta T = 1°C;\ \delta h = 0$ | 0 | 0 | 0 | 4.7 | 4.2 | 3.5 | 12.4 | 9.9 | 236 |
| $\delta T = 1°C;\ \delta q_v = \delta\tau = \delta\varepsilon_a = 0$ | 0 | 0 | 0 | 4.0 | 4.2 | 0 | 8.3 | 6.6 | 157 |
| $\delta T = 1°C;\ \delta q_v = 0;\ \delta\tau,\ \delta\varepsilon_a$ | 22.6 | 0 | 10.2 | −7.8 | 4.2 | 0 | 6.6 | 5.3 | 125 |
| $\delta q_v = 1\ g\,kg^{-1};\ \delta\tau = \delta\varepsilon_a = 0$ | 0 | 0 | 0 | 0 | 0 | 10.5 | 10.5 | 3.2 | 76 |
| $\delta q_v = 1\ g\,kg^{-1};\ \delta\tau,\ \delta\varepsilon_a$ | −41.8 | 0 | −18.8 | 24.1 | 0 | 10.5 | 15.7 | 4.7 | 112 |
| $\delta v = 1\ m\,s^{-1}$ | 0 | 0 | 0 | 0 | 8.3 | −1.4 | 6.9 | 2.1 | 50 |
| $\delta Q_0 = 1\ W\,m^{-2}$ | 0.6 | 0 | 0.3 | 0 | 0 | 0 | 0.3 | – | – |
| $\delta Q_S^\downarrow = 10\ W\,m^{-2}$ | 10.0 | 0 | 4.5 | 0 | 0 | 0 | 4.5 | 6.3 | 150 |
| $\delta Q_L^\downarrow = 10\ W\,m^{-2}$ | 0 | 0 | 0 | 10 | 0 | 0 | 10.0 | 6.0 | 143 |
| $\delta\tau = 0.1$ | 36.0 | 0 | 16.2 | −19.6 | 0 | −4.6 | −8.0 | −3.2 | −76 |
| $\delta\alpha_S = 0.1$ | 0 | 0.1 | −22.7 | 0 | 0 | 0 | −22.7 | −13.6 | −323 |

**Table 4.** Net energy balance sensitivity to meteorological perturbations in the surface energy balance model, based on regressions to the sensitivity curves (cf. Figure 6). Also shown is the change in net energy associated with a 1-$\sigma$ increase in each parameter, averaged over JJA.

| Perturbation | Sensitivity | $\delta Q_N$ for +1$\sigma$ |
|---|---|---|
| 1. $\delta T = \pm 2°C;\ \delta h = 0;\ \delta\alpha_S = 0$ | $\partial Q_N/\partial T = 13\ W\,m^{-2}\,(°C)^{-1}$ | $+10\ W\,m^{-2}$ |
| 2. $\delta T = \pm 2°C;\ \delta h = 0$ | $\partial Q_N/\partial T = 27\ W\,m^{-2}\,(°C)^{-1}$ | $+21\ W\,m^{-2}$ |
| 3. $\delta T = \pm 2°C;\ \delta q_v = \delta\tau = \delta\varepsilon_a = 0$ | $\partial Q_N/\partial T = 21\ W\,m^{-2}\,(°C)^{-1}$ | $+17\ W\,m^{-2}$ |
| 4. $\delta T = \pm 2°C;\ \delta q_v = 0;\ \delta\tau,\ \delta\varepsilon_a$ | $\partial Q_N/\partial T = 17\ W\,m^{-2}\,(°C)^{-1}$ | $+13\ W\,m^{-2}$ |
| 5. $\delta q_v = \pm 50\%;\ \delta\tau,\ \delta\varepsilon_a = 0$ | $\partial Q_N/\partial q_v = 15\ W\,m^{-2}\,(g/kg)^{-1}$ | $+5\ W\,m^{-2}$ |
| 6. $\delta q_v = \pm 50\%;\ \delta\tau,\ \delta\varepsilon_a$ | $\partial Q_N/\partial q_v = 25\ W\,m^{-2}\,(g/kg)^{-1}$ | $+8\ W\,m^{-2}$ |
| 7. $\delta v = \pm 50\%$ | $\partial Q_N/\partial v = 14\ W\,m^{-2}\,(m/s)^{-1}$ | $+3\ W\,m^{-2}$ |
| 8. $\delta\tau = \pm 0.1$ | $\partial Q_N/\partial\tau = -9\ W\,m^{-2}\,(0.1)^{-1}$ | $-4\ W\,m^{-2}$ |
| 9. $\delta\alpha_S = \pm 0.1$ | $\partial Q_N/\partial\alpha_S = -27\ W\,m^{-2}\,(0.1)^{-1}$ | $-16\ W\,m^{-2}$ |
| 10. $\delta b_w = \pm 1\ m\,w.e.$ | $\partial Q_N/\partial b_w = -12\ W\,m^{-2}\,(m\ w.e.)^{-1}$ | $-3\ W\,m^{-2}$ |

**Table 5**. Summer surface energy balance fluxes on Haig Glacier as forced by the North American Regional Reanalysis (NARR) daily weather fields, 1979-2014. NARR inputs are taken as perturbations to the mean observed values. Melt is in m w.e., and all fluxes have units $\text{W m}^{-2}$.

| Period | $Q_S^{\downarrow}$ | $\alpha_s$ | $Q_L^{\downarrow}$ | $Q_L^{\uparrow}$ | $Q_H$ | $Q_E$ | $Q_C$ | $Q_N$ | melt |
|---|---|---|---|---|---|---|---|---|---|
| JJA | $227 \pm 7$ | $0.53 \pm 0.05$ | $275 \pm 4$ | $311 \pm 1$ | $27 \pm 4$ | $-3 \pm 3$ | $2 \pm 1$ | $95 \pm 14$ | $2.28 \pm 0.42$ |
| MJJAS | $215 \pm 6$ | $0.55 \pm 0.04$ | $271 \pm 4$ | $308 \pm 2$ | $22 \pm 3$ | $-5 \pm 3$ | $3 \pm 1$ | $73 \pm 10$ | $2.68 \pm 0.50$ |

**Table 6**. Correlation and sensitivity of different weather variables to the mean summer (JJA) net energy flux, $Q_N$, for the NARR simulations, 1979-2014. 'cloud' is the NARR total cloud fraction.

| Variable | Correlation | Sensitivity | $\delta Q_N$ for $+1\sigma$ |
|---|---|---|---|
| $T\,(°C)$ | 0.84 | $\partial Q_N / \partial T = 14\ \text{W m}^{-2}\,(°C)^{-1}$ | $+10\ \text{W m}^{-2}$ |
| $q_v\,(\text{g kg}^{-1})$ | 0.50 | $\partial Q_N / \partial q_v = 25\ \text{W m}^{-2}\,(\text{g/kg})^{-1}$ | $+7\ \text{W m}^{-2}$ |
| $v\,(\text{m s}^{-1})$ | 0.00 | $\partial Q_N / \partial v = -4\ \text{W m}^{-2}\,(\text{m/s})^{-1}$ | $-1\ \text{W m}^{-2}$ |
| $Q_S^{\downarrow}\,(\text{W m}^{-2})$ | 0.14 | $\partial Q_N / \partial Q_S^{\downarrow} = 0.3\ \text{W m}^{-2}\,(\text{W m}^{-2})^{-1}$ | $+2\ \text{W m}^{-2}$ |
| $Q_L^{\downarrow}\,(\text{W m}^{-2})$ | 0.64 | $\partial Q_N / \partial Q_L^{\downarrow} = 2\ \text{W m}^{-2}\,(\text{W m}^{-2})^{-1}$ | $+8\ \text{W m}^{-2}$ |
| $\tau$ | 0.25 | $\partial Q_N / \partial \tau = 15\ \text{W m}^{-2}\,(0.1)^{-1}$ | $+4\ \text{W m}^{-2}$ |
| cloud | $-0.19$ | $\partial Q_N / \partial c = -8.1\ \text{W m}^{-2}\,0.1)^{-1}$ | $-3\ \text{W m}^{-2}$ |
| $\alpha_S$ | $-0.83$ | $\partial Q_N / \partial \alpha_S = -26\ \text{W m}^{-2}\,(0.1)^{-1}$ | $-11\ \text{W m}^{-2}$ |
| $b_w\,(\text{m w.e.})$ | $-0.15$ | $\partial Q_N / \partial b_w = -3\ \text{W m}^{-2}\,(\text{m w.e.})^{-1}$ | $-1\ \text{W m}^{-2}$ |

**Figures**

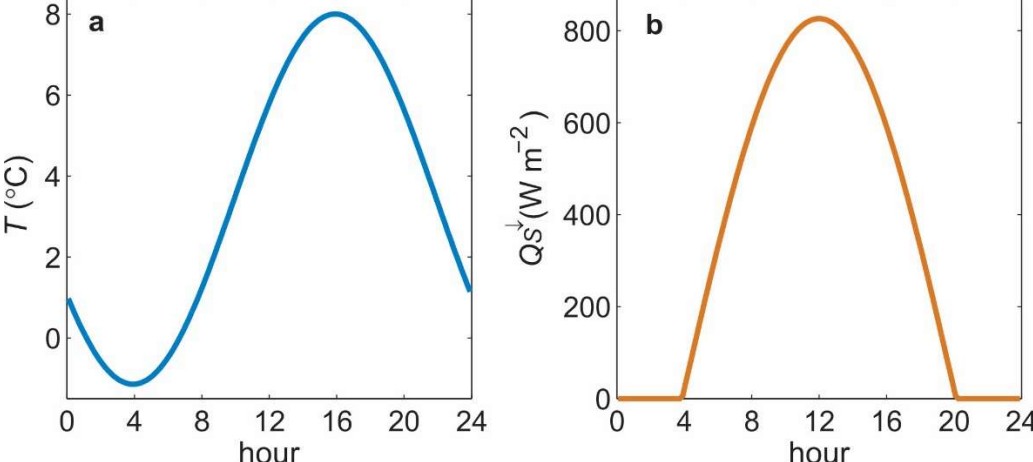

 **Figure 1.** Idealized diurnal cycles of (a) temperature and (b) incoming shortwave radiation used
 in the energy balance model. These two examples are for a sample day, July 1, 2010, parameterized
 from daily minimum and maximum temperature in (a) and day of year plus mean daily incident
 shortwave radiation in (b).

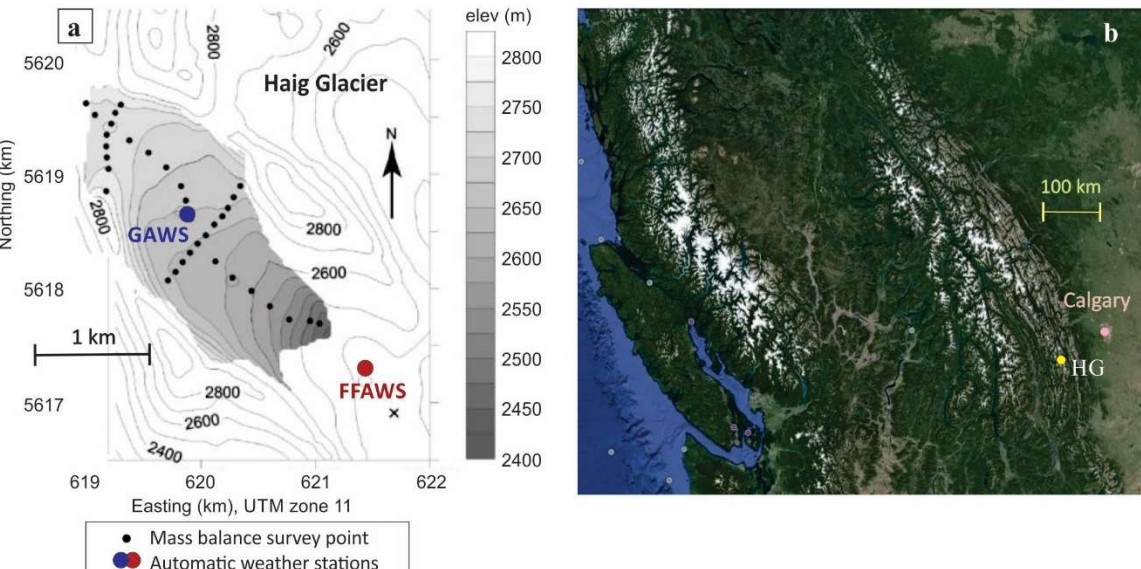

 **Figure 2.** (a) The topography and automatic weather stations on Haig Glacier (GAWS) and the
 glacier forefield (FFAWS). The smaller black dots are mass balance survey points. (b) The
 location of Haig Glacier is labelled HG on the Google Earth map of southwestern Canada.


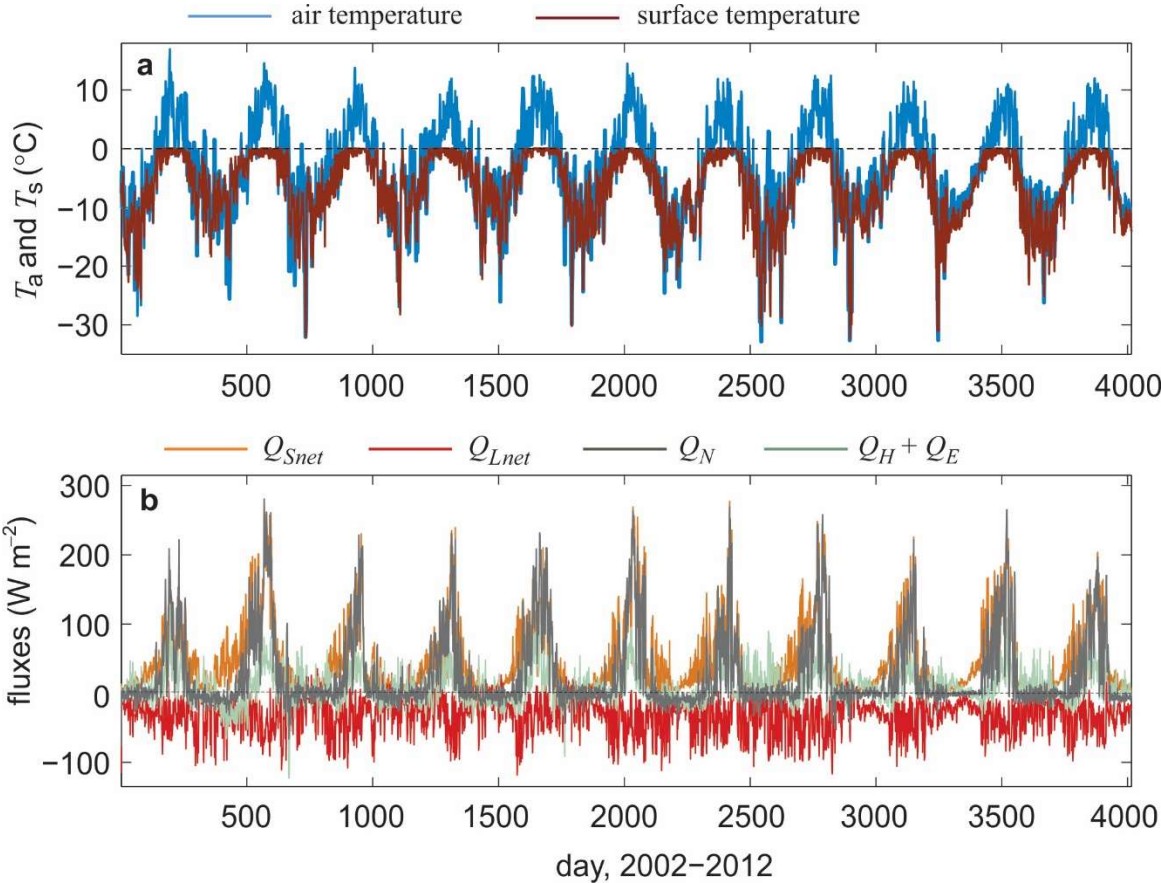

**Figure 3.** The 11-year record of (a) air temperature, modelled surface temperature, and (b) surface
energy fluxes at the Haig Glacier AWS site. Daily mean values are plotted from Jan 1, 2002-Dec
1344 31, 2012.


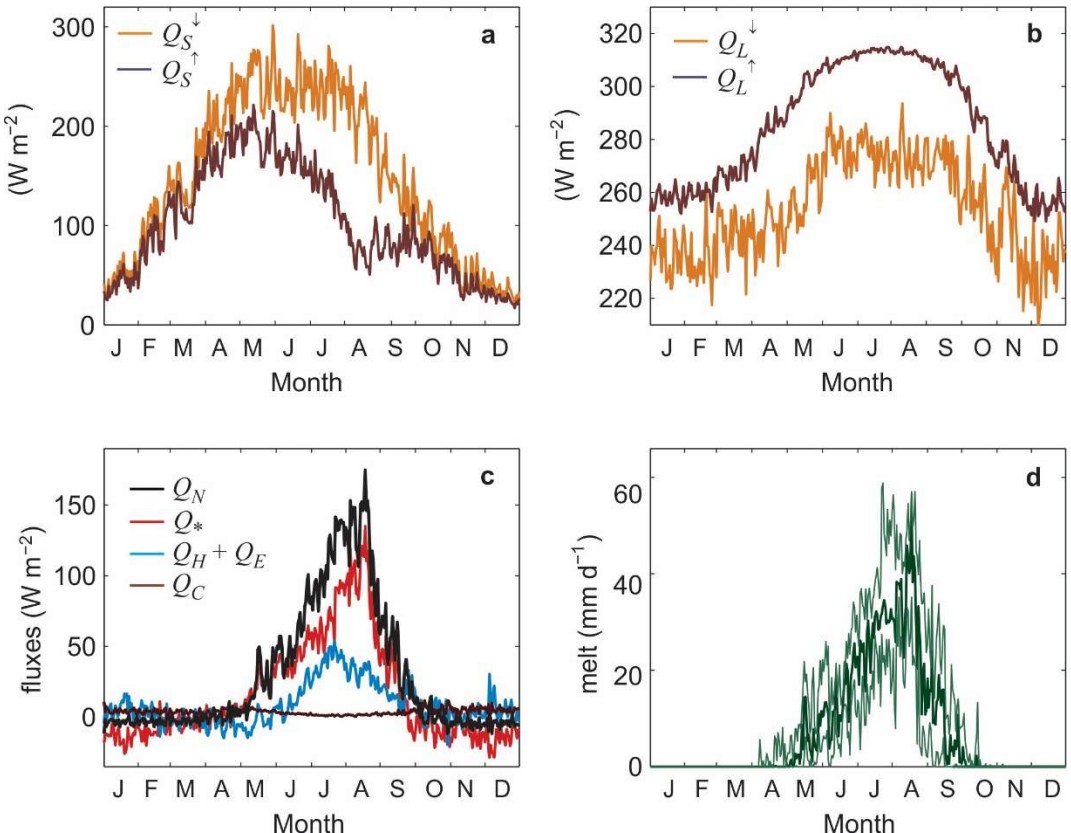

**Figure 4.** The average annual cycle of (a-c) surface energy fluxes and (d) daily melt at the Haig Glacier AWS. Daily mean values are plotted for the period 2002-2012. For melt rates, the heavy line is the median value and the thin lines indicate the interquartile range.


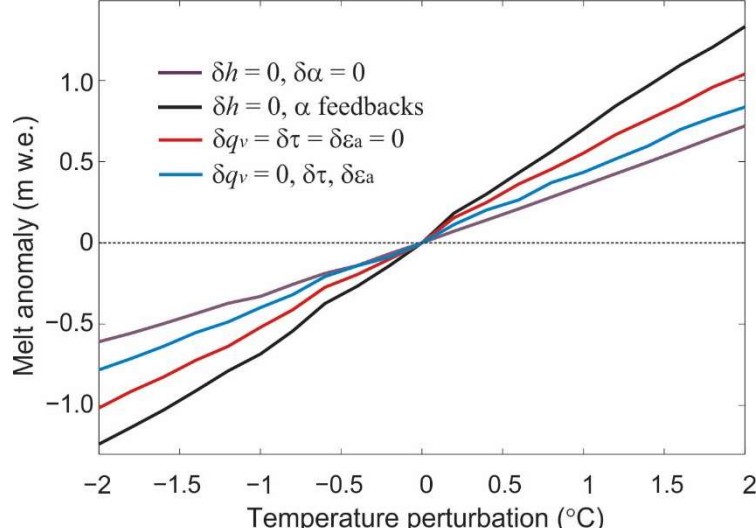

**Figure 5**. Sensitivity of modelled summer (JJA) melt to temperature perturbations for different
assumptions, as per Table 4. The reference (mean 2002-2012) JJA melt is 2.32 m w.e.

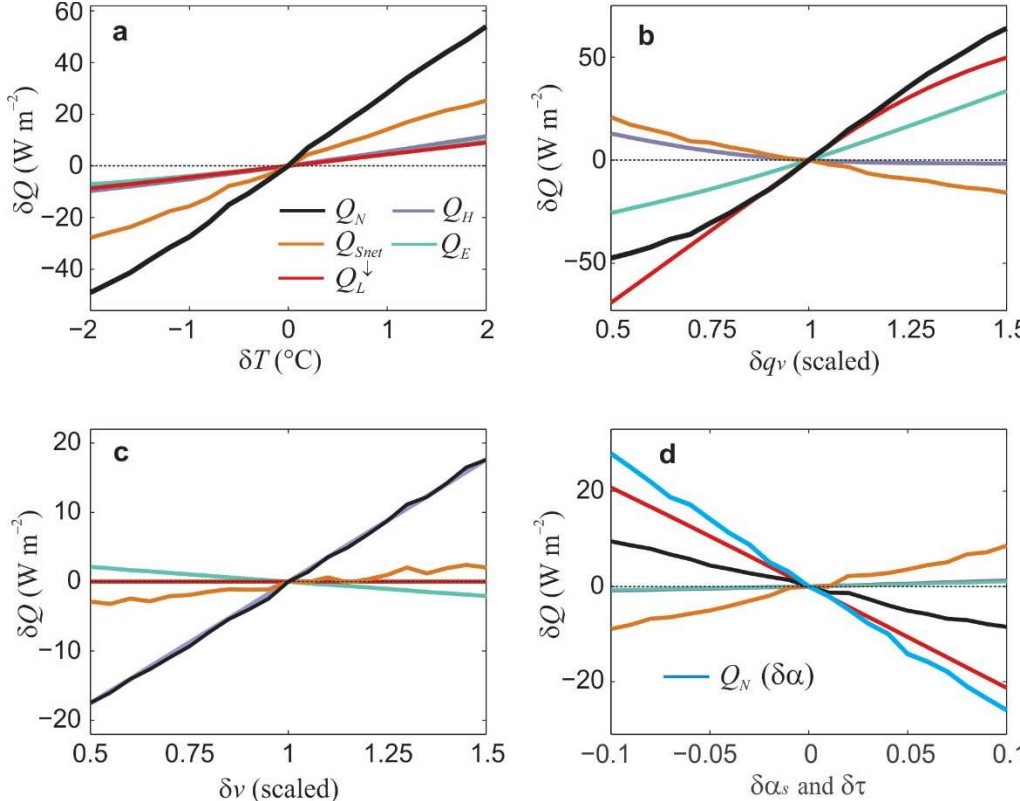

**Figure 6**. Sensitivity of the surface energy fluxes at Haig Glacier to changes in (a) temperature
(case 2), (b) specific humidity (case 6), (c) wind speed (case 7), and (d) atmospheric transmittance
(case 8) and albedo (blue line, case 9). All lines are anomalies relative to the baseline data from
the period 2002-2012, and indicate the mean sensitivity of the different energy fluxes over this
period. Please note the different $y$ ($\delta Q$) scales.




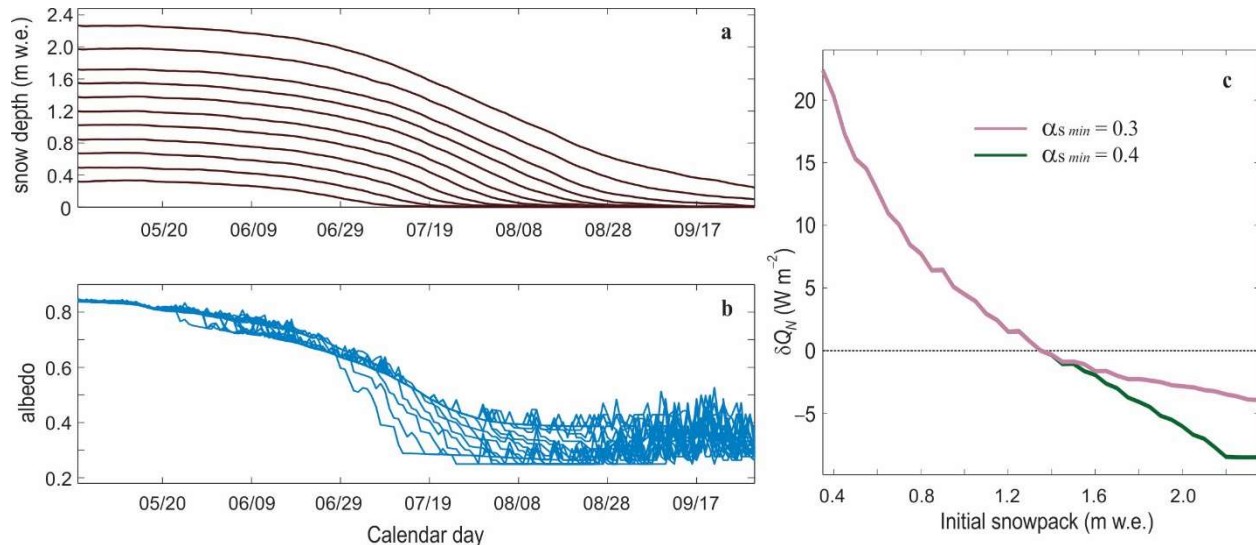

**Figure 7**. Sensitivity to the winter mass balance, examined by varying May 1 snow depth from 0.36-2.36 m w.e., relative to the reference value of 1.36 m w.e. at the glacier AWS. (a) Snow depth and (b) albedo through the summer melt season, May 1-Sept 30, for the different initial snow depths. (c) Net summer (JJA) energy balance change as a function of the winter mass balance for two different settings of the minimum snow albedo.

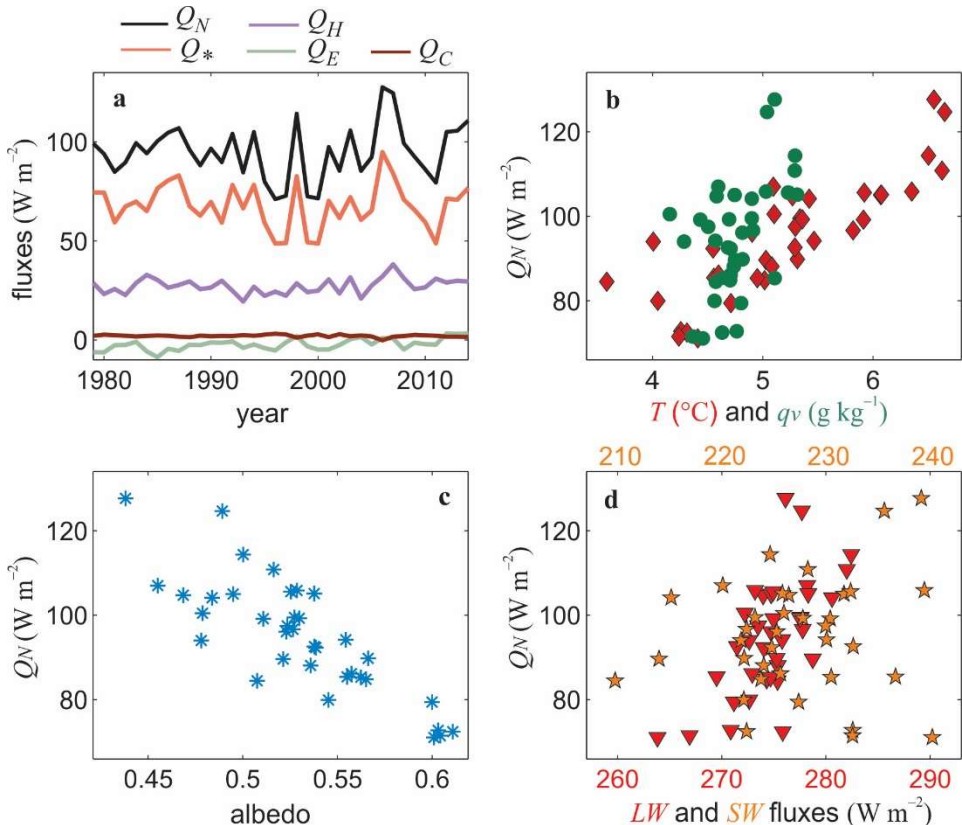

**Figure 8**. a) Mean summer (JJA) NARR-forced surface energy fluxes at Haig Glacier, 1979-2014. Mean summer net energy as a function of (b) temperature and specific humidity, (c) albedo, and (d) incoming shortwave and longwave radiation. Table 6 gives the associated correlations.

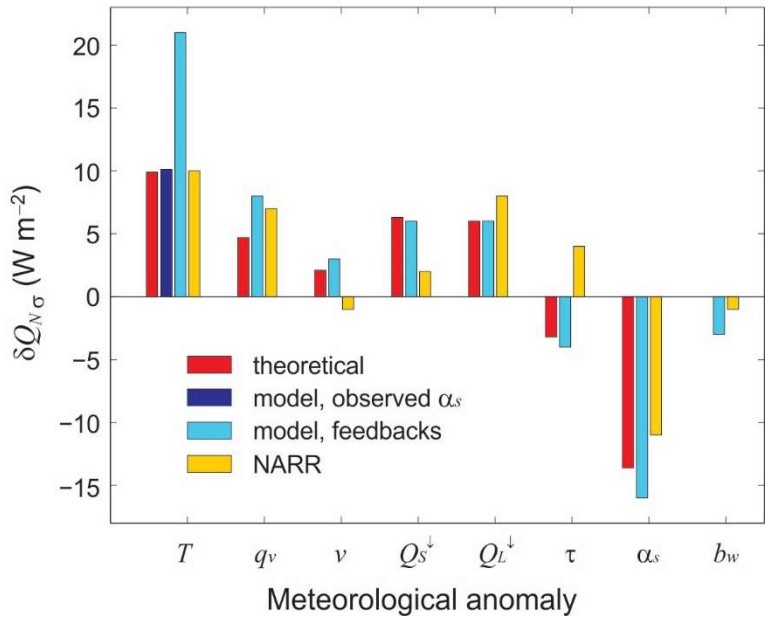


**Figure 9**. Net energy sensitivity to a 1-σ perturbation in different meteorological variables:
comparison of theoretical, *in situ* numerical model, and NARR-based estimates.







