# Peer review of "Surface Energy Balance Sensitivity to Meteorological Variability on Haig Glacier, Canadian Rocky Mountains"

_The Cryosphere, 2016_

## Referee Comment (RC1) · Anonymous Referee #1 · 25 Feb 2016

General Comments

The manuscript explores the sensitivity of surface energy balance components to summer climate perturbations for site over a small mountain glacier in the Canadian Rocky Mountains. Theoretical sensitivity is calculated using mean summer conditions, while empirical sensitivity is established using daily variability from 11 years of in-situ data. The paper also presents a reconstruction of summer melt from reanalysis data for the later part of the 20th Century. The paper is generally well written with well-presented figures and a logical progression through the results.

However, there are significant shortcomings in the methods that limit the usefulness of the results in address the key questions posed. In particular, it is well established

that feedbacks are important mechanisms in determining glacier sensitivity to climate, in particular those between air temperature, precipitation and albedo (Oerlemans and Fortuin, 1992). For this reason, models assessing the sensitivity of melt (or mass balance) to climate perturbations require, 1. Driving data that covers the full range of meteorological conditions through multiple seasons, 2. A model that includes formulations for surface energy balance components that allow for important feedbacks (i.e. dynamic albedo, variation of incoming longwave with air temperature and humidity, dynamic surface temperature to include variations in refreezing/sub-surface conduction, 3. Study periods that include the full season to include air temperature/ precipitation / albedo feedbacks.

The authors reflect on most of these points throughout the results/discussion, but fail to adequately address them in the methods chosen. This undermines the results and ultimately reduces the interpretations that can be made from the data. The use of theoretical sensitivity based on mean summer conditions does not meet the criteria above and is subject to many assumptions implicit in the formulae used. It may provide an efficient way to assess the sensitivity over a large number of glaciers (and elevations on each glacier), but it would have to be carefully compared to the sensitivity assessed using realistic meteorological forcing across the full season over a large number of glaciers. Similarly, the use of reanalysis data perturbations could provide a useful method to derive sensitivity, again if the method could be shown to work for a number of glaciers in a variety of geographic settings. Unfortunately, the results of the NARR reanalysis driven surface energy balance conflict with the in-situ data here, so little can be interpreted from the seemingly accidental good model performance.

If the authors can work to robustly test their methods at a number of sites, and carefully redefine the focus of the work as presenting a new method for efficiently assessing sensitivity then it may be acceptable. If the authors wish to remain focused on the climate sensitivity of this particular glacier, then they need to employ methods appropriate to the task and put their results more carefully in the context of previous efforts to understand climate sensitivity.

Specific Comments

Ln 27 – The abstract needs to be clearly state what the results of NARR analysis indicated.

Ln 40 – Interesting choice of the words 'banal' and 'trivial'. Perhaps these apply to the general public but likely not the readers of the current journal. Please revise.

Ln 45 – The introduction needs to present a more thorough review of the atmospheric controls on glacier mass balance, in particular the link between air temperature and mass balance (melt) on extratropical glaciers discussed in papers such as Oerlemans (2005) and Sicart et al. (2008) and references therein.

Ln 66 – "capture the impact of shifts" perhaps add "in other climate variables such as".

Ln 75 – while perhaps not commonplace, surface energy balance – mass balance models have been used extensively to investigate glacier-climate interactions and sensitivity (Gerbaux et al., 2005; Greuell and Smeets, 2001; Klok and Oerlemans, 2004; Mölg et al., 2008). Please revise.

Ln 85-90 – The introduction needs to more clearly define what is being examined – the sensitivity of surface energy balance components, or melt, or mass balance? – and over what time period – the sensitivity of melt to summer meteorology or annual climatology? The results should then align with the objective defined. Certainly inter-annual variations in air temperature will impact the fraction of rain vs snow and thus the winter accumulation and from this the albedo and melt through the timing of the snow-ice transition. If the authors wish to examine the sensitivity of mass balance or melt to climate change it is imperative that modelling is conducted over full seasons.

Ln 143 – It is contradictory to state a sophisticated model is 'needed' if you go on to use a parameterization that does not perform these calculations. Perhaps it would be accurate to state that one needs to take into account the profile of lower tropospheric

water vapour, cloud and temperature. Ln 206-214 – This paragraph appears to be out of place. Please move to introduction.

Ln 240 – It is ambiguous how the diurnal cycle of is parameterized. Please explain.

Ln 261 – Theoretical sensitivity – As discussed in the general comments, a robust assessment of sensitivity needs to consider the full range of meteorological variation. The results of the theoretical and empirical sensitivity differ in important ways and thus, the theoretical sensitivity cannot be said to add anything beyond the standard of modelling the full season. Either this section needs to be removed, or developed further into a distinct methodology that is validated at a number of sites.

Ln 468 - Please explain why daily time steps were used when the computational cost of hourly sub-hourly steps is not great? Much important information is lost at a daily time step, even with a parameterized diurnal cycle and further discussion of the effects on the results is warranted.

Ln 478 – Please state what fraction of data are missing/gap filled, in particular the incoming longwave data.

Ln 492 – The feedbacks need to be clearly explained here, as equation 14 indicates there will be positive feedbacks that will enhance the variation of incoming longwave with humidity.

Ln 517 – It is essential that incoming longwave vary with humidity for an assessment of sensitivity to be robust. By using measured and parameterized data this becomes ambiguous and parameterized data should be used exclusively.

Ln 517 – You have the opportunity to include the effects of humidity on incoming shortwave radiation (through equation 9). As you note, this can overwhelm influence on incoming longwave radiation (Ln 319). The inclusion of this effect would be novel application of the empirical model.

Ln 524 – Your results indicate the feedbacks are important (Ln 541) and your conclusions should echo this more strongly.

Ln 560 – This assumption is likely to be incorrect and the effect of subsurface heat fluxes needs to be considered (e.g. Pellicciotti et al. (2009).

Ln 574-576 – Further explanation of this method is needed i.e. how did you treat variations in moisture - as changes in qv or in RH? If the former, then perhaps you will overestimate the actual variation as qv variations at lower altitudes will be larger.

Ln 586 – This statement seems to contradict the previous statement that most important radiative inputs are not well correlated on an inter-annual basis and that the variance of the shortwave does not correspond with the in-situ. As there is distinct seasonal variations in air temperature and solar radiation, these variables are heavily auto-correlated and a more meaningful correlation would remove the seasonal trend before correlating variables between NARR and in-situ data.

Ln 629 – As biases in NARR results only happen to cancel and thus produce correct estimates of melt energy, these results cannot be considered robust enough to provide a meaningful interpretation of the inter-annual variations in the surface energy fluxes. Either the interpretations need to be carefully explained in this light, or further work is needed to demonstrate acceptable model skill.

Ln 654 – The approach presented in this paper has already been fairly well established in the literature (see comment for Ln 75) and so some additional novelty needs to be displayed here.

Ln 763-765 – Further explanation of the differences between theoretical and empirical sensitivities is needed.

Ln 770-771 – The trends in energy fluxes need to be more closely tied into the results of the sensitivity study.

References

Gerbaux, M., Genthon, C., Etchevers, P., Vincent, C., and Dedieu, J. P.: Surface mass balance of glaciers in the French Alps: distributed modeling and sensitivity to climate change, Journal of Glaciology, 51, 561-572, 2005.

Greuell, W. and Smeets, C. J. P. P.: Variations with elevation in the surface energy balance on the Pasterze (Austria), Journal of Geophysical Research, 106, 31717-31727, 2001. Klok, E. J. and Oerlemans, J.: Modelled climate sensitivity of the mass balance of Morteratschgletscher and its dependence on albedo parameterization, International Journal of Climatology, 24, 231-245, 2004.

Mölg, T., Cullen, N. J., Hardy, D. R., Kaser, G., and Klok, L.: Mass balance of a slope glacier on Kilimanjaro and its sensitivity to climate, International Journal of Climatology, 28, 881-892, 2008.

Oerlemans, J.: Extracting a climate signal from 169 glacier records, Science, 308, 675-677, 2005.

Oerlemans, J. and Fortuin, J. P. F.: Sensitivity of glaciers and small ice caps to greenhouse warming, Science (New York, N.Y.), 258, 115-117, 1992.

Pellicciotti, F., Carenzo, M., Helbing, J., Rimkus, S., and Burlando, P.: On the role of subsurface heat conduction in glacier energy-balance modelling, Annals of Glaciology, 50, 16-24, 2009. Sicart, J. E., Hock, R., and Six, D.: Glacier melt, air temperature, and energy balance in different climates: The Bolivian Tropics, the French Alps, and northern Sweden, Journal of Geophysical Research, 113, 2008.

---

## Referee Comment (RC2) · Anonymous Referee #2 · 1 Mar 2016

**General comments**

In this paper, theoretical considerations as well as an energy balance model are employed to assess the surface energy balance sensitivity to variations in meteorological variables. The methods are applied at an automatic weather station (AWS) site on a mid-latitude glacier in the Canadian Rocky Mountains. In addition to the in situ AWS observations over the period 2002–2012, meteorological data from a reanalysis product (1979–2014) are used to force the model. Only the main melt season (June-August or May-September) is considered.

The paper reads well and is written in good English. However, the methods used are not always described in enough detail, in particular regarding the energy balance

model. Some model elements are not introduced at all, others are mentioned at a too late point in the manuscript. See the specific comments below for an overview.

Apart from model parts not being described, I do not think the model and approach used are suitable for the sensitivity analysis performed in this paper. The surface energy balance contains important feedback mechanisms, which are pointed out by the authors at places in the manuscript. Although they account for albedo changes associated with increased surface melt, they do not seem to include the opposite effect of summer snowfalls on the albedo. More importantly, they do not calculate surface temperature internally in the model, while this variable is easily affected by changing atmospheric conditions. In its turn, it changes the outgoing longwave radiation and the turbulent fluxes. The authors do mention that surface temperature is generally at the melting point in the summer months, but not in the early and late melt season. Still, most of their results are presented for the entire melt season.

The same applies to the theoretical derivations of the energy flux sensitivity, they also do not take changes in surface temperature into account. However, here the main results are presented for the months June-August only. This theoretical approach does present a simple method to estimate changes in the surface energy balance resulting from variability in the meteorological conditions. The results compare well to the model results, but not for all variables, suggesting some feedbacks are overlooked in the energy balance model. Whether this is a general method that can be transferred to other glaciers can only be established by similar applications on other glaciers with energy balance observations.

Another major shortcoming of the energy balance model used is that incoming longwave radiation is taken from the measurements in the sensitivity analysis and not recalculated. As incoming longwave radiation is affected by both changes in air temperature and humidity (and cloudiness, here parameterized through relative humidity changes), the sensitivities are severely underestimated. This is also revealed from the comparison with results from the theoretical approach.

[Figure]

The model simulations with reanalysis input serve as an application of the 'perturbation' method presented. After reading the paper, I am still not sure what this method exactly is, but it is not as novel as the authors present it to be. I think the authors mean that the energy balance model is run with anomalies imposed on the 2002–2012 in situ conditions. But in fact, they are just forcing the model with a different set of (bias-corrected) meteorological data. As the connection of this exercise to the sensitivity analysis presented before is rather weak, I doubt whether this is a valuable addition to the paper.

**Specific comments**

72-76: These lines give the impression that it is not very common to perform sensitivity studies of the surface energy balance on glaciers. The sensitivity to changes in temperature and precipitation is however assessed in numerous studies, therefore I suggest to change the word 'Several' to something more appropriate. Sensitivities to other variables are indeed less often investigated, but there are more examples than the one given here (e.g. Oerlemans (1991) and Gerbaux et al. (2005)).

105: I wonder what the net energy flux $Q_N$ actually represents, the authors need to give a better description. If it is positive, it generates surface melt and is equivalent to what is often called the melt energy in other studies. But can it also be negative? The net energy as presented here seems to represent a residual flux, that should remain close to zero if the surface is not melting. Is this the case and is it set to zero then? Otherwise, it means that important processes in the energy balance are missing. Much later in the manuscript, on lines 471-473, I read that negative values are associated with refreezing. I do not think this can be assumed that simply, refreezing requires a snow/ice model which seems not to be included here.

171: This equation implies that $h > 30$ for all times, is this indeed the case? It would be neater to include a minimum condition, in case $h < 30$.

192-193: More detail is needed about the roughness length scales, as there are different ways to derive their values and treat/calculate them in the model. According to the cited Marshall (2014) paper, constant values were used for all three length scales with a predefined ratio between them (mentioned much later in line 326-327). Their values were obtained by closing the surface energy balance. This should be mentioned here as well. I also wonder whether values differ for snow and ice surfaces?

205: The paper does not mention how the subsurface conductive heat flux is calculated, is a vertical model used to keep track of snow/ice temperatures, densities and water content? Please add a few lines.

212-214: I have the impression that $Q_E$ is generally positive on mid-latitude glaciers during the melt season, or slightly negative. See for more examples the tables in Ohmura (2001) and Giesen et al. (2009).

231-233: Which percentage of the data needed to be gap-filled? Do you mean that factors are derived for months when data from both AWSs were available?

237-239: Is the mean daily value taken from the same day in other years?

239-244: I am puzzled why the authors chose to use daily input data with imposed daily cycles instead of running the model at the resolution of the observations. There is a slight gain in efficiency, but at the cost of loosing important information to calculate the surface energy balance fluxes. Especially for the sensitivity of the surface energy balance, it is important to have enough detail. Perhaps the climate model output has a lower time resolution, but then a daily cycle can be imposed there. In any case, it would be better to provide details about the daily cycle here, where the first questions arise and not at the later point in the paper.

324-325: Air density is also assumed not to vary with temperature changes. Instead of using 'independent of temperature', which is of course not true, it might be more accurate to use 'can be assumed constant for small temperature changes'.

382-389: With unit forcings, as is done here, the sensitivities to changes in the different

variables cannot really be compared. Better compare the effects on the different energy fluxes only per variable and leave the comparison between variables for the standard-deviation-based forcings later in the section.

404: This subsection title is not well chosen, since the sensitivity of all energy balance fluxes to changes in meteorological variables is considered in this section. Net solar radiation is an energy flux itself and it is not the variable that is changed in this subsection. Instead, the effect of changes in top-of-the-atmosphere insolation, atmospheric transmissivity and surface albedo on the energy balance are the subject of this subsection. Please change the title accordingly.

471-473: As mentioned before, to get a good estimate of the amount of refreezing meltwater and the associated heat release, a vertical snow/ice model is needed. Here, refreezing occurs whenever air (not surface?) temperature is negative, regardless of the amount of available water. If the period before has been cold as well, there will not be any water present. Even if water is present following a melt event, there may not be enough to release the amount of heat following from Eq. (2). I therefore think this is not a good way to compute refreezing and would either neglect it altogether or use a proper subsurface model.

476-478: I do not understand how sensitivity analysis can be done if measured longwave radiation is used. Incoming longwave radiation needs to be adapted for different temperature and humidity. If the authors first show (in a figure) that using Eqs. (6) and (7) gives good correspondence with measured incoming longwave radiation, then they can use these equations with new temperature and/or humidity. This would largely reduce the difference in sensitivities to temperature and humidity changes obtained from the theoretical approach and the energy balance model. Outgoing longwave radiation should also be allowed to change, unless the surface temperature is always at the melting point. But for negative air temperature anomalies, the surface temperature will often be lower as well.

481-483: Many questions arose here, concerning the implementation of the changes in the energy balance model. Changes in air temperature will affect the fraction of precipitation falling as snow/rain, is this included in the model? Is snow depth tracked in the model to determine changes in the moments of ice (dis)appearance? How is albedo treated if the ice appears earlier than in the observations, is an ice albedo prescribed then?

486-488: The authors should make clear here which part of the year is used in the analysis. They mention that anomalies are applied to the entire year. But nowhere, except in the title of Fig. 2, it is mentioned that the analysis is performed over the months May-Sep.

538-542: The albedo feedback has a smaller effect for negative temperature perturbations. Is this because increases of snowfall events are not included?

557-560: How representative is the assumption of a melting surface, this can easily be judged from the measurements. In Table 2, I see that especially in May, outgoing longwave radiation is considerably lower than 315 W m$^{-2}$. Can you give the fraction of the time with a melting surface to the total time?

616-621: Why include the shoulder months in the analysis if they are not represented well in the model? Although it would still be better if the processes themselves would be included in the energy balance model.

694-695: Summer snow events also bring additional mass to the glacier, further reducing the net melt.

757-758: Please be more specific about which feedbacks are actually included. Only the internally modelled snow-aging is described in lines 529-536, it is still not clear to me to what extent and how the snow/ice transition and snowfall events are included.

Table 6: In general, I think the manuscript contains a relatively large number of tables and a small number of figures. Especially this table contains too much information to

serve a purpose and it also needs to be compared to another table. Please make the comparison easier, by visualizing the monthly energy balance fluxes for the in situ data, the NARR perturbed data (and optionally the NARR raw data) together in a figure.

Figure 2: As longwave radiation is (not yet) allowed to change, the effect of net radiation corresponds to the effect of net shortwave radiation alone. Better present it this way and add a line for net longwave radiation, when it is also varied. I would also like to see a line for the summed effect on $Q_N$, which is especially illustrative for the opposite effects found for wind speed changes.

Figure 6: Why is albedo shown for JJA instead of MJJAS, as the other variables? I would like to see the net shortwave and net longwave radiation separately instead of net radiation, as these are treated individually throughout the manuscript. I do not think it is necessary to show both net energy and melt, because they are directly related.

**Technical corrections**

40: I would not consider the word 'banal' fit for scientific papers, please rephrase.

55: 'reanalyses'

60: 'for snow and ice melt factors'

69: 'crucial to ablation on'

116: 'solar radiation that is reflected'

123: $\phi_0$ is used in Equation (3) instead of $\psi_0$

150: As Kwadacha Glacier is not the subject of this paper, better rewrite as 'At two study sites'

158: 'ratio of potential direct to measured' (or is measured radiation only direct radiation as well?)

159: Include a reference here, is it the paper mentioned in the next line?

186-187: Split into two sentences: ' and $q$ ... humidity. Measurements... levels, at the surface-air... and at height ... surface.'

189-190: Reorder: 'We estimate $T_s$ from an inversion of Eq. (5), using'

193: 'can be'

229: 'meteorological conditions'

264: 'Warm summers generally cause'

265: 'but the energy balance is sensitive to'

286-287: 'of the response to a temperature change'

305: Remove the spaces in 100 (1 00)

332: Include the dot on $m$ as in Eq. (2)

340: 'at the AWS site'

392-393: Split into two sentences: 'Following Eq. (9),'

407: I wondered what was meant by 'solar variability' and found the answer in line 424-425, better move it here.

415: Is $Q_{S0}$ equivalent to $Q_0$ introduced in Eq. (3)? If yes, use the same notation, if no, clarify the difference.

445: 'last two lines'

495-496: Mention that results for simultaneous changes in temperature and humidity are not shown here.

497-500: These lines belong in the figure caption, not in the main text.

500: 'Sensitivity to albedo changes over'

507: 'directly'

510: 'The sensitivities computed with/resulting from the surface...'

512-514: These may be advantages, but are these effects included in the model used here?

528: 'induce'

558: What is the 'summer melt season'? May-Sep or Jul-Aug?

609-610: The wording should make clear that these energy fluxes are not taken from the NARR reanalysis, but calculated with the energy balance model using NARR meteorological forcing. Further down, 'NARR-based' is used frequently, this is already better.

661: 'changes in most meteorological variables'

668: 'Increases'

669: 'through the sensible and latent heat and incoming'

692-693: 'fraction of time with surface temperatures at the melting point'

698: 'as in the simple experiments presented in this paper'

699: What is meant with 'everything', please be more specific here.

726: 'balance'

747: 'allows for a'

771: Just write 'Net solar radiation', as longwave radiation is not allowed to change.

Table 1: Write out the definition of 'summer melt season' in the caption. Use SI units for air pressure (Pa or hPa)

Table 2: Caption: 'Mean monthly surface energy balance components/fluxes and monthly melt totals.' All details about the location can be left out, this can be read in the text and is also included in the caption of Table 1. Can you use symbol notation

for melt as well, being the sum of the melt rate?

Table 3: Note that all sensitivities are calculated using the JJA mean values, now this is only stated for $\delta Q_N$. Furthermore, in the table on line 998, there is no apparent change with regard to the previous line. However, $\delta h$ is not zero here, which should be mentioned. On line 1003, it is not $Q_S$ (a variable that has not even been introduced) that is varied, but $Q_0$.

Table 6: 'NARR-based mean monthly...'

Figure 1: Either note that KG indicates Kwadacha Glacier, which is mentioned once in the paper or remove the dot and zoom in on the map around Haig Glacier. I suggest to do the latter.

Figure 2: Remove the figure title above the panels and add the the melt season period to the caption. Include a legend to indicate the different fluxes and remove from the caption, this makes the figure and caption easier to read. Showing albedo changes as absolute or relative (%) values is not exactly the same, if you like to use the same scale as for shortwave radiation, then just say 10 x albedo change. Since the x-axis label also only mentions the shortwave perturbation, it may be a better solution to use the upper x-axis to indicate the albedo scale and title. Furthermore, 'SW' is now used for shortwave radiation instead of $S$, please be consistent with notation throughout the manuscript.

Figure 3: Please use the same variables and colours as in Figure 2.

Figure 4: 'Table 5 gives the bias and correlations.'

Figure 5: More tick marks are needed on the x-axis, at least for every five years.

**References**

Gerbaux, M., Genthon, C., Etchevers, P., Vincent, C., Dedieu, J. P.: Surface mass balance of glaciers in the French Alps: distributed modeling and sensitivity to climate

change. Journal of Glaciology, 2005, 51, 561-572, doi:10.3189/172756505781829133

Giesen, R. H., L. M. Andreassen, M. R. van den Broeke en J. Oerlemans: Comparison of the meteorology and surface energy balance on Storbreen and Midtdalsbreen, two glaciers in southern Norway. The Cryosphere, 2009, 3, 57-74, doi: 10.5194/tc-3-57-2009.

J. Oerlemans: The mass balance of the Greenland ice sheet: sensitivity to climate change as revealed by energy-balance modelling. The Holocene, 1991, 1, 40-48, doi:10.1177/095968369100100106

Ohmura: Physical basis for the temperature-based melt-index method. J. Appl. Meteor., 2001, 40, 753–761, doi:10.1175/1520-0450(2001)040<0753:PBFTTB>2.0.CO;2

---

## Author Comment (AC1) · 25 Jun 2016

Interactive comment on "Surface Energy Balance Sensitivity to Meteorological Variability on Haig Glacier, Canadian Rocky Mountains" by S. Ebrahimi and S. J. Marshall

Anonymous Referee #1

General Comments

The manuscript explores the sensitivity of surface energy balance components to summer climate perturbations for site over a small mountain glacier in the Canadian Rocky

[Figure]

Mountains. Theoretical sensitivity is calculated using mean summer conditions, while empirical sensitivity is established using daily variability from 11 years of in-situ data. The paper also presents a reconstruction of summer melt from reanalysis data for the later part of the 20th Century. The paper is generally well written with well-presented figures and a logical progression through the results. However, there are significant shortcomings in the methods that limit the usefulness of the results in address the key questions posed. In particular, it is well established that feedbacks are important mechanisms in determining glacier sensitivity to climate, in particular those between air temperature, precipitation and albedo (Oerlemans and Fortuin, 1992). For this reason, models assessing the sensitivity of melt (or mass balance) to climate perturbations require, 1. Driving data that covers the full range of meteorological conditions through multiple seasons, 2. A model that includes formulations for surface energy balance components that allow for important feedbacks (i.e. dynamic albedo, variation of incoming longwave with air temperature and humidity, dynamic surface temperature to include variations in refreezing/sub-surface conduction, 3. Study periods that include the full season to include air temperature/ precipitation /albedo feedbacks. The authors reflect on most of these points throughout the results/discussion, but fail to adequately address them in the methods chosen. This undermines the results and ultimately reduces the interpretations that can be made from the data. The use of theoretical sensitivity based on mean summer conditions does not meet the criteria above and is subject to many assumptions implicit in the formulae used. It may provide an efficient way to assess the sensitivity over a large number of glaciers (and elevations on each glacier), but it would have to be carefully compared to the sensitivity assessed using realistic meteorological forcing across the full season over a large number of glaciers. Similarly, the use of reanalysis data perturbations could provide a useful method to derive sensitivity, again if the method could be shown to work for a number of glaciers in a variety of geographic settings. Unfortunately, the results of the NARR reanalysis driven surface energy balance conflict with the in-situ data here, so little can be interpreted from the seemingly accidental good model performance. If the authors can work to robustly

test their methods at a number of sites, and carefully redefine the focus of the work as presenting a new method for efficiently assessing sensitivity then it may be acceptable. If the authors wish to remain focused on the climate sensitivity of this particular glacier, then they need to employ methods appropriate to the task and put their results more carefully in the context of previous efforts to understand climate sensitivity.

Thanks to the reviewer for this insightful summary, and for pointing out here and below some of the limitations in our approach and analysis. We acknowledge that most of the reviewer's concerns are valid and we have done considerable extra work to address some of the main limitations in our study. Specific points are discussed in detail below. Concerning the main points here: we agree that it would be valuable to examine several different sites and glacio-climatic environments, and that was indeed our original idea within the PhD research of S. Ebrahimi – to examine glacier sensitivity to meteorological variability in different regions. This still needs to be done, but as always happens, we found that it was already relatively rich and involved to perform this analysis thoroughly at one site. Our particular glacier is small and is not of global interest, but the glacier and the climatic regime are typical of mid-latitude mountain glaciers in e.g. the Rockies or the Alps, and the general findings are relevant to these environments. As the reviewer points out, sensitivity analysis is not new. However, most prior studies focus on temperature and precipitation (appropriately so, these are the two most important variables for mountain glacier mass balance). We attempt to present a detailed sensitivity analysis for the full array of meteorological conditions that affect surface energy balance, both with and without feedbacks. One of our objectives is to systematically explore and document the magnitude of different feedbacks – we certainly recognize that these are essential in understanding glacier response to climate variability. The study takes advantage of an 11-year in-situ dataset that permits some exploration of interannual variability (requirement 1 above). We recognize that comparable datasets are available from a few other sites that would allow us to extend this work and its value, but this would be a different contribution involving a broader network of collaborators. To keep this manuscript focused, we propose instead to re-
main with our original goal of an in-depth analysis at this one site, but we take seriously the reviewers' concerns about the limitations of our methodology and analysis. Specifically, we have: (i) added a subsurface temperature and drainage/refreezing model to allow free determination of subsurface heat flux and surface temperature, which feeds into the sensible heat flux and outgoing longwave radiation; (ii) extended to year-round simulations, though our focus remains on the summer melt season (MJJAS and JJA); (iii) better described aspects of the precipitation, albedo and humidity feedbacks and effects, which were there already but perhaps not properly described and explored; and (iv) better couched our methods and results in the context of previous studies. We believe that the modelling approach addresses requirements (2) and (3) above and is appropriate to the objectives of our study, and we thank the reviewer for pushing us on this. Our response has taken some months because of the model development and testing that were needed to improve this study. Based on the two reviewers' comments, the reanalysis-based melt reconstructions have now been discussed differently in the manuscript, with this section reduced by about 50%. In the original manuscript, this section represented an application of the energy balance/melt model more than an extension of the sensitivity analysis, wandering into questions of mass balance reconstructions and trends. While this topic is certainly of interest, it is not relevant to the rest of the manuscript and so it was distracting from the focus. We now restrict the NARR discussion to an exploration of energy balance (summer mass balance) sensitivities, in line with the rest of the paper.

Specific Comments

Ln 27 – The abstract needs to be clearly state what the results of NARR analysis indicated. Abstract revised.

Ln 40 – Interesting choice of the words 'banal' and 'trivial'. Perhaps these apply to the general public but likely not the readers of the current journal. Please revise. Rewritten.

Ln 45 – The introduction needs to present a more thorough review of the atmospheric

controls on glacier mass balance, in particular the link between air temperature and mass balance (melt) on extratropical glaciers discussed in papers such as Oerlemans (2005) and Sicart et al. (2008) and references therein. Introduction rewritten to better describe past studies on this question, atmospheric controls on mass balance, as well as previous studies using sensitivity analyses.

Ln 66 – "capture the impact of shifts" perhaps add "in other climate variables such as". Revised as suggested.

Ln 75 – while perhaps not commonplace, surface energy balance – mass balance models have been used extensively to investigate glacier-climate interactions and sensitivity (Gerbaux et al., 2005; Greuell and Smeets, 2001; Klok and Oerlemans, 2004; Mölg et al., 2008). Please revise. Agreed, more commonplace than we conveyed. Several studies along these lines will be included in the revised submission.

Ln 85-90 – The introduction needs to more clearly define what is being examined the sensitivity of surface energy balance components, or melt, or mass balance? and over what time period – the sensitivity of melt to summer meteorology or annual climatology? The results should then align with the objective defined. Certainly interannual variations in air temperature will impact the fraction of rain vs snow and thus the winter accumulation and from this the albedo and melt through the timing of the snow-ice transition. If the authors wish to examine the sensitivity of mass balance or melt to climate change it is imperative that modelling is conducted over full seasons.

Apologies for our lack of clarity here. This will be rewritten. We have not aimed to examine the sensitivity of annual mass balance, rather just the summer (melt) season surface energy balance and summer melt. That said, we of course agree that summer energy balance and melt will be sensitive to the winter snowpack. This is implicitly included in our model/observations for the study period, 2002-2012, as we initialize each summer melt season with the observed May snow depth (mm w.e., based on winter mass balance surveys that are carried out each May). Hence we do not model

the snow accumulation through the winter, but observed interannual variability in the snow accumulation is included as an initial condition for the summer melt season simulations. Interannual variability in measured and modelled summer albedo therefore includes this influence, although we have not isolated or examined it. This will be discussed in the revised manuscript.

Unfortunately, we do not have a good model or empirical understanding of winter snow accumulation sensitivity to meteorological conditions at this glacier. We have 15 years of winter mass balance data from this site, but that is limited when it comes to statistical modelling, and winter mass balance does not have a significant correlation with simple metrics, such as mean winter temperature. It is much more synoptically governed, e.g., responding to variability in Pacific storm tracks. Hence we do not include a direct model of winter or annual mass balance here, or of the sensitivity of winter mass balance to climate variability. We certainly agree that this is necessary in model-based studies over longer time periods (e.g. climate change studies), where temperature-dependent processes such as rain/snow fractionation need to be included in model-derived winter snow accumulation.

Our focus is on the summer energy and mass balance, and we agree that summer melt is sensitive to the winter snowpack, through its influence on albedo. Hence we introduce a new sensitivity test to explore the effects of different winter snow accumulation, bw, on summer energy balance and melt. A new Figure 7 presents these results, and we have a broader discussion of these influences on the summer melt season. In the revised NARR work, we also include a brief analysis of the effects of winter mass balance variability (as modelled in a simple way) on summer melt.

Ln 143 – It is contradictory to state a sophisticated model is 'needed' if you go on to use a parameterization that does not perform these calculations. Perhaps it would be accurate to state that one needs to take into account the profile of lower tropospheric water vapour, cloud and temperature. Clarified as suggested; we mean only to emphasize that ours is a simplistic parameterization of something that is complicated to calculate

rigorously. But the parameterization still has some skill, vs. for instance reanalysis-based estimates of incoming longwave radiation or the null hypothesis of assuming the mean value.

Ln 206-214 – This paragraph appears to be out of place. Please move to introduction. Removed from here, with some of this content retained in the revised introduction.

Ln 240 – It is ambiguous how the diurnal cycle of is parameterized. Please explain. This detail is now added in section 2, which describes the model. We apologize for the lack of clarity. Our methodology is simple, so we were not sure this warranted the space, but it is not documented elsewhere and it is important to describe the methods plainly and explicitly. Where we use a 'directly observed' surface energy balance, we drive the energy balance model with observed 30-minute data (including measured albedo and outgoing longwave radiation). Where we do sensitivity tests or run the model with other meteorological input, such as from climate models, we follow the following procedure, which allows for internal (e.g. albedo) feedbacks:

(i) we input the daily mean variables for all meteorological fields, as well as daily minimum and maximum temperature;

(ii) a diurnal temperature cycle is parameterized as a cosine wave with a lag to give min/max temperature at 04:00 and 16:00 (as per local observations), with an amplitude AT = (Tmax – Tmin)/2;

(iii) a diurnal cycle for incoming shortwave radiation is parameterized as a half-cosine wave (values above 0), with a period T(d) = 2hs(d), where d is the day of year and hs is the number of hours of sunlight on day d. Sunlight hours can be calculated as a function of latitude and day of year (see the revised text). A lag is specified to give peak shortwave radiation at local noon, and the amplitude of the cosine wave is specified from ASW = $\pi$ QSd(down)/2, where QSd(down) is the mean daily incoming shortwave radiation. This last relation is derived from integrating the area under the cosine wave and equating it to the average daily value. This treatment implicitly includes daily cloud

effects that will reduce incoming shortwave radiation (via QSd(down)), but distributed evenly through the day; this neglects any systematic tendency for e.g. afternoon vs morning clouds. For simplicity, we also neglect the effect of zenith angle on atmospheric transmittance (i.e., lower transmittance for larger atmospheric path lengths in the morning and late afternoon), although this could be built into a more refined model.

(iv) we assume that wind, incoming longwave radiation, air pressure, and specific humidity are constant through the day, held to the mean daily value.

(v) albedo is modelled on a daily basis, decreasing as a function of melting (cumulative PDD) or increasing in the event of summer snow falls (see the text);

(vi) relative humidity has a diurnal cycle following temperature, which impacts incoming longwave radiation where we parameterize this from near-surface conditions;

(vii) subsurface and surface temperature (Ts) and QC are modelled with 10-minute to one-hour time steps (chosen for stability of the temperature solution), and Ts is used in the calculation of outgoing longwave radiation, sensible heat flux, and latent heat flux (via qs). The model is run year-round;

Taken together, this gives an estimate of 10-minute to one-hour melting,

(viii) meltwater percolates and either refreezes or runs off based on a simple drainage model, described briefly in the text. This is part of the snowpack model used to calculate Ts and QC.

(ix) the snowpack depth and surface albedo are updated and the integration continues through the year.

(x) winter snow accumulation is not directly modelled, but winter mass balance (the May snowpack) is treated as an 'initial condition' for the summer melt model. It is set to measured values of bw, which are from winter mass balance observations that are carried out each May, including a snow pit at the AWS site. For purposes of the subsurface temperature model, snow accumulates linearly through the winter (October

to May) to reach the annual observed value of bw.

Ln 261 – Theoretical sensitivity – As discussed in the general comments, a robust assessment of sensitivity needs to consider the full range of meteorological variation. The results of the theoretical and empirical sensitivity differ in important ways and thus, the theoretical sensitivity cannot be said to add anything beyond the standard of modelling the full season. Either this section needs to be removed, or developed further into a distinct methodology that is validated at a number of sites. We have retained this section, but rewritten it in places and added some analysis to take it a bit further and permit some direct comparisons with the empirical/numerical model. One thing that it shows, for instance, is the strength of different feedbacks relative to the idealized situation where only one variable changes. It also provides a basis for thinking about meteorological perturbations, e.g. if temperature increases, do we assume that specific humidity stays the same, such that RH will drop, or do we assume that qv will increase, to maintain constant RH? This is introduced in the theoretical sensitivities, and then used as two 'end members' in the empirical model. This is also true for estimation of atmospheric radiation feedbacks that can be roughly parameterized from the humidity – it is introduced in the theoretical discussion and then applied in the model.

Ln 468 - Please explain why daily time steps were used when the computational cost of hourly sub-hourly steps is not great? Much important information is lost at a daily time step, even with a parameterized diurnal cycle and further discussion of the effects on the results is warranted. This is now discussed more clearly. In fact, we use sub-daily time steps (right now, 10- or 30-minute), and the reference energy fluxes are based on the 30-minute AWS data. But for a more flexible model that can be driven by climate model reanalyses or projections, for instance, we developed the model to work with daily inputs, along with parameterizations of the diurnal cycle (see above) and sub-daily time steps to capture the important diurnal processes.

Ln 478 – Please state what fraction of data are missing/gap filled, in particular the incoming longwave data. This will be added to Table 1. It depends on the variable of

interest. For most AWS variables, such as temperature, data coverage is 63% annually for the period 2002-2012 (2519 of 4018 days). 90% for the core summer months, JJA (909 of 1012 days), and 86% for MJJAS (1441 of 1683 days). The longwave radiation sensor was installed in July 2003 so there is more missing data. Coverage is as follows: annual - 46% (1835/4016 days); JJA – 76% (773/1023 days); MJJAS – 70% (1184/1683 days).

Ln 492 – The feedbacks need to be clearly explained here, as equation 14 indicates there will be positive feedbacks that will enhance the variation of incoming longwave with humidity. Atmospheric temperature increases enhance the longwave radiation. However, the humidity has a reverse relationship with the temperature change (Eq. 14). As a result, a good amount of temperature increase is cancelled with the response of vapour pressure. We have expanded the discussion on this.

Ln 517 – It is essential that incoming longwave vary with humidity for an assessment of sensitivity to be robust. By using measured and parameterized data this becomes ambiguous and parameterized data should be used exclusively. Agreed, we now use parameterized longwave radiation as the default in the model, and we only use mea-sured LW fluxes when we wish to control for this.

Ln 517 – You have the opportunity to include the effects of humidity on incoming short-wave radiation (through equation 9). As you note, this can overwhelm influence on incoming longwave radiation (Ln 319). The inclusion of this effect would be novel ap-plication of the empirical model. Agreed again, we had explored this in the theoretical sensitivity but not in the empirical model. It is now included as the default treatment: atmospheric clearness (tau) changes with the humidity.

Ln 524 – Your results indicate the feedbacks are important (Ln 541) and your conclu-sions should echo this more strongly. We had thought that we had emphasized this in the conclusions, but will state this more clearly and strongly.

Ln 560 – This assumption is likely to be incorrect and the effect of subsurface heat

fluxes needs to be considered (e.g. Pellicciotti et al. (2009). Now rectified through a complete year-round subsurface model, see above. In fact, QC is minor in the summer months here, on average, but the surface temperature does drop below 0°C frequently, particularly in May and September. This is now captured.

Ln 574-576 – Further explanation of this method is needed i.e. how did you treat variations in moisture - as changes in qv or in RH? If the former, then perhaps you will overestimate the actual variation as qv variations at lower altitudes will be larger. This is an interesting point, we had not thought of that. We do use the specific humidity from NARR, which originates from a grid cell with an elevation of 2216 m. This is about 450 m below the glacier AWS, so it is not terrible, but there will potentially be larger variations in qv, incoming LW, etc., from this altitude effect. Perhaps even larger an effect will be the temperature variability in summer months over a non-glacierized surface, which can warm up above 0°C. We will add a brief discussion of these sources of uncertainty.

Ln 586 – This statement seems to contradict the previous statement that most important radiative inputs are not well correlated on an inter-annual basis and that the variance of the shortwave does not correspond with the in-situ. As there is distinct seasonal variations in air temperature and solar radiation, these variables are heavily auto-correlated and a more meaningful correlation would remove the seasonal trend before correlating variables between NARR and in-situ data. We no longer discuss this.

Ln 629 – As biases in NARR results only happen to cancel and thus produce correct estimates of melt energy, these results cannot be considered robust enough to provide a meaningful interpretation of the inter-annual variations in the surface energy fluxes. Either the interpretations need to be carefully explained in this light, or further work is needed to demonstrate acceptable model skill. We have changed the focus and presentation of the NARR results, and believe that the new discussion is more relevant to the manuscript and grounded on these points. Because of the large biases in NARR

and the questionable skill in the annual energy balance and melt reconstructions, vs. the observations, we no longer present the NARR-driven simulations as mass balance reconstructions. We actually think this may be possible, through more work to assess model skill, but here we restrict the analysis to the covariance of NARR-driven net energy fluxes (summer melt) and different meteorological variables. Our aim is to see how the theoretical and empirical sensitivities hold up when multiple variables are perturbed at once, in a meteorologically consistent way. The means and variances of the NARR-based energy fluxes (Table 5) are close enough to the observed values to permit this comparison, with the important exception of the shortwave radiation. This is discussed.

Ln 654 – The approach presented in this paper has already been fairly well established in the literature (see comment for Ln 75) and so some additional novelty needs to be displayed here. We have rewritten to try and better address what is new in our approach.

Ln 763-765 – Further explanation of the differences between theoretical and empirical sensitivities is needed. Agreed, we have added this to the discussion, as well as the NARR-derived sensitivities.

Ln 770-771 – The trends in energy fluxes need to be more closely tied into the results of the sensitivity study. This discussion now removed, cf. Ln 629.

Many thanks for the detailed and thought-provoking review. Whether the revised manuscript is acceptable or not, it is certainly improved and our work going forward has benefitted from many of these ideas.

References Gerbaux, M., Genthon, C., Etchevers, P., Vincent, C., and Dedieu, J. P.: Surface mass balance of glaciers in the French Alps: distributed modeling and sensitivity to climate change, Journal of Glaciology, 51, 561-572, 2005.

Greuell, W. and Smeets, C. J. P. P.: Variations with elevation in the surface energy balance on the Pasterze (Austria), Journal of Geophysical Research, 106, 31717-31727, 2001. Klok, E. J. and Oerlemans, J.: Modelled climate sensitivity of the mass balance of Morteratschgletscher and its dependence on albedo parameterization, International Journal of Climatology, 24, 231-245, 2004.

Mölg, T., Cullen, N. J., Hardy, D. R., Kaser, G., and Klok, L.: Mass balance of a slope glacier on Kilimanjaro and its sensitivity to climate, International Journal of Climatology, 28, 881-892, 2008.

Oerlemans, J.: Extracting a climate signal from 169 glacier records, Science, 308, 675-677, 2005.

Oerlemans, J. and Fortuin, J. P. F.: Sensitivity of glaciers and small ice caps to greenhouse warming, Science (New York, N.Y.), 258, 115-117, 1992.

Pellicciotti, F., Carenzo, M., Helbing, J., Rimkus, S., and Burlando, P.: On the role of subsurface heat conduction in glacier energy-balance modelling, Annals of Glaciology, 50, 16-24, 2009. Sicart, J. E., Hock, R., and Six, D.: Glacier melt, air temperature, and energy balance in different climates: The Bolivian Tropics, the French Alps, and northern Sweden, Journal of Geophysical Research, 113, 2008.

Please also note the supplement to this comment:
http://www.the-cryosphere-discuss.net/tc-2016-6/tc-2016-6-AC1-supplement.pdf

---

## Author Comment (AC2) · 26 Jun 2016

Interactive comment on "Surface Energy Balance Sensitivity to Meteorological Variability on Haig Glacier, Canadian Rocky Mountains" by S. Ebrahimi and S. J. Marshall

Anonymous Referee #2

General comments

In this paper, theoretical considerations as well as an energy balance model are employed to assess the surface energy balance sensitivity to variations in meteorological

variables. The methods are applied at an automatic weather station (AWS) site on a mid-latitude glacier in the Canadian Rocky Mountains. In addition to the in situ AWS observations over the period 2002–2012, meteorological data from a reanalysis product (1979–2014) are used to force the model. Only the main melt season (June-August or May-September) is considered. The paper reads well and is written in good English. However, the methods used are not always described in enough detail, in particular regarding the energy balance model. Some model elements are not introduced at all, others are mentioned at a too late point in the manuscript. See the specific comments below for an overview.

Our apologies for the poor presentation of methods. We have completely rewritten this and moved it up front, to Section 2. It has added some length to the manuscript, but we hope that our approach and assumptions are now clear.

Apart from model parts not being described, I do not think the model and approach used are suitable for the sensitivity analysis performed in this paper. The surface energy balance contains important feedback mechanisms, which are pointed out by the authors at places in the manuscript. Although they account for albedo changes associated with increased surface melt, they do not seem to include the opposite effect of summer snowfalls on the albedo. More importantly, they do not calculate surface temperature internally in the model, while this variable is easily affected by changing atmospheric conditions. In its turn, it changes the outgoing longwave radiation and the turbulent fluxes. The authors do mention that surface temperature is generally at the melting point in the summer months, but not in the early and late melt season. Still, most of their results are presented for the entire melt season.

These are good points and we have a mixed response. We do (and did) have a parameterization of summer snow events (see Marshall, 2014), but we failed to explain it properly here. Summer snowfall is treated as a stochastic variable with a specified number of precipitation distributed over the summer. The amount of each summer snowfall is also random, between 1 and 10 mm w.e, and we use a temperature-dependent fractionation between rain and snow. This is now explained in Section 2. However, we did not have a subsurface/surface temperature model in the original submission. We have now added this to the model, and outgoing longwave radiation and the turbulent fluxes are calculated from the temperature in the upper (10-cm) surface layer. The subsurface model is run year-round at 10-minute to 1-hour time steps, solving temperature and with a simple treatment of meltwater drainage and refreezing in the upper 10 m. This is also described in Section 2.

The same applies to the theoretical derivations of the energy flux sensitivity, they also do not take changes in surface temperature into account. However, here the main results are presented for the months June-August only. This theoretical approach does present a simple method to estimate changes in the surface energy balance resulting from variability in the meteorological conditions. The results compare well to the model results, but not for all variables, suggesting some feedbacks are overlooked in the energy balance model. Whether this is a general method that can be transferred to other glaciers can only be established by similar applications on other glaciers with energy balance observations.

Here in the theoretical model it is not possible to account for changes in albedo or surface temperature through diurnal or seasonal cycles; it is meant to be a simple tool to provide a rough estimate of average summer energy balance sensitivities. We do restrict ourselves to JJA because of this important point about the assumption that the surface is at the melting point, which holds well in these months. We agree on the need to apply this approach to other glaciers and meteorological conditions to see if it is a useful way to assess glacier energy balance sensitivity. We do not add this to the current manuscript, but emphasize in the conclusions that is needed and potentially useful. If it does prove useful, we see that this framework as we lay it out, or variants on this, would be one of the main contributions of our paper.

Another major shortcoming of the energy balance model used is that incoming longwave radiation is taken from the measurements in the sensitivity analysis and not recalculated. As incoming longwave radiation is affected by both changes in air temperature and humidity (and cloudiness, here parameterized through relative humidity changes), the sensitivities are severely underestimated. Agreed, we now use parameterized longwave radiation as the default in the model, and we only use measured LW fluxes when we wish to control for this for comparison.

This is also revealed from the comparison with results from the theoretical approach. The model simulations with reanalysis input serve as an application of the 'perturbation' method presented. After reading the paper, I am still not sure what this method exactly is, but it is not as novel as the authors present it to be. I think the authors mean that the energy balance model is run with anomalies imposed on the 2002–2012 in situ conditions. But in fact, they are just forcing the model with a different set of (biascorrected) meteorological data. As the connection of this exercise to the sensitivity analysis presented before is rather weak, I doubt whether this is a valuable addition to the paper.

In revising the manuscript and rethinking this part of the work, we broadly agree with this criticism. Indeed, the reviewer understands exactly what we are doing, and it is really just forcing the model with bias-corrected meteorological data from NARR, a regional reanalysis. This is not new, although relatively few mountain glacier studies forced by climate model output use the full surface energy balance rather than PDD methods. But to the reviewer's point, we drifted off into historical energy and mass balance reconstructions, when this is not at all the point of the manuscript. We have rewritten and removed much of the NARR analysis, but keep some of this as a way to explore energy balance sensitivity to meteorological variations over a longer period, 36 years, vs. the 11-year observational record. More importantly, NARR meteorological variations are in combination (vs. one at a time), with a realistic level of interannual variability (vs. idealized sensitivity tests) and implicitly including meteorologically-consistent covariance of variables.

Thank you for these high-level comments: quite insightful and the manuscript is much

improved through consideration of these criticisms.

Specific comments

72-76: These lines give the impression that it is not very common to perform sensitivity studies of the surface energy balance on glaciers. The sensitivity to changes in temperature and precipitation is however assessed in numerous studies, therefore I suggest to change the word 'Several' to something more appropriate. Sensitivities to other variables are indeed less often investigated, but there are more examples than the one given here (e.g. Oerlemans (1991) and Gerbaux et al. (2005)).

This is a fair comment, we did not do justice to the literature on this. Our approach is a bit different, but this idea has been explored much more than we discussed. To our embarrassment, we were actually unaware of the Gerbaux et al. paper, completely missed it. This is now added and discussed in the context of a largely rewritten introduction. We also add a few other papers, including some of the work of Oerlemans and colleagues, in a brief review. Most emphasis to date has been on precipitation and temperature sensitivity, but this provides a useful context for consideration of the broader energy balance sensitivity – which in the end, points mostly back to temperature. We took out the word 'Several' as recommended as part of this rewrite.

105: I wonder what the net energy flux QN actually represents, the authors need to give a better description. If it is positive, it generates surface melt and is equivalent to what is often called the melt energy in other studies. But can it also be negative? The net energy as presented here seems to represent a residual flux, that should remain close to zero if the surface is not melting. Is this the case and is it set to zero then? Otherwise, it means that important processes in the energy balance are missing. Much later in the manuscript, on lines 471-473, I read that negative values are associated with refreezing. I do not think this can be assumed that simply, refreezing requires a snow/ice model which seems not to be included here.

We have not explained this well enough, and hope that it is now clear in the revised

manuscript. Net energy, QN, can be thought of as a residual, but really it is the energy surplus or deficit that attends the surface energy budget at any one time. If it is a surplus and the surface is at the melting point, then this is the melt energy. If snow/ice temperatures are sub-zero, this warms the system. But QN can also be negative, which drives either cooling or refreezing. The latter two processes are now modelled properly within the subsurface model. We explain this more explicitly in section 2 now.

171: This equation implies that h > 30 for all times, is this indeed the case? It would be neater to include a minimum condition, in case h < 30.

Good point, to be more general we now add this minimum condition as part of Eq. 9, to maintain physically bounded values in the parameterization. We did not record days with mean h below 30%, but they are of course possible in principle.

192-193: More detail is needed about the roughness length scales, as there are different ways to derive their values and treat/calculate them in the model. According to the cited Marshall (2014) paper, constant values were used for all three length scales with a predefined ratio between them (mentioned much later in line 326-327). Their values were obtained by closing the surface energy balance. This should be mentioned here as well. I also wonder whether values differ for snow and ice surfaces?

Revised and added in Section 2.

205: The paper does not mention how the subsurface conductive heat flux is calculated, is a vertical model used to keep track of snow/ice temperatures, densities and water content? Please add a few lines.

Now explained in Section 2 – a 10-m subsurface model is newly added.

212-214: I have the impression that QE is generally positive on mid-latitude glaciers during the melt season, or slightly negative. See for more examples the tables in Ohmura (2001) and Giesen et al. (2009).

This is mixed to our knowledge. Small positive latent heat fluxes have been reported

for sites in the Alps (Greuell and Smeets, 2001; Klok and Oerlemans, 2002). Giesen et al (2009) looked at the surface energy balance over two glaciers (60 &61 N) in Norway that are more maritime influenced, and had more strongly positive latent heat fluxes (9 and 16 W/m2). However, the results for latent heat flux in Table 2 in Ohmura (2001) indicate negative latent heat fluxes for a number of studies, and this generally reflects more dry, continental climates such as the Rockies. In our study, mean daily QE is negative 77% of the times from May to Sept and 66% of the time for JJA, and weakly negative overall.

231-233: Which percentage of the data needed to be gap-filled? Do you mean that factors are derived for months when data from both AWSs were available?

We have added two sentences to discuss the percentage of missing data, and clarified how we assign the factors. They are based on monthly values for all available joint data over the 11-year period. For most AWS variables, such as temperature, data coverage is 63% annually for the period 2002-2012 (2519 of 4018 days). 90% for the core summer months, JJA (909 of 1012 days), and 86% for MJJAS (1441 of 1683 days). The longwave radiation sensor was installed in July 2003 so there is more missing data. Coverage is as follows: annual - 46% (1835/4016 days); JJA – 76% (773/1023 days); MJJAS – 70% (1184/1683 days).

237-239: Is the mean daily value taken from the same day in other years?

Yes, exactly. So for May 10, for instance, we have 10 values from 2002-2012, but one year is missing at both weather stations; so we use the 10-year mean value from the glacier AWS to gap-fill here.

239-244: I am puzzled why the authors chose to use daily input data with imposed daily cycles instead of running the model at the resolution of the observations. There is a slight gain in efficiency, but at the cost of loosing important information to calculate the surface energy balance fluxes. Especially for the sensitivity of the surface energy balance, it is important to have enough detail. Perhaps the climate model output has

a lower time resolution, but then a daily cycle can be imposed there. In any case, it would be better to provide details about the daily cycle here, where the first questions arise and not at the later point in the paper.

This is valid as well, of course, and we do our reference observationally-driven runs forced by the actual 30-minute AWS data. However, as we discuss now in the manuscript, we moved to daily forcing (with parameterized diurnal cycles for the temperature and shortwave radiation) for two main reasons: a) climate model and reanalysis forcing is typically only available 8x, 4x, or once daily, and we are looking ahead to a surface energy balance model that can be driven in this way, and b) some of our parameterizations and gap-filling strategies are better suited to daily values, such as our parameterization of incoming longwave radiation in Eq. (7). We looked at 30-minute parameterizations of this form, but they do not perform as well. This is a central part of our sensitivity tests, to approximate humidity feedbacks on incoming longwave radiation, so we are pushed towards daily inputs. That said, we do lose efficiency despite this, by running at 10-minute time steps for the subsurface temperature model (which takes QN as an upper boundary condition); so lacking efficiency in any case, this model could certainly be adapted for finer meteorological input forcing, if available.

324-325: Air density is also assumed not to vary with temperature changes. Instead of using 'independent of temperature', which is of course not true, it might be more accurate to use 'can be assumed constant for small temperature changes'.

Good point, revised as suggested.

382-389: With unit forcings, as is done here, the sensitivities to changes in the different variables cannot really be compared. Better compare the effects on the different energy fluxes only per variable and leave the comparison between variables for the standard deviation-based forcings later in the section.

Agreed, this paragraph was to highlight the same point you comment on. Our purpose here was to explain the conditions when the unit forcings are (and are not) likely on the

glacier, in order to explain why we performed the 1$\sigma$ of all perturbation in the end.

404: This subsection title is not well chosen, since the sensitivity of all energy balance fluxes to changes in meteorological variables is considered in this section. Net solar radiation is an energy flux itself and it is not the variable that is changed in this subsection. Instead, the effect of changes in top-of-the-atmosphere insolation, atmospheric transmissivity and surface albedo on the energy balance are the subject of this subsection. Please change the title accordingly.

Good point, well caught. We changed the subtitle to "Change in Net Shortwave Radiation"

471-473: As mentioned before, to get a good estimate of the amount of refreezing meltwater and the associated heat release, a vertical snow/ice model is needed. Here, refreezing occurs whenever air (not surface?) temperature is negative, regardless of the amount of available water. If the period before has been cold as well, there will not be any water present. Even if water is present following a melt event, there may not be enough to release the amount of heat following from Eq. (2). I therefore think this is not a good way to compute refreezing and would either neglect it altogether or use a proper subsurface model.

Agreed. A subsurface model has been added, including a (simple) model of meltwater drainage. It does not influence things much in JJA, but does improve our shoulder-season results especially in May where there are cooler temperatures and an effective snow aquifer.

476-478: I do not understand how sensitivity analysis can be done if measured longwave radiation is used. Incoming longwave radiation needs to be adapted for different temperature and humidity. If the authors first show (in a figure) that using Eqs. (6) and (7) gives good correspondence with measured incoming longwave radiation, then they can use these equations with new temperature and/or humidity. This would largely reduce the difference in sensitivities to temperature and humidity changes obtained from

the theoretical approach and the energy balance model. Outgoing longwave radiation should also be allowed to change, unless the surface temperature is always at the melting point. But for negative air temperature anomalies, the surface temperature will often be lower as well.

The longwave fluxes are now allowed to change, based on the humidity anomalies and longwave parameterization (LW in) and the modelled surface temperature (LW out). Also, incoming shortwave is modified based on the clearness index parameterization. We still do some simulations without these feedbacks, to control for the magnitude of different fluxes and feedbacks, but our 'default' model allows these to change, in accord with these suggestions.

481-483: Many questions arose here, concerning the implementation of the changes in the energy balance model. Changes in air temperature will affect the fraction of precipitation falling as snow/rain, is this included in the model? Is snow depth tracked in the model to determine changes in the moments of ice (dis)appearance? How is albedo treated if the ice appears earlier than in the observations, is an ice albedo prescribed then?

The revised section 2 on methodology hopefully answers these questions. In short: changes in temperature will affect the rain/snow fraction in summer, but not the winter accumulation – this is assigned based on observed May snow accumulation (winter mass balance) for each year. Snow depth is tracked, along with snow albedo evolution and a shift to ice albedo when the snowpack has been removed.

486-488: The authors should make clear here which part of the year is used in the analysis. They mention that anomalies are applied to the entire year. But nowhere, except in the title of Fig. 2, it is mentioned that the analysis is performed over the months May-Sep.

Hopefully clear now. The simulations are year round, and melt is permitted year-round, but 99% of the melt occurs in the months of May-Sept and ∼80% in JJA. For calculating

and reporting mean energy fluxes and melt, we mostly report JJA, but also note MJJAS in places – explicitly noted in these case.

538-542: The albedo feedback has a smaller effect for negative temperature perturbations. Is this because increases of snowfall events are not included?

This is now discussed. It is due to a 'saturation' condition where cooling causes the snowpack to survive the summer melt season, i.e. with no transition to darker ice. Further cooling has less impact. Warming influences cause the snow to melt away sooner and this effect does not 'saturate' in the same way.

557-560: How representative is the assumption of a melting surface, this can easily be judged from the measurements. In Table 2, I see that especially in May, outgoing longwave radiation is considerably lower than 315 W m−2. Can you give the fraction of the time with a melting surface to the total time?

We have been able to do away with this assumption now, probably the largest improvement in the model. Now the surface is free to drop below 0C, and this feeds into the calculations of LW out, QH and QE. Because this is interesting, we added a new Fig. 3a that shows the annual cycle of Ts (daily mean values for 11 years). Indeed, surface temperatures drop below 0 degC on most nights through the summer (97% of March nights, and 71% of the time in JJA). Mean monthly values for Ts during the melt season are: May -2.8, June -0.9, July -0.4, Aug -0.6, Sept -1.6. So yes – assuming the surface is always at the melting point is not valid. That said, our results have not changed much with the subsurface/Ts model, perhaps because the melt energy (positive QN) is generally during the day, when Ts=0.

616-621: Why include the shoulder months in the analysis if they are not represented well in the model? Although it would still be better if the processes themselves would be included in the energy balance model.

We think including the shoulder months could give a better representation of the

model's efficiency for different meteorological conditions. Also, since we are introducing this method to be used for different locations, the importance of these months can vary. Therefore, we report both JJA and MJJAS results. That said, it is more valid to report this now that Ts is being modelled. Our attempt to include the shoulder season in the initial model helped to illuminate the need for this subsurface model. And we would still say that our September results are not as good as we would like, and point to the need for a better model of the 'end of summer/start of winter' transition, heralded by the arrival of snow that persists. We discuss this as a place to invest in for future efforts.

694-695: Summer snow events also bring additional mass to the glacier, further reducing the net melt.

Yes, now commented on; it was included but not discussed.

757-758: Please be more specific about which feedbacks are actually included. Only the internally modelled snow-aging is described in lines 529-536, it is still not clear to me to what extent and how the snow/ice transition and snowfall events are included.

These are hopefully clear now in the expanded Section 2.

Table 6: In general, I think the manuscript contains a relatively large number of tables and a small number of figures. Especially this table contains too much information to serve a purpose and it also needs to be compared to another table. Please make the comparison easier, by visualizing the monthly energy balance fluxes for the in situ data, the NARR perturbed data (and optionally the NARR raw data) together in a figure.

This Table has been removed, along with this content. Our Figure/Table ratio is higher now, and NARR analyses are mostly just in Figures.

Figure 2: As longwave radiation is (not yet) allowed to change, the effect of net radiation corresponds to the effect of net shortwave radiation alone. Better present it this way and add a line for net longwave radiation, when it is also varied. I would also like to

see a line for the summed effect on QN, which is especially illustrative for the opposite effects found for wind speed changes.

We now have lines for longwave and net radiation, also QN.

Figure 6: Why is albedo shown for JJA instead of MJJAS, as the other variables? I would like to see the net shortwave and net longwave radiation separately instead of net radiation, as these are treated individually throughout the manuscript. I do not think it is necessary to show both net energy and melt, because they are directly related.

This Figure has been removed.

Technical corrections 40: I would not consider the word 'banal' fit for scientific papers, please rephrase. Revised and removed.

55: 'reanalyses' Revised.

60: 'for snow and ice melt factors' Revised.

69: 'crucial to ablation on' Revised.

116: 'solar radiation that is reflected' Revised.

123: _0 is used in Equation (3) instead of 0 Revised.

150: As Kwadacha Glacier is not the subject of this paper, better rewrite as 'At two study sites' Revised.

158: 'ratio of potential direct to measured' (or is measured radiation only direct radiation as well?) The measured radiation in our study is the combined direct and diffused solar radiation.

159: Include a reference here, is it the paper mentioned in the next line? The reference, Ebrahimi and Marshall (2015), has been added now.

186-187: Split into two sentences: ' and q ... humidity. Measurements... levels, at the surface-air... and at height ... surface.' Revised.

189-190: Reorder: 'We estimate Ts from an inversion of Eq. (5), using' Revised.

193: 'can be' Revised.

229: 'meteorological conditions' This section is rewritten.

264: 'Warm summers generally cause' Revised.

265: 'but the energy balance is sensitive to' Revised.

286-287: 'of the response to a temperature change' Revised.

305: Remove the spaces in 100 (1 00)? Revised.

332: Include the dot on m as in Eq. (2) Revised.

340: 'at the AWS site' Revised.

392-393: Split into two sentences: 'Following Eq. (9),' Revised.

407: I wondered what was meant by 'solar variability' and found the answer in line 424-425, better move it here. Revised and rewritten.

415: Is QS0 equivalent to Q0 introduced in Eq. (3)? If yes, use the same notation, if no, clarify the difference. Revised.

445: 'last two lines' Revised.

495-496: Mention that results for simultaneous changes in temperature and humidity are not shown here. Revised.

497-500: These lines belong in the figure caption, not in the main text. Revised and rewritten.

500: 'Sensitivity to albedo changes over' n/a as this section is rewritten.

507: 'directly' This section is rewritten.

510: 'The sensitivities computed with/resulting from the surface...' Revised.

512-514: These may be advantages, but are these effects included in the model used here?

In rewriting we have tried to be more clear on what is and is not included; in general, there are advantages to the model in that it permits most feedbacks to be included, but they can also be 'turned off' to isolate and understand different influences.

528: 'induce' Revised.

558: What is the 'summer melt season'? May-Sep or Jul-Aug?

Both are examined, but in revisions we have tried to clarify to which we refer. MJJAS is the melt season, really, but JJA the core melt months.

609-610: The wording should make clear that these energy fluxes are not taken from the NARR reanalysis, but calculated with the energy balance model using NARR meteorological forcing. Further down, 'NARR-based' is used frequently, this is already better.

Revised thoroughly, throughout this section.

661: 'changes in most meteorological variables' Revised.

668: 'Increases' Revised.

669: 'through the sensible and latent heat and incoming' Revised.

692-693: 'fraction of time with surface temperatures at the melting point' Revised.

698: 'as in the simple experiments presented in this paper' Revised.

699: What is meant with 'everything', please be more specific here. Revised. As the previous sentence starts with 'meteorological variables' we now revised the second sentence by referring to these variables.

726: 'balance' Revised.

747: 'allows for a' Revised.

771: Just write 'Net solar radiation', as longwave radiation is not allowed to change. n/a now, as we have removed discussion of trends.

Table 1: Write out the definition of 'summer melt season' in the caption. Use SI units for air pressure (Pa or hPa)

The months and summer are now clear, and units are changed from mbar to hPa.

Table 2: Caption: 'Mean monthly surface energy balance components/fluxes and monthly melt totals.' All details about the location can be left out, this can be read in the text and is also included in the caption of Table 1. Can you use symbol notation for melt as well, being the sum of the melt rate? The caption of Table 2 has been revised.

Table 3: Note that all sensitivities are calculated using the JJA mean values, now this is only stated for $\delta$QN. Furthermore, in the table on line 998, there is no apparent change with regard to the previous line. However, $\delta$h is not zero here, which should be mentioned. On line 1003, it is not QS (a variable that has not even been introduced) that is varied, but Q0. The caption is revised to indicate that the results are for the summer mean, and the descriptions have been clarified wrt the perturbations. Qs changed to Q0, thanks for catching this.

Table 6: 'NARR-based mean monthly...' The table's caption is revised (Table 5 now).

Figure 1: Either note that KG indicates Kwadacha Glacier, which is mentioned once in the paper or remove the dot and zoom in on the map around Haig Glacier. I suggest to do the latter. We will zoom in, as suggested (now Figure 2).

Figure 2: Remove the figure title above the panels and add the melt season period to the caption. Include a legend to indicate the different fluxes and remove from the caption, this makes the figure and caption easier to read. Showing albedo changes as absolute or relative (%) values is not exactly the same, if you like to use the same

scale as for shortwave radiation, then just say 10 x albedo change. Since the x-axis label also only mentions the shortwave perturbation, it may be a better solution to use the upper x-axis to indicate the albedo scale and title. Furthermore, 'SW' is now used for shortwave radiation instead of S, please be consistent with notation throughout the manuscript.

Figure revised as per the suggestion.

Figure 3: Please use the same variables and colours as in Figure 2. Revised, now consistent.

Figure 4: 'Table 5 gives the bias and correlations.' This figure has been removed.

Figure 5: More tick marks are needed on the x-axis, at least for every five years.

This figure has been removed.

Many thanks for this unusually careful and detailed review. Lots of insights and valuable suggestions. Much appreciated.

References Gerbaux, M., Genthon, C., Etchevers, P., Vincent, C., Dedieu, J. P.: Surface mass balance of glaciers in the French Alps: distributed modeling and sensitivity to climate change. Journal of Glaciology, 2005, 51, 561-572, doi:10.3189/172756505781829133

Giesen, R. H., L. M. Andreassen, M. R. van den Broeke en J. Oerlemans: Comparison of the meteorology and surface energy balance on Storbreen and Midtdalsbreen, two glaciers in southern Norway. The Cryosphere, 2009, 3, 57-74, doi: 10.5194/tc-3-57-2009.

J. Oerlemans: The mass balance of the Greenland ice sheet: sensitivity to climate change as revealed by energy-balance modelling. The Holocene, 1991, 1, 40-48, doi:10.1177/095968369100100106

Ohmura: Physical basis for the temperature-based melt-index method. J. Appl. Me-

teor., 2001, 40, 753–761, doi:10.1175/1520-0450(2001)040<0753:PBFTTB>2.0.CO;2
Please also note the supplement to this comment:
http://www.the-cryosphere-discuss.net/tc-2016-6/tc-2016-6-AC2-supplement.pdf

Interactive
comment

---

## Author Response (AR2)

**Report #1**

**Suggestions for revision or reasons for rejection (will be published if the paper is accepted for final publication)**

General comments

First of all, I am glad to see the authors have put great effort into improving the (methods behind the) manuscript, partly based on my suggestions. My main point of concern was that their energy balance model did not allow for changes in incoming longwave radiation, surface temperature and albedo, with associated feedbacks. The authors have now extended their model to include all these processes, which makes the results more consistent and more convincing. Furthermore, they have clarified the model description and revised the section based on the NARR data, such that it is in line with the rest of the manuscript.

Due to these changes, the methods behind the manuscript have improved significantly. However, I still have some comments on the manuscript, as outlined below.

The most important is that the revised manuscript is considerably longer than the first version and I do not think this is an improvement in all places. For the model description, it was indeed necessary to include more detail. The Introduction has also increased in length. Although it reads smoothly, it is very long for an introduction and I suggest to make it more concise. The Discussion section has been rewritten completely. The present form reads like an evaluation of the results from the three different experiments grouped per perturbed variable. In my opinion, it is primarily a repetition of previously presented results with little discussion added. The last two paragraphs deal with suggested model improvements (without new subtitle...) and are more at place in a discussion section. On the other hand, some results are (also) discussed in the Conclusions section, which is also rather lengthy. I would suggest the authors to look critically at the Discussion and Conclusions sections and rewrite them. They should make sure that repetition is kept to a minimum and that no new discussion items are introduced in the Conclusions. Furthermore, they could perhaps address the representativity of their results for other parts of the glacier and shortly discuss the applicability of their methods on other glaciers.

Thanks to the reviewer for these suggestions. We rewrote much of the manuscript for the first re-submission, and it is true that parts of the discussion and conclusion were redundant. We have restructured the discussion and conclusion following the suggestions of both reviewers, and it is now shorter. Most of the 'summary' content in the discussion and conclusions that was covered in the results has been removed. The new content about insolation in the conclusions has been removed, and we bring in a little bit about glacier-wide applications in the discussion (ll. 970-985). The introduction has also been shortened by about 10%, but remains longer than the original submission in the interest of giving proper attention to some previous work on energy balance sensitivity studies.

The effect of the snowpack depth at the beginning of the melt season is now also investigated, which is definitely interesting. However, I am not convinced by the method used. It is not clear to me how the experiment is conducted, it seems like the model is only run for one year with averaged daily meteorological values (also mentioned below). The new figure (Fig. 7) shows small effects of initial snowpack depth for depths above 1.2 m w.e., but the values fluctuate around zero. I do not understand where this variability comes from, it seems random. Is it because snowfall events are prescribed randomly and would it not be better to keep the timing of snowfall events the same for all runs? Furthermore, I am a bit surprised by the low albedo prescribed for snow remaining all summer. It is only slightly larger than the ice albedo; whether the surface consists of snow or ice after day 220 makes little difference for the energy balance. Is this realistic?

This is useful feedback, and we are certainly open to ideas here. The reviewer is correct – we were running for only one year, using average daily weather conditions for the period 2002-2012. Our wish is to isolate the effects of the initial snowpack, so this seems like a sensible way to do this: repeat the same weather but for different initial (May 1) snow conditions. But it is true the averaging gives a weather time series that is not real. We now do this differently, running for the 11 years of actual weather but over the suite of initial snowpacks. In the end it does not change much, as we are averaging the final result for presentation.

Yes, the random element in these graphs comes from the summer snowfall events – they are an internal part of the code and we left this on through these experiments. This could be specified to control for this, to make the graph cleaner, although the process is separate from and independent of the winter snow extent so it is not systematically interfering with the experiment. It just gives some variability between realizations. We left this on, but comment on it and note that the summer snow gives 'internal variability' of about $Q_N = 1$ W/m$^2$, averaged over the summer (control experiments, not shown).

The final point, concerning the minimum snow albedo, is insightful and the reviewer is quite right that this 'old snow' value (0.3 for us) is very low and explains why the difference between exposed ice and old snow did not matter very much in the previous submission. Our number is based on observations from Haig Glacier firn, and our default treatment was to set the minimum snow albedo to that of firn. But this is may not be appropriate – firn on this glacier has an accumulation of impurities, similar to what occurs in ablation zones, so it is darker than old seasonal snow. Values of wet, impurity-rich, late-July snow at the AWS site on Haig are about 0.36. This may still be darker than values higher up in the glacier accumulation area, or at other glacier sites. To be a bit more 'typical' we set a new default value of 0.4 for the minimum snow albedo. All experiments in the paper now use this value. Figure 7c is new, showing the net energy sensitivity to winter mass balance for both $\alpha_{min} = 0.3$ and 0.4. Results are indeed sensitive to this choice for the late-summer energy balance. As expected, higher values of $\alpha_{min}$ give a stronger influence of a deep winter snowpack, although the graph is still asymmetric – a shallow winter snowpack leads to large increases in summer melt, while a deep winter snowpack moderately reduces melt at this site. This result depends on how close one is to the ELA – lower on the glacier, the result would be reversed. This is discussed in the text.

Regarding the figures, I would strongly recommend the authors to add legends explaining the different lines/symbols. Every figure shows new variables, many with multiple lines. Determining the meaning of all lines from the (sometimes incorrect, see below) captions is complicated and unnecessary. Related to this, I would suggest to show a smaller variation of fluxes (with standard colours) to make the figures more consistent. For example, the turbulent fluxes are sometimes shown separately, sometimes combined and then in another figure combined with the subsurface heat flux.

This is a good suggestion – there is not always room to add legends, so we put this information in the captions and tried to stay consistent, but agreed that we have too many lines and colours, and it changes every plot. We now have legends for each figure, where applicable. We also simplified a bit, e.g. removed $Q_C$ in Figure 3, and radiation and turbulent fluxes are now combined in Fig 6 to reduce the amount of information.

Where the figures are discussed in the text, the authors sometimes refer to the specific line colour in the text. This should be avoided, it should be easy to derive from the figures, by means of a legend. In general, the authors may try to refer less directly to the figures and tables, by only adding references in brackets and not in the main text.

Revised as suggested throughout the text.

Some detailed comments (and elsewhere):   The authors now investigate the effect of changes in winter snowpack depth. However, they refer to this variable as 'winter snowpack', while they should add a measurable quantity like 'depth' or 'thickness'.

Revised as suggested, l.16 and in Section 4 (discussion of Figure 7).

127-128:   Positive net energy will not drive subsurface warming, as this has already been taken care of by Q_C.

Revised, l.112

133-134:   If the unit is given for Eq. (3), the unit for Eq. (2) should also be given (W m-2), as one follows from the other.

Added, l.105

245:   'phi_t(z)'

Revised, l.231

249-260: How is the refreezing rate calculated?

We added a brief explanation on this, ll.234-238. In essence through an enthalpy model. If liquid water is present in the pore space and conductive cooling gives an energy deficit, the available

'negative energy' is diverted to latent heat of freezing; temperature are not allowed to cool in a layer until liquid water content θw = 0.

293: In what sense is the grid fixed, with respect to the surface or a reference layer? How are changes in snow depth and ice surface lowering incorporated? Are layers added/removed or is the grid shifted?

Good questions, we briefly explain this now, l.282 and ll.293-296. It is a fixed grid with respect to the surface, to a depth of 10 m (irregularly spaced, with nz=33). Near-surface layers are 10 cm thick. Snow depth $d$ is modelled in a sort of Lagrangian sense, to the mm, so it is allowed to continuously accumulate, melt, or undergo densification (on a daily time step). Then at depth $d$ below the surface, the grid cell has a weighted combination of thermal properties and densities to reflect the mixture of snow and either firn or ice in that layer. There is also a discrete step involved: every time 10 cm of snow accrues or ablates, the grid is shifted to propagate up/down the internal density/liquid water/ice layer structure. We hope this makes sense – happy to explain this further but we don't want to go sideways in this paper on the details of the subsurface model. It probably needs to be described elsewhere in proper detail.

340-341: The reference to Eq. (11) is not correct.

Revised, l.330

355-356: Why is no aging included for summer snow events like for the seasonal snow pack?

It is, the clock starts again and albedo will decay. But this does not happen much as summer snows usually melt within 1-2 days. Clarified, l.346

360: I am a bit confused that internal melting can occur in this model. The main source would be penetration of solar radiation, but this is not included here. Where does the melt energy come from and is it a large term?

Quite right, there should be no internal melting since we don't account for shortwave radiation penetration or meltwater/rainwater temperatures above 0°C. It is built into the code as an option (when we get to some attempt to include these processes), but is a 'latent' option right now. This statement has been removed, l.353.

442-449: The main reason to use JJA for the theoretical sensitivities is that the surface is at the melting point, as a good approximation. This is not mentioned here. Please also replace 'here' by 'in this section' or 'to calculate the theoretical sensitivity' to stress the contrast with 'the next section' with the 'modelled sensitivity', where the full melt season is considered.

True of course, now noted, l.433. Clarification on ll.444-447 for the second point.

(and Table 4): Not MJJAS melt energy as mentioned before?

No, we are trying our best to make it shorter and more focused where we can – so as of the first re-submission we now report only JJA, although we run the model year-round for the 11-year period (including May and Sept melt). The sensitivities in MJJAS are not so different from JJA, so we are sticking to that to increase the focus.

758-759: How is the energy balance model forced with this mean annual record? Do I understand it correctly that this record spans one year and has the mean value over the entire period for each day and each variable? This seems rather artificial to me, it does not represent real meteorological situations anymore. Why not run over the entire period using the same winter snow pack depth for each year per run?

Discussed above. It is true, averaging makes for an unrealistic time series in lots of ways, not too hot and not too cold. This can certainly be done as suggested, and it is consistent with the other perturbations (i.e. 11 realizations for each meteorological anomaly). We reworked this section thoroughly, starting at l.710.

789-793: Why are standard deviations not compared over the same period (2002-2012)?

That is a fair point, we are interested in the variability over the full NARR period, 1979-2014, but it is probably not appropriate to compare those numbers to the 11 years of observation. We revised this to report the NARR variances over the common period, 2002-2012, ll.761-764. Values did not change much.

1021: Refreezing of melt water acts as an energy source (not sink) through release of latent heat.

Thanks yes, this was loose language. We were thinking of the energy sink as the positive net energy that is required to re-melt refrozen meltwater. At night when there is sometimes refreezing, the energy that is released is often dissipated (e.g., as QC to the surface, LW emissions, etc). Then the next day new energy is required to melt some 'recycled' meltwater (the overnight ice crust). This text has been removed as part of the discussion rewrite.

759-760: The range is 0.35-2.35 in the caption of Fig. 7.

Revised, we ran in the end for 0.36-2.36 (the observed mean, 1.36, \pm 1)

Fig. 2b: As also suggested before: Either note that KG indicates Kwadacha Glacier, which is mentioned once in the paper or remove the dot and zoom in on the map around Haig Glacier. I suggest to do the latter.

Revised as suggested

Fig. 3: Better show net shortwave and net longwave radiation instead of incoming shortwave radiation and net radiation, then all fluxes are shown exactly once. I would also suggest to use a long horizontal axis for both plots and show them above/below each other instead of next to each other. Then the interannual variation is more clearly visible and corresponding days can be compared.

These are good suggestions, adopted. QC is also removed.

Fig. 6: Net radiation is mentioned twice in the caption.

The black line is net energy, thank you for noticing, it is revised now.

Figs. 9 and 10: Numbering of the figures is incorrect, the numbers are still from the previous manuscript.

Apologies, amended. These figures were not meant to be included – we removed this part of the analysis and discussion in the first revision, but the figures lingered.

Fig. 10: Why are the turbulent fluxes both underestimated with the NARR forcing? The meteorological variables show good correspondence...

These figures and the associated results are no longer discussed, as we chose to focus just on the sensitivities. But for the reviewer's interest, this problem went away with the revised code – it was because we were assuming a melting surface in the initial submission. Now that $T_s$ is internally modelled with the subsurface temperatures, the modelled turbulent fluxes are much improved. But no longer discussed….

**Report #2**

**Referee review for manuscript tc-2016-06-manuscript-version-3, submitted for publication in The Cryosphere**

"Surface Energy Balance Sensitivity to Meteorological Variability on Haig Glacier, Canadian Rocky Mountains" by S. Ebrahimi and S. J. Marshall

General Comments

The revised manuscript is an improvement on the initial submission. In particular, the model now includes more dynamic feedbacks which has increased the confidence with which the results can be interpreted. The re-setting of the NARR results as an extension of the sensitivity analysis is good to see and improves the focus of the paper. The authors appear to have considered and addressed the main points in their response to my first review. However, it was difficult to assess the actual changes made to address each point as no text changes were given in the response and a marked up version of the manuscript was not supplied.

Apologies, the overall was so large that we did not quote specific revisions or invoke 'track changes'. The first draft was almost completely written in accord with the reviewers' suggestions, so this did not seem productive. In our second round of revisions, we use track-changes mode and the specific changes can be seen – still very extensive in the rewritten discussion, but possible to examine specifically through the rest of the text.

Also, the paper still contains many ambiguities of method and much inference that isn't always supported by robust results. The main result appears to be an extremely large increase in temperature sensitivity when albedo is allowed to vary. While one would expect the sensitivity to increase, the magnitude of the increase here needs to be better supported by the validation of the albedo scheme against measurements and further discussion or analysis around the role of impurities to justify the low minimum value for snow albedo used. The choice of minimum values of albedo for snow and ice also impacts the conclusion that winter balance is of less importance, as with higher minimum albedo values the contrast between snow and ice is larger and this will increase the sensitivity. The authors also need to comment on the processes driving this sensitivity - to what extent the sensitivity is driven by decreases in snow albedo, earlier transition to ice and summer snowfall.

The temperature sensitivity with albedo feedbacks is still high, doubling the response, but it is much reduced with the various model changes that went into the revisions. Now it is well in line with previous literature. We made numerous changes, but mostly we now introduce all parameterizations for consistency, in particular cloud feedbacks (via the clearness index), such that shortwave radiation decreases with increased humidity (more cloud), offsetting the increase in longwave radiation. This and the increase in minimum snow albedo to 0.4 have reduced the sensitivity. With the revised experiments, the only difference is temperature forcing with and without albedo feedbacks, i.e., the other temperature feedbacks such as length of the melt season, rain vs. snow, are the same in all temperature experiments (cases 1-4 in Table 4).

The use of parameterisation for the radiative fluxes that responds to various drivers is encouraging. There is still a need for a validation of the melt produced by the fully parameterised model against that driven with measurements if the results are to be trusted. It is also ambiguous which version of the model is being used for any particular run and this needs to be carefully detailed and explained.

We had not brought in a discussion of the energy balance and subsurface model performance and validation vs. observations or a 'reference' model, mostly because these models are based on Marshall (2014) and Ebrahimi and Marshall (2015), for the longwave parameterization, and the current paper is already long. We could spend some time on a model validation section, but fear that this would be a diversion. The reviewer is correct though, confirmation that the parameterized daily model gives a good representation of the summer energy balance and melt is needed in order to trust the results. We added two paragraphs on this (Section 3, ll.407-423).

The new text gives a summary of the parameterized model performance in a sample dataset from summer, 2015, for which the melt model was not tuned or calibrated. We observed/measured a total summer melt of $3.1 \pm 0.1$ m w.e. at the AWS site from May to September, 2015, based on a May snowpit (measured snow water equivalent), the AWS SR50 record, and ablation stake data. The energy balance and susbsurface snow temperature/ drainage model are run in two modes, forced by the 30-minute AWS data (best case or reference model) and with the parameterized model, that degrades the AWS data to daily forcing with parameterized albedo (Eq. 20), incoming longwave radiation (Eq. 8), and diurnal cycles of temperature and shortwave radiation from Eqs. (17) and (18). The reference model that is driven by 30-minute AWS data gives a total summer melt of 3.04 m w.e., and the parameterized daily model gives 2.98 m w.e., with a small under-prediction of melt due to over-estimated summer albedo values (figure R1). The RMS error in daily melt totals is 3% (0.7 mm, relative to a mean summer melt value of 23 mm w.e.). Daily melt predictions from each model are shown here:

[Figure]

Figure R1. Parameterized model performance (blue) vs. the reference model (black), that is driven by 30-minute AWS data. There can be large departures in the melt daily melt rates (lower right), mostly attributable to discrepancies between the measured and modelled albedo (top left). The model overestimates albedo in summer 2015 (May-August) and underestimates it in September, when new snow (an early start of winter) is not adequately captured. Note that the downward-looking shortwave radiometer was not properly wired in summer 2015 from May 12-June 6 (day 133-157), so albedo is estimated for this portion of the record. Bare-ice albedo at the AWS site was close to 0.1 in summer 2015, but is not always this dark.

--

We do not include this figure or other model calibration discussions in the main text, as we don't want to lengthen it or take a tangent, but we summarize these statistics on model performance to address the question of whether model skill is adequate for the sensitivity analyses that are our main focus. This is just one summer, but the model was tuned for the period 2002-2012 and the performance is similar here; it is not perfect, especially for the albedo, but we believe the parameterized model has reasonable skill and is a reliable tool for our present purpose.

The authors identify a series of significant feedbacks between air temperature, humidity and the incoming fluxes of short and longwave radiation. It would appear most of these are all connected through cloud cover, and it would be much clearer and insightful for the authors to explicitly examine changes in the frequency and attenuation of cloud cover alongside with air temperature and humidity, rather than inferring these from the sensitivity of incoming shortwave radiation in the analysis of the in-situ dataset. This should be possible as the authors already have a parameterisation of cloud cover that includes humidity. If not, then the variability in incoming longwave also needs to be included in Table 4.

This is a good point and suggestion. We have rerun everything for the revised submission, with a simplified (identical) set of model parameterizations in all runs and with consistency of the shortwave and longwave radiation perturbations, through the transmissivity \tau. This is effectively our cloud proxy, and is now described as such. So we perturb \tau (\pm 0.1) and describe this as changes in cloud cover, which have opposite influences on incoming LW and SW radiation. Table 4 includes this now. Also, all experiments with a change in humidity see a corresponding change in \tau, hence incoming SW and LW. This is now internally consistent.

Moreover, all sensitivity studies in section 5 now use the parameterized version of the code, in the daily energy balance/subsurface temperature model. That is to say, we do not use the raw 30-minute AWS data to force the model, but rather the parameterized diurnal cycle, albedo model, modelled surface temperature, parameterized incoming longwave radiation, and parameterized clearness index (shortwave transmissivity), \tau. This allows clear understanding of our experiments and comparability across results. All of our numbers in Table 4 are revised as a result of these changes. Most have not changed much, but the temperature sensitivity with feedbacks is now much less – now more in line with observations and the NARR results.

In general, the paper is fairly well written though some of the text in the results and discussion is quite methodological and repetitive and perhaps is better suited to a methods section. Further work is needed to distill the main results from a rather large body of work and succinctly present them here.

Agreed, our results and discussion were repetitive and we did not do a good job of distilling the main results. We have rewritten and shortened this. We did keep some of the methodological details within the results, e.g., the details concerning the NARR forcing in section 6. The paper covers a lot, and we initially had much of this material in section 2 (methods), but it was far removed from the eventual results or NARR experiments and made for difficult flow/reading. The same holds true for the partial derivatives/theoretical sensitivities in Section 4. They are tiresome but are relevant locally (section 4) and not in the other results' sections. Hence we choose to combine methods and results to some degree in Sections 4-6, where specific to that section. We did our best to clean up the discussion, however.

The authors have managed to refocus their analysis but the new results and novelty of their results are often quite hidden. I would suggest re framing the discussion around 1. A summary of the important sensitivities and feedbacks that are observed on the Haig, 2. A discussion of the utility of the theoretical sensitivity based on mean summer conditions (good correspondence with full summers when feedbacks are omitted) 3. A discussion of the utility of exploring sensitivity to inter-annual variability with reanalysis datasets (mixed results). This structure would alert the reader to what is new and avoid some of the repetition.

The discussion and conclusion have been rewritten, somewhat but not fully along the lines suggested. We hope that the main results are now more clearly presented.

Line comments

- replace 'profile method' with 'bulk aerodynamic method'. The profile method uses wind speeds and temperature at two heights.

Revised, l.187.

- the symbol psi has a different case from the equation.

Revised, l.227.

- please include an explanation of how the refreezing rate is calculated and what constraints are put on the volumetric water content.

Added as per request of both reviewers, ll.234-238 and ll.248-251. The upper constraint on the volumetric water content is the porosity of the snow or firn, but drainage in the seasonal snow is efficient enough that this is never approached; tracking $\theta_w$ evolution through the summer, it reaches a maximum of about 10%. There is a minimum (irreducible) water content in temperate snow, associated with capillary pressure, which we set to 3%.

- for consistency please state the model uses a variable timestep from 10 minutes to 1 hour to allow for stability of the subsurface temperature prognosis.

Revised as suggested, ll. 271-272.

- the minimum value for snow is very low. For the same site, Marshall (2014) gives 0.4 as a minimum. This will have a large impact on the sensitivity. Other authors have used values around 0.5 (Oerlemenas and Knap, 1998) and further justification of this low value is needed.

This is well observed, and R1 also questioned this. As discussed in the response to R1, this does of course impact the sensitivity; we now include experiments with both 0.3 and 0.4 as the minimum snow albedo (Figure 7), and have reverted to 0.4 as the default for the sensitivity studies (l.340). We initially used 0.3 because our observations at Haig Glacier indicate a value of 0.3 for firn, but old seasonal snow is closer to 0.4 at our site (see, e.g., the black line in Fig R1 above, just before the glacier ice is exposed). We sometimes see values below 0.4 for old snow at the Haig Glacier AWS site, in late summer, but 0.3 is probably too low to be representative of the accumulation area or the region, in general. It was our oversight to treat aged seasonal snow as firn; the latter has had more time to accumulate impurities.

- need to provide more discussion and evidence for the reasons for the decrease in albedo - i.e. is the increase in particulate concentrations documented?

The increase in particulate concentration is certainly observed, both in old snow and exposed firn and ice, but has never been quantified or supported by measurements. Empirically though, the seasonal albedo decrease at this site is strong, as has been documented elsewhere and as evident in e.g. Fig. R1 and Fig. 4a. Also see Fig. R2 below, for the average summer albedo evolution from 2002-2012. We assume that the seasonal decline is due to the water content, impurities, and grain growth in the temperate summer snowpack (after the conventional wisdom in Cuffey and Paterson and elsewhere), but it is fair to say that we are speculating as we don't have measurements or process studies to attribute the causes of the seasonal albedo decrease. We just observe a snow albedo decline from ~0.8 to ~0.4 each summer, before it drops to ~0.2 on the bare ice (Fig. R2). A detailed study of this may be warranted, but is out of scope here. For now, we modify the text (ll.394-398) to note the observations and empirical record but stop short of attributing cause.

[Figure]

Figure R2. Average summer albedo evolution at the AWS site, 2002-2012.

- the contribution to melt should really be computed for melting periods only, as non-melting periods will bias these fractions towards the sensible heat flux (see Conway and Cullen, 2016). Please either show the fluxes for melting periods only or discuss only the contribution to the energy balance and not to melt.

Point taken, we now refer just to the contribution to the energy balance, l.404.

423-430 - please make it clear that the feedbacks and NARR analyses are presented in the following two sections, rather than the current section.

Revised; the NARR discussion has been removed from here as it was evidently distracting.

and 439 - some more context is needed to justify conditions on the Haig being 'typical' of other mid-latitude glaciers. Please add either a table showing this or some references to papers with similar climatologies.

This is fair; we suspect but cannot substantiate that all of the weather conditions here are 'typical' of mid-latitude glaciers. For instance, the elevation (hence pressure and vapour pressure) are lower than for glacier in the Alps and higher than the ablation zone of coastal mid-latitude glaciers; winds at other sites are often stronger than here; etc., etc. It is difficult to argue that mean conditions here are typical, and that is a loose qualitative word. In the end we removed this line and a bit of text around it from the manuscript, near l.430.

- need further analysis in order to justify that the JJA has more impact on melt? Table 3 shows an almost identical combined sensitivity.

Yes, it is true in the results – this is why we report it, as we did not necessarily expect MJJAS sensitivity to be the same. But JJA conditions have more impact on melt simply because 80% of the melt occurs in these months. A cold May or Sept does not have the same impact on summer mass balance as a cold July. Net energy sensitivities may not change much, but the impact on melt does. In any case, this is now N/A as the MJJAS sensitivities are no longer reported, at the reviewer's request (see below).

442-448 - this is quite a confusing paragraph as the rationale for including/excluding months changes from the start to the end of the paragraph and results are introduced, but not referenced properly. Please either point to the figures/tables that justify these statements or move this text to the discussion.

Paragraph has been revised and simplified, ll.444-447, sorry for the confusion.

- Are these perturbations calculated as the average of positive and negative deviations from the mean? Some of the text (e.g. 518) seems to suggest that only positive perturbations were considered, which is not ideal.

This is a bit interesting. Yes, in Section 5, with anomalies introduced into the numerical model, perturbations are always introduced as positive and negative deviations from the mean. Here where we consider theoretical sensitivities, the values are based on derivatives at a point (the mean state), so sensitivities can be considered to apply only at this point, i.e. for infinitesimal negative or positive perturbations from the mean. The result is the same for either sign of perturbation, as it is essentially the slope at the point. If the relationship is nonlinear, it becomes invalid for large negative or positive perturbations.

- I am not sure at this stage it is appropriate to transfer the calculations of net energy to melt, as in reality not all periods will have melting conditions. Perhaps it is better to state the increase or decrease in the net energy available. Along with this I would remove the melt column from table 3.

We agree, of course, and transfer the net energy perturbation to melt with a cautionary note (ll.504-508), but have retained this in Table 3 and the discussion because melt rates give a more intuitive idea of the potential impact, or lack thereof. The values can also be compared across the different perturbations. Moreover, JJA mean $Q_N$ translates well to JJA melt at the site; the reference value is 97 W/m$^2$, which gives 2.30 m w.e. melt if this is converted directly to melting. The reference JJA melt is 2.32 m w.e. (Table 2), so within 1%.

- please be consistent with the symbols used - the text uses vapour pressure while table 3 shows specific humidity.

Revised, specific humidity throughout now, for the perturbations

- Please be explicit this is top of atmosphere solar variability.

This subsection has been rewritten, hopefully clear now.

- For comparing the relative importance of each variable - it would be more useful to present the individual sensitivities relative to 1 standard deviation perturbations in Table 3. Agreed – we have added the individual sensitivities to 1-σ perturbations and written this into the results. This also facilitates comparisons with Table 4 (modelled sensitivities).

- The way these variables have been perturbed is not meaningful as they are physically unrealistic. For example - some of the standard deviation in vapour pressure will be due to increases in relative humidity, but you also increase incoming shortwave in this experiment - which as you noted earlier is likely to decrease with increased relative humidity. Thus, the experiment is contradictory. Please exclude these last two lines in Table 3.

This is a fair point, we agree that it is inconsistent for all variables to change in the same sense. This was meant only to explore joint variability of multiple weather variables, as occurs in reality, but as implemented this was not meaningful. These two lines have been removed. It is better to explore meteorologically-meaningful covariability through NARR (section 6) or another means.

- It is still ambiguous which variables are held constant at their measured values and which are parameterised for each run (in particular incoming longwave and surface temperature). Please provide a comprehensive table.

We have clarified this in our numerical experiments and in the text. We now use the parameterized daily model for all experiments that we present; LW radiation, albedo, and surface temperature are all parameterized/internally calculated. The introduction hopefully clarifies this, ll.627-634.

- why were changes in incoming longwave not examined?

We now perturb the clearness index \tau, which jointly impacts incoming SW and LW radiation; experiments on the radiation fluxes are not considered. This is consistent with our effort to perturb observable weather variables (e.g., T, v, qv, clouds), and allows a sensible (albeit parameterized) co-variation of the incoming radiation fluxes.

- this result is likely to be strongly dependent on the choice of the minimum albedo of snow, which in this case does not differ much from an ice albedo. Either the sensitivity of this result should be tested, or a more thorough justification made for the very low value chosen here. (0.3)

Agreed and revised, as discussed above in some depth. The default minimum albedo is now 0.4, and Fig. 7c includes an illustration and brief discussion of the sensitivity to this parameter.

- are the anomalies calculated with respect to the mean in-situ conditions? I suspect you took the daily anomalies of NARR from the NARR climatology, then applied these daily anomalies to the mean in-situ conditions - please clarify.

Yes, this is correct. Clarified, l.751.

- it would be useful to see the standard deviation of relative humidity included here.

We don't actually use NARR relative humidity; it is derived from the specific humidity and temperature, for thermodynamic consistency. Hence any errors and variability in RH will flow from the NARR-derived T and qv. But for interest, the NARR RH is 65 \pm 3% (sigma = 2.8%).

- this line needs more context to link it with the previous sentences.

Revised, l.781.

- do the interannual anomalies in SWin and LWin from NARR correlate with the anomalies from the in-situ dataset? If not, it is hard to see how the NARR represents realistic interannual variability in these fluxes. This severely limits the inferences that can be made from model runs made with these anomalies.

We more or less agree – these are only weakly correlated, 0.52 for incoming SW radiation and 0.17 for incoming LW. Variables like temperature are much stronger (0.81). These are for only 11 years of mean JJA conditions, so the data is a bit limited. In summary, incoming SW is OK and incoming LW is very weak in the reanalysis. We note that NARR does not represent realistic interannual variability in these fluxes (l.765, l.779). Noted again in the discussion.

- this statement needs more justification. It would seem that the NARR based reconstructions performed satisfactorily in describing interannual variations in net energy flux, but that this is based on the accidental cancelation of errors in the radiative fluxes driving the model (Figures 5 and 6 from the original manuscript. The approach is worthy of further exploration, but a more thorough evaluation of the performance of the model, including biases and areas for improvement, is needed here.

Yes, agreed, this is still the case that further work is needed if one wishes to drive glacier mass balance reconstructions with NARR forcing. We deliberately stopped short of that here, and revised the discussion to stay within the sensitivity study and recommend further work, as per the reviewer's suggestion.

- needs a legend describing the colours. Also, box 4 could be better as a separate figure, as the colours indicate the change in Qn due to different forcing, while in the other boxes the colours show the response of different fluxes to the same forcing.

Legend now added and the figure has been simplified to show the same fluxes in each case.

- both black and red are listed as net radiation - please fix.

Revised.

- add 'please note the different y scales'.

Revised as suggested.

- please clarify in the caption which scenarios these figures relate to in Table 4.

Revised as suggested.

- net melt has decreased 7-8% while net energy fluxes have remained similar. It would be useful to show the fraction of time the surface is diagnosed as melting in each month to provide some justification here.

We no longer discuss the M-S conditions, to shorten and focus the ms. Agree though that this is interesting.

- I am not sure this figure adds much as you cannot see the detail in the daily values over the 11-year period, and the results presented in the figure are not discussed in the text. I would suggest either removing the figure, or modifying the figure so it is readable and discussing the results further. If the figure is kept the size of each box needs to be expanded and the line weight reduced to make a more readable figure.

Figure has been modified to better illustrate the data. It is meant to give a sense of the observed/driving data. It is now discussed a bit more in the main text, ll.383-387

- please use thinner lines on these figures. Also, as months and not day of year are discussed in the text, it would be good to have months as the x-axis label, or at the very least, further tick marks that are at monthly or 30 day intervals.

On panel d, the median + interquartile range would better present the seasonal variation of melt rate. As it is, the mean appears to be greatly influenced by individual large melt events (such as around day 230).

Revised as suggested, also panel d (now median and interquartile range). It is true in the early spring or late fall, the mean is influenced by large events, though not so much in JJA.

- the y axis of figure 7a should be m w.e. in line with the text and Figure 7c.

Revised as suggested

- 1495 - were these figures meant to be included?

Our mistake, apologies. Remnants of the first submission.

[revised manuscript text omitted]
 ~~change are amplified dramatically at this site: roughly a five-fold increase in the melt sensitivity, to ~50% $°C^{-1}$. This is equivalent to about $-1.2$ m w.e. $°C^{-1}$. The~~

relationship is nonlinear and is strongest near the observed present day temperature. Several effects can contribute to this strong amplification of the temperature signal in the melt model. A longer andinfluences the seasonal albedo evolution. A more intense melt season gives rise to a lower albedo through higher impurity concentration and water contentsnow albedo and an earlier transition from seasonal snow cover to glacial ice. These positive feedbacks also operate (in reverse) under a cool perturbation. We do not explicitly model impurities or snow-albedo processes (e.g., grain metamorphism, effects of snow-water content on the albedo), but we parameterize the seasonal albedo evolution as a function of cumulative *PDD* (Eq. 20). This is a rough proxy for cumulative melt effects that lower the albedo, and is empirically supported, but positive degree days are a direct function of temperature so this may make our albedo model overly20), which makes the model directly sensitive to temperature perturbations.

Temperature changes have several additional, indirect impacts, including: (i) a longer melt season, starting earlier and ending later, (ii) a greater fraction of time with surface temperatures at the melting point during the year, i.e., with reduced overnight cooling and refreezing, and (iii) an increase in the frequency of summer rain vs. snow events. Summer snow events have an important impact on surface albedo, with fresh snow strongly attenuating melt. Each of these processes contributes to the strong impact of increased temperaturestemperature anomalies on glacier melt. Combined with the albedo feedbacks, these processes and the model results help to explain why glaciers are sostrongly sensitive to temperature change, as they clearly are in natural settings (e.g., Marzeion et al., 2014)..

When multiple meteorological perturbations are introduced at the same time, in the NARR-based surface energy balance modelling, interannual temperature fluctuations appear to be weaker than the sensitivity experiments would suggest, ~14% °C⁻¹, although mean summer net energy and temperature are highly correlated ($r = 0.84$). All feedbacks discussed above are active in the NARR-based simulations. The impacts of temperature variability on net energy and melt could be partially compensated by other systematic changes in the energy budget. For instance, warm temperatures could be associated with calm, dry conditions that reduce the incoming longwave radiation and the turbulent fluxes. NARR mean summer temperature over the 36-year period is negatively correlated with wind speed ($r = -0.11$) and cloud cover ($r = -0.50$), which supports this possibility.

*Albedo Perturbations*

Direct changes to albedo have an influence on summer energy balance and melt extent that is comparable to the temperature influence. The three different methods of gauging albedo sensitivity give similar results, a summer energy balance impact of 22-26 W m⁻², ~17% for a change in albedo of 0.1 (Tables 3, 4 and 6). Interannual albedo fluctuations are associated with net energy and melt variations of about 12%, a large fraction of the interannual variability.

equal to the interannual albedo fluctuations, 0.06. Mean summer albedo differences arise as a feedback to other meteorological forcings that drive the summer snowpack evolution, such as temperature. Interannualsnow melt, but interannual albedo variations can also occur more directly, as a consequence of frequent summer snowfall events or, as a resultfunction of low or high winter accumulation totals, which influence how long the seasonal snowpack will persist through the summer. or due to impurity loading (e.g., black carbon deposition). The latter has been observed in association with forest fires in British Columbia. Strong fire seasons  occurred twice during our period of study , in 2003 and 2015, and each left a  measurably darker glacier surface. For instance, the average albedo recorded at the AWS site in August 2003 was 0.13.

**

 We found a  relatively

~~Mean summer latent heat flux was weakly negative through the observational period at the Haig Glacier AWS site, but increases in specific humidity increase this flux and it switches signs to a small positive flux with a 5% (~0.2 g kg$^{-1}$) increase in JJA humidity. Radiation fluxes are more strongly sensitive to humidity, at least as parameterized in our study, with relative humidity being the main influence. Atmospheric emissivity and incoming longwave radiation increase with humidity, while incoming shortwave radiation is reduced, based on the empirical link between relative humidity and cloud cover. These fluxes largely compensate and offset each other, but the longwave radiation has a stronger response in our results, for both the theoretical and modelled perturbations in humidity. There is a net cooling influence when specific humidity is reduced, as decreases in incoming longwave radiation exceed the attendant increases in shortwave radiation. Increases in humidity give an increase in net radiation, as gains in incoming longwave radiation again exceed the reductions in net shortwave radiation. The balance will depend on the surface albedo, which reduces the magnitude of shortwave radiation anomalies in the net energy budget.~~

**

~~Top-of-atmosphere shortwave radiation fluctuations, i.e. solar variability, have only minor influences on glacier melt, as top-of-atmosphere forcing is diminished through atmospheric extinction and the glacier surface albedo. Fluctuations of ~3 W m$^{-2}$ are attenuated to 1 W m$^{-2}$, which is negligible relative to the daily and interannual variability associated with cloud cover.~~

~~The latter does have a significant impact on year-to-year melt conditions. Surface-level interannual variability in shortwave radiation forcing equates to fluctuations of about 6 W m$^{-2}$ (6%) of the JJA net energy budget, and can compound the effects of warm temperatures in associated with hot, dry, clear-sky periods on the glacier. This is empirically borne out at the site, but shortwave radiation fluctuations are less important in the NARR-driven energy balance than they are in the observations. NARR shortwave radiation variations correlate positively but weakly with summer melt, and interannual variability of incoming shortwave radiation is muted in the reanalysis. The NARR dataset may not be picking up some of the persistent ridging conditions which are observed to drive strong summer melt events at the site.~~

*Winter Mass Balance*

We found only a minorweak influence of winter mass balance on the summer melt extent, based on observed interannual variability in winter snow accumulation as well as sensitivity experiments. . A low snowpack depth has a greater impact, through an earlier transition to low-albedo bare ice. A deep winter snowpack has the opposite influence, supporting a higher average summer albedo, but the influence is weaker because the AWS site is in the upper ablation area, where the seasonal snowpack persists until late summer in most years. The effects of greater winter accumulation plateau once there is enough snow to survive the summer; beyond this point, additional snow has no effect on the summer albedo or melt extent. Sensitivity to winter mass balance would likely be stronger at lower altitudes on the glacier, and for the overall glacier mass balance.

*Multivariate Perturbations*

Meteorological variables do not vary as idealistically as in the simple experiments presented in this paper. In reality, meteorological variables all vary at once, and different weather systems will have tendencies for the combined meteorological perturbations to compensate (buffer) or accentuate (amplify) impacts on energy balance and melt. This is implicit in the NARR-forced simulations, which sample a 36-year record of interannual variability with physically-consistent covariance of meteorological variables.

[revised manuscript text omitted]

We do not test the ability and skill of NARR-forced energy and mass balance reconstructions here. This requires further study. In general, the perturbation method eliminates biases in the mean NARR variables, but a realistic representation of the variability and long-term trends in reanalysis fields is important to realistic representations of the glacier mas balance record and meltwater runoff. It would be instructive to analyze the synoptic weather patterns and weather anomalies in high-melt vs. low-melt summers in the NARR-driven simulations. We recommend an investigation of specific weather systems and their associated meteorological and energy balance conditions in followup work.

at a particular site. The method could similarly be applied to climate model output for future projections.

*Representativeness of the Results*

We have designed the sensitivity approach and the model to be applicable in regional studies, e.g. in a distributed model of glacier energy balance, forced by climate model reanalyses or projections. However, we did not expand our scope to other sites within the present study. In principle, the theoretical sensitivities (i.e. from the same set of equations) could be calculated for different baseline meteorological conditions, such as maritime or tropical environments. The method, rather than the specific Haig Glacier results, could be exported to other glacierized environments.

At regional scales, Haig Glacier energy balance sensitivities might be more transferrable, since similar summer climate conditions prevail across the Canadian Rocky Mountains (Ebrahimi and Marshall, 2015). Regional, multi-year reconstructions of glacier meltwater runoff might be feasible through a perturbation approach to summer mass balance, driven by meteorological anomalies from station data or climate models. This needs to be tested, however, for sensitive parameterizations such as the albedo model. It is uncertain whether the Haig glacier bare-ice and old-snow albedo are regionally representative.

Within Haig Glacier itself, our AWS site is in the upper ablation area, near the equilibrium ELA. Results are specific to the snow and ice albedo, snowpack depth, and meteorological/energy balance conditions at this location. We have not examined the representativeness of the results to other parts of the glacier, but summer melt extent and mass balance at the AWS site are strongly correlated with glacier-wide mass balance. We recommend additional work to calculate an average set of glacier sensitivities and assess whether the values presented here are representative. We suspect that sensitivity of net energy to winter snow depth and the strength of albedo feedbacks will vary across the glacier.

*Recommended Model Improvements*

Model improvements are recommended with respect to our treatment of the glacier surface albedo and precipitation modelling.  The energy balance, albedo, and melt models perform well in the core summer melt season, June through August, when summer snowfall is infrequent and impacts on the albedo are transient. We systematically underestimate September albedo, however; better treatments of late-summer snow accumulation and the transition to the winter accumulation season are needed.

Our meltwater drainage model is also simplistic. We assume that water drains efficiently from the glacier surface, but in fact water has been observed to pond and refreeze on the surface. Re-melting of this superimposed ice consumes energy and reduces the total summer runoff .

A more realistic treatment of year-round snow accumulation is also needed in order to carry out model-based glacier mass balance reconstructions. We rely on observed winter mass balance for the studies here, but historical reconstructions and future projections require a way to reliably estimate snow accumulation from climate models. NARR precipitation in the Haig Glacier grid cell poorly represents the observed winter accumulation totals.

We have done tests to verify that the daily, parameterized model performs well relative to direct forcing with 30-minute AWS data, but some simplifications embedded in the daily model need to be examined. For instance, we assume constant cloud cover/clearness index over the day; systematic diurnal variations in cloud cover would affect the net radiation in ways that we do not capture. Overnight clouds serve to increase energy flux to the glacier, while daytime clouds reduce the incoming radiation. Effects like these become complicated to model or parameterize, but could bias our sensitivity results to cloud cover.

**8. Conclusions**

Sensitivity studies presented here extend the foundational work of Oerlemans and Fortuin (1992) and others, which has generally been done on glacier mass balance sensitivity to changes in temperature and precipitation. Our study is limited to summer mass balance at one location, but our results offer insight into  the influence of different meteorological variables and energy fluxes, their year-to-year variability, and the role of isolated vs. collective forcings, feedbacks , and interactions on summer melt extent.

There is a good correspondence between the theoretical sensitivities and those derived from  the numerical energy balance model, when feedbacks are omitted. This supports the potential application of the theoretical sensitivities to explore energy balance sensitivities under different climate regimes. This method can be transferred directly to other sites.

~~The model runs year-round, to simulate sub-surface temperature evolution in the winter snowpack and to include the complete summer melt season (May to September), but our analysis concentrates on mean summer (JJA) surface energy balance and melt. Just over 80% of the annual melt at the site occurs in JJA, and we find similar energy balance and melt sensitivity to meteorological variability when we look at MJJAS.~~

Temperature and albedo variations exert the strongest controls on year-to-year variability in summer melt at our site. While albedo can fluctuate independent of temperature, e.g., through the influence of the winter snowpack depth or aerosol loading, it is also a powerful feedback mechanism to temperature and melt season evolution. In our model, albedo feedbacks give a two-fold increase in the net energy balance sensitivity to a temperature perturbation, amplifying the summer melt response from 13% °C$^{-1}$ to ~28% °C$^{-1}$. Temperature and albedo fluctuations are also the strongest influences on interannual melt variations in the NARR-forced surface energy balance, but the melt sensitivity to temperature variations is about 15% °C$^{-1}$, weaker than our result from the control experiments. This may be because the co-variation of other variables in the surface energy balance partially offsets the temperature forcing. ~~For example, temperature increases are associated with lower relative humidity and cloud cover, which reduces incoming longwave radiation. It is also possible that NARR climate reconstructions are not adequately capturing the weather conditions and their interannual variability over the field site, as suggested by a poor representation of cloud conditions and radiation fluxes compared to in situ observations.~~

~~Other meteorological variables cannot be neglected in the surface energy balance and its interannual variability. At Haig Glacier, incoming shortwave radiation fluctuations are particularly influential on summer melt extent. The strongest melt seasons and most negative mass balance years on record at the site, 2003, 2006, and 2015 (not shown), were each associated with persistent anticyclonic ridging in the summer months, giving warm, dry, clear-sky conditions, i.e. co-variance of strong positive anomalies in temperature and incoming shortwave radiation.~~

[revised manuscript text omitted]
 0.0\cancel{04}05$ | $275 \pm 4$ | $311 \pm 1$ | $27 \pm 4$ | $-3 \pm 3$ | $2 \pm 1$ | $95 \pm 14$ | $2.28 \pm 0.3\cancel{2}42$ |
| MJJAS | $215 \pm 6$ | $0.55 \pm 0.0\cancel{03}04$ | $271 \pm 4$ | $308 \pm 2$ | $22 \pm 3$ | $-5 \pm 3$ | $3 \pm 1$ | $73 \pm 10$ | $2.\cancel{95}68 \pm 0.3\cancel{8}50$ |

**Table 6**. Correlation and sensitivity of different weather variables to the mean summer (JJA) net energy flux, $Q_N$, for the NARR simulations, 1979-2014. ‘cloud’ is the NARR total cloud fraction.

| Variable | Correlation | Sensitivity | $\delta Q_N$ for $+1\sigma$ |
|---|---|---|---|
| $T$ (°C) | 0.84 | $\partial Q_N/\partial T = 14$ W m$^{-2}$(°C)$^{-1}$ | $+10$ W m$^{-2}$ |
| $\cancel{h\ (\%)}$ | $\cancel{0.33}$ | $\cancel{\partial Q_N/\partial h = -2\ \text{W m}^{-2}(\%)^{-1}}$ | $\cancel{-6\ \text{W m}^{-2}}$ |
| $q_v$ (g kg$^{-1}$) | 0.50 | $\partial Q_N/\partial q_v = 25$ W m$^{-2}$(g/kg)$^{-1}$ | $+7$ W m$^{-2}$ |
| $v$ (m s$^{-1}$) | $-\cancel{0.07}00$ | $\partial Q_N/\partial v = -4$ W m$^{-2}$(m/s)$^{-1}$ | $-1$ W m$^{-2}$ |
| $Q_S^{\downarrow}$ (W m$^{-2}$) | 0.14 | $\partial Q_N/\partial Q_S^{\downarrow} = 0.3$ W m$^{-2}$ (W m$^{-2}$)$^{-1}$ | $+2$ W m$^{-2}$ |
| $Q_L^{\downarrow}$ (W m$^{-2}$) | 0.64 | $\partial Q_N/\partial Q_L^{\downarrow} = 2$ W m$^{-2}$ (W m$^{-2}$)$^{-1}$ | $+8$ W m$^{-2}$ |
| $\underline{\tau}$ | $\underline{0.25}$ | $\underline{\partial Q_N/\partial\tau = 15\ \text{W m}^{-2}\ (0.1)^{-1}}$ | $\underline{+4\ \text{W m}^{-2}}$ |
| $\underline{\text{cloud}}$ | $\underline{-0.19}$ | $\underline{\partial Q_N/\partial c = -8.1\ \text{W m}^{-2}\ 0.1)^{-1}}$ | $\underline{-3\ \text{W m}^{-2}}$ |

| | | | |
|---|---|---|---|
| $\alpha_S$ | −0.83 | $\partial Q_N/\partial\alpha_S = -26\ \text{W m}^{-2}\,(0.1)^{-1}$ | −11 W m$^{-2}$ |
| $b_w$ (m w.e.) | −0.15 | $\partial Q_N/\partial b_w = -3\ \text{W m}^{-2}\,(\text{m w.e.})^{-1}$ | −1 W m$^{-2}$ |

**Figures**

[Figure]

**Figure 1.** Idealized diurnal cycles of (a) temperature and (b) incoming shortwave radiation used in the energy balance model. These two examples are for a sample day, July 1, 2010, parameterized from daily minimum and maximum temperature in (a) and day of year plus mean daily incident shortwave radiation in (b).

[Figure]

[Figure]

**Figure 2.** (a) The topography and automatic weather stations on Haig Glacier (GAWS) and the glacier forefield (FFAWS). The smaller black dots are mass balance survey points. (b) The location of Haig Glacier is labelled HG on the Google Earth map of southwestern Canada.

[Figure]

[Figure]

**Figure 3.** The 11-year record of (a) air temperature, modelled surface temperature, and (b) surface
energy fluxes at the Haig Glacier AWS site. Daily mean values are plotted from Jan 1, 2002-Dec
31, 2012.

[Figure]

[Figure]

**Figure 4.** The average annual cycle of (a-c) surface energy fluxes and (d) daily melt at the Haig Glacier AWS. Daily mean values are plotted for the period 2002-2012. For melt rates,  the heavy line is the median value and the thin lines indicate the interquartile range.

[Figure]

[Figure]

**Figure 5**. Sensitivity of modelled summer (JJA) melt to temperature perturbations for different assumptions, as per Table 4.

Black line: relative humidity is constant, including albedo feedbacks. Red line: specific humidity is constant, no atmospheric feedbacks. Blue line: specific humidity is constant, including atmospheric feedbacks. The reference The reference (mean 2002-2012) JJA melt is 2.32 m w.e.

[Figure]

[Figure]

**Figure 6**. Sensitivity of the surface energy fluxes at Haig Glacier to  changes in (a) temperature (case 2), (b) specific humidity (case 6), (c) wind speed (case 7), and (d) atmospheric transmittance (case 8) and albedo (blue line, case 9). All lines are anomalies relative to the baseline data from the period 2002-2012, and indicate the mean sensitivity of the different energy fluxes over this period. Please note the different $y$ ($\delta Q$) scales.

[Figure]

**Figure 7**. Sensitivity to the winter mass balance , examined by varying May 1 snow depth from 0.36-2.36 m w.e., relative to the reference value of 1.36 m w.e. at the glacier AWS. (a) Snow depth  and (b) albedo through the summer melt season, May 1-Sept 30, for the different   initial snow depths. (c) Net summer (JJA) energy balance change as a function of the winter mass balance for two different settings of the minimum snow albedo.

[Figure]

[Figure]

**Figure 8**. a) Mean summer (JJA) NARR-forced surface energy fluxes at Haig Glacier, 1979-2014.  Mean summer net energy as a function of (b) temperature  and specific humidity  (c)  albedo, and (d)  incoming shortwave  and longwave radiation . Table 6 gives the associated correlations.

[Figure]

**Figure 5**.

[Figure]

**Figure 6**. The evolution of modelled summer surface energy balance and melt from the
perturbed NARR output (blue), 1979-2014, and from the in situ data (black), 2002-2012. (a)
albedo, (b) net radiation, (c) sensible heat flux, (d) latent heat flux, (e) net energy, and (f) total
summer melt (mm). All fields are for MJJAS except for albedo, which is shown for JJA, the

[Figure]

main melt season.

**Figure 9**. Net energy sensitivity to a 1-σ perturbation in different meteorological variables:
comparison of theoretical, *in situ* numerical model, and NARR-based estimates.